# Deep Transformers without Shortcuts: Modifying Self-attention for Faithful Signal Propagation

**Bobby He**[1]  **James Martens**[2]  **Guodong Zhang**[2]  **Aleksandar Botev**[2]
**Andrew Brock**[2]  **Samuel L. Smith**[2]  **Yee Whye Teh**[1,2]
[1]University of Oxford, [2]DeepMind
Correspondence to: `bobby.he@stats.ox.ac.uk,jamesmartens@google.com`.

## Abstract

Skip connections and normalisation layers form two standard architectural components that are ubiquitous for the training of Deep Neural Networks (DNNs), but whose precise roles are poorly understood. Recent approaches such as Deep Kernel Shaping have made progress towards reducing our reliance on them, using insights from wide NN kernel theory to improve signal propagation in vanilla DNNs (which we define as networks without skips or normalisation layers). However, these approaches are incompatible with the self-attention layers present in transformers, whose kernels are intrinsically more complicated to analyse and control. And so the question remains: *is it possible to train deep vanilla transformers?* We answer this question in the affirmative by designing several approaches that use combinations of parameter initialisations, bias matrices and location-dependent rescaling to achieve faithful signal propagation in vanilla transformers. Our methods address several intricacies specific to signal propagation in transformers, including the interaction with positional encoding and causal masking. In experiments on WikiText-103 and C4, our approaches enable deep transformers without normalisation to train at speeds matching their standard counterparts, and deep vanilla transformers to reach the same performance as standard ones after about 5 times more iterations.

## 1 Introduction

Despite numerous impressive successes, the practice of training deep neural networks (DNNs) has progressed to a large extent independently of theoretical justification. Most successful modern DNN architectures rely on particular arrangements of skip connections and normalisation layers, but a general principle for how to use these components in new architectures (assuming they are even applicable) remains unknown, and their roles in existing ones are still not completely understood.

The *residual architecture*, arguably the most popular and successful of these, was first developed in the context of convolutional networks (CNNs) (He et al., 2016), and later in self-attention networks yielding the ubiquitous transformer architecture (Vaswani et al., 2017). One proposed explanation for the success of residual architectures is that they have superior signal propagation compared to vanilla DNNs (e.g. Balduzzi et al., 2017; Xiao et al., 2018; Hayou et al., 2019; De & Smith, 2020; Martens et al., 2021), where *signal propagation* refers to the transmission of geometric information through the layers of a DNN, as represented by a kernel function (Daniely et al., 2016; Poole et al., 2016; Schoenholz et al., 2017).

Recently, using signal propagation principles to train DNNs at high depths, without the skip connections and/or normalisation layers found in residual architectures, has become an area of interest in the community. The reasons are two-fold. First, it would validate the signal propagation hypothesis for the effectiveness of residual architectures, thus clarifying our understanding of DNN trainability. And second, it could lead to general principles and techniques for achieving trainability in DNNs beyond the residual paradigm, with the potential for improved or more efficient architectures.

For CNNs, Xiao et al. (2018) showed that improved signal propagation from better initialisation enables very deep vanilla networks to be effectively trained, although at significantly reduced speeds

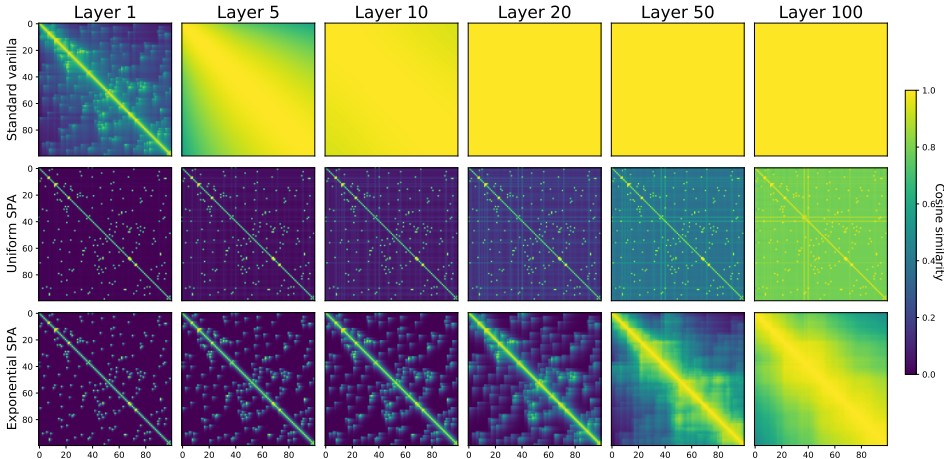

**Figure 1:** Normalised kernel matrices $\mathrm{diag}(\Sigma_l)^{-\frac{1}{2}} \cdot \Sigma_l \cdot \mathrm{diag}(\Sigma_l)^{-\frac{1}{2}}$ (which are like kernel matrices except with cosine similarities instead of inner-products) at various depths for standard attention-only vanilla transformers and two of our proposed alternatives (Section 3). Standard attention-only vanilla transformers (top) quickly suffer from rank collapse where all entries of the normalised kernel converge to 1, whereas our approaches, U-SPA and E-SPA, maintain controlled signal propagation even at large depths. Moreover, our main method E-SPA (bottom) exhibits a *recency bias*, where cosine similarities corresponding to nearby pairs of locations are larger, akin to positional encoding. Equivalent plots for attention-only transformers with skips and normalisation can be found in Fig. 7.

compared to residual networks. Martens et al. (2021) later proposed Deep Kernel Shaping (DKS) which uses activation function transformations to control signal propagation, achieving training speed parity between vanilla and residual networks on ImageNet assuming the use of strong 2nd-order optimizers like K-FAC (Martens & Grosse, 2015). Zhang et al. (2022) extended ideas from DKS to a larger class of activation functions, achieving near parity in terms of generalisation as well.

The key quantity that is analysed in signal propagation is the DNN's initialisation-time kernel, or more precisely, the approximate kernel given by the infinite width limit (Neal, 2012; Matthews et al., 2018; Lee et al., 2018; Yang, 2019). For MLPs, and for CNNs that use a Delta-initialisation (Balduzzi et al., 2017; Xiao et al., 2018), this kernel can be written as a simple recursion over layers that involves only 2D functions, facilitating a straightforward analysis.

Unfortunately, the evolution of the kernel across layers of a transformer is more complicated, and as a result, existing approaches like DKS are not applicable to transformers (or indeed any architecture that contains self-attention layers). More concretely, if $\mathbf{X}_l \in \mathbb{R}^{T \times d}$ denotes a length-$T$ sequence of activations at layer $l$ of a transformer, then the *kernel matrix* $\Sigma_l = \mathbf{X}_l \mathbf{X}_l^\top / d \in \mathbb{R}^{T \times T}$ for layer $l$ (or more precisely its limit as $d \to \infty$), can be written as a function of the kernel matrix $\Sigma_{l-1}$ of the previous layer (Hron et al., 2020). In the case of self-attention layers, the relationship of $\Sigma_l$ on $\Sigma_{l-1}$ cannot be simplified or decomposed into lower dimensional functions, leading to a recursion that is intrinsically high dimensional and harder to analyse or control.

Analogously to the case of MLPs, where signal propagation is judged by looking at the behavior of the (one-dimensional) kernel, signal propagation in transformers can be judged by looking at the evolution of these (high-dimensional) kernel matrices through the layers of the network. One situation we must avoid is where the diagonal entries rapidly grow or shrink with depth, which corresponds to uncontrolled activation norms and can lead to saturated losses or numerical issues. A more subtle form of signal degradation can occur where $\Sigma_l$ converges to a rank-1 matrix, which is known as *rank collapse* (Dong et al., 2021). Dong et al. (2021) showed that skip connections are essential to avoid the collapsed state: skipless transformers quickly converge to rank collapse at large depths, which we corroborate in Fig. 1 (top). Moreover, Noci et al. (2022) showed that rank collapse may lead to zero gradients for certain parameters in attention layers, hindering the trainablility of deep transformers. Thus, avoiding rank collapse is necessary for deep transformers to be trainable, and the question of whether one can train deep skipless transformers remains open.

In the present work we address this question, demonstrating for the first time that it is possible to successfully train deep transformers without skip connections or normalisation layers. To do so, we study the problem of signal propagation and rank collapse in deep skipless transformers, and derive three approaches to prevent it in Section 3. Our methods use combinations of: 1) parameter initialisations, 2) bias matrices, and 3) location-dependent rescaling, and highlight several intricacies

specific to signal propagation in transformers, including the interaction with positional encoding and causal masking. In Section 4, we empirically demonstrate that our approaches result in trainable deep skipless transformers. On WikiText-103 and C4 datasets we show that using our main approach, *Exponential Signal Preserving Attention* (E-SPA), it is possible to match the training loss of standard transformers with our skipless ones by training for around 5 times longer. Moreover, by combining this approach with skip connections, we show that transformers without normalisation layers are able to match the training *speed* of standard ones.

## 2 PROBLEM SETTING

**Transformer models** The input to a transformer consists of a sequence $\boldsymbol{x} = (x_i)_{i=1}^{T}$ over $T$ locations consisting of tokens from a vocabulary $V$: $x_i \in \{1, \ldots, |V|\}$. The model takes this sequence, and using a trainable embedding matrix $\mathbf{E} \in \mathbb{R}^{|V| \times d}$, creates a matrix of vector representations $\mathbf{X}_0 \in \mathbb{R}^{T \times d}$ by performing a direct look-up for each location: $[\mathbf{X}_0]_i = \mathbf{E}_{x_i} \in \mathbb{R}^d$.

After this, the sequence is successively transformed via a series of $L$ "transformer blocks", with independently initialised parameters. We will denote by $\mathbf{X}_l \in \mathbb{R}^{T \times d}$ the output sequence for block $l$, and will sometimes refer to the rows of $\mathbf{X}_l$ as *representation vectors*. Each transformer block consists of two component blocks that both employ a standard residual structure (He et al., 2016), computing the sum of a residual branch which performs the main computation, and a shortcut branch (aka a skip connection) which just copies the block's inputs to its output. The attention block, which is the first of these component blocks, applies layer normalisation (LN) (Ba et al., 2016) or RM-SNorm (Zhang & Sennrich, 2019), followed by multi-head attention (MHA), on its residual branch. The MLP block, which comes next, applies LN followed by a standard (typically shallow) MLP on its residual branch. The LN and MLP operations are applied to each sequence element independently (using shared parameters), so that information is only communicated between sequence elements via the MHA operation (which we will define later). In summary we have

$$\mathbf{X}_l = \alpha \mathbf{X}_{l-1} + \beta \, \mathrm{MHA}(\mathrm{RMSNorm}(\hat{\mathbf{X}}_{l-1})) \quad \text{and}$$
$$\hat{\mathbf{X}}_l = \alpha \mathbf{X}_l + \beta \, \mathrm{MLP}(\mathrm{RMSNorm}(\mathbf{X}_l)), \tag{1}$$

where the shortcut and residual weights $\alpha, \beta$ are typically both 1. In this work we will focus on *skipless* transformers with $\alpha = 0$ and $\beta = 1$, and on *vanilla* transformers, which are skipless transformers without normalisation layers. For simplicity we will also devote much of our analysis to *attention-only* models, which are transformers without MLP blocks (so that $\hat{\mathbf{X}}_l = \mathbf{X}_l$).

Note that we restrict our analysis to decoder-only transformers in this work, as they have a simpler structure which is easier to analyse, and are widely used in practice. Also note that Eq. (1) corresponds to the "Pre-LN" (Baevski & Auli, 2018; Child et al., 2019) rather than the original "Post-LN" transformer (Wang et al., 2019). In Post-LN transformers, the normalisation operation is applied at the output of each MLP and attention block instead of at the beginning of each residual branch.

**Self-attention** Given an input sequence $\mathbf{X} \in \mathbb{R}^{T \times d}$, the self-attention mechanism computes

$$\mathrm{Attn}(\mathbf{X}) = \mathbf{A}(\mathbf{X})\mathbf{V}(\mathbf{X}), \quad \text{with } \mathbf{A}(\mathbf{X}) = \mathrm{softmax}\left(\frac{1}{\sqrt{d^k}}\mathbf{Q}(\mathbf{X})\mathbf{K}(\mathbf{X})^{\top}\right), \tag{2}$$

where the softmax function is applied row-wise. $\mathbf{Q}(\mathbf{X}) = \mathbf{X}\mathbf{W}^Q$, $\mathbf{K}(\mathbf{X}) = \mathbf{X}\mathbf{W}^K$ and $\mathbf{V}(\mathbf{X}) = \mathbf{X}\mathbf{W}^V$ denote the *queries*, *keys* and *values* respectively, with trainable parameters $\mathbf{W}^Q, \mathbf{W}^K \in \mathbb{R}^{d \times d^k}$ and $\mathbf{W}^V \in \mathbb{R}^{d \times d^v}$. In practice, the attention mechanism, Eq. (2), is applied over $h$ "heads" (with independent parameters), giving rise to so-called *multi-head attention*:

$$\mathrm{MHA}(\mathbf{X}) \triangleq \mathrm{Concat}\big(\mathrm{Attn}_1(\mathbf{X}), \ldots, \mathrm{Attn}_h(\mathbf{X})\big)\mathbf{W}^O, \tag{3}$$

where $\mathbf{W}^O \in \mathbb{R}^{hd^v \times d}$ are trainable parameters and usually $d^k = d^v = \frac{d}{h}$. In this case, we can define $\mathbf{W}^V = \mathrm{Concat}(\mathbf{W}_1^V, \ldots, \mathbf{W}_h^V) \in \mathbb{R}^{d \times d}$ where $\mathbf{W}_n^V$ denotes the value parameters for head $n$.

We focus our investigation to models that perform *next-token prediction*: at location $i-1$ the model outputs a prediction over the identity of the $i^{\text{th}}$ target token, but using only information from input tokens 1 through $i-1$. This corresponds to using a causal masked attention with mask $\mathbf{M} \in \mathbb{R}^{T \times T}$ satisfying $\mathbf{M}_{i,j} = \mathbb{1}\{i \geq j\}$, where the attention matrix $\mathbf{A}$ in Eq. (2) is replaced with

$$\mathbf{A}(\mathbf{X}) = \mathrm{softmax}\left(\mathbf{M} \circ \frac{1}{\sqrt{d^k}}\mathbf{Q}(\mathbf{X})\mathbf{K}(\mathbf{X})^{\top} - \Gamma(1 - \mathbf{M})\right), \tag{4}$$

where $\Gamma$ is a large positive constant that zeros the attention coefficients corresponding to future tokens, making $\mathbf{A}$ a lower triangular matrix.

**Signal propagation in transformers**   As discussed in Section 1, Dong et al. (2021) showed that deep skipless transformers suffer from *rank collapse*, where the kernel matrix converges in depth to have rank 1, and Noci et al. (2022) showed that rank collapse can prevent trainability.

Moreover, Noci et al. (2022) demonstrated that rank collapse in transformers can occur in the absence of normalisation layers even *with* skip connections, and that downscaling the residual branch by setting $\beta = \frac{1}{\sqrt{L}}$ can alleviate this issue. This latter observation is in line with previous findings concerning the benefits of downscaling residual weights in ResNets (Hanin & Rolnick, 2018; Zhang et al., 2018; Arpit et al., 2019; Hayou et al., 2021; Bachlechner et al., 2021) and transformers (Zhang et al., 2019; Xu et al., 2020; Huang et al., 2020; Touvron et al., 2021; Wang et al., 2022). Davis et al. (2021) showed that concatenation acts similarly to a downweighted skip as an alternative way to connect skip and residual branches. De & Smith (2020) noted that the interaction of standard skip connections and normalisations can also effectively downweight the residual branch to give better signal propagation properties, but only if the normalisation layer is placed on the residual branch, like in Pre-LN transformers. Such an effect does not occur for Post-LN transformers, where the normalisation layer is after the residual branch, and we indeed observe in Fig. 7 that Post-LN attention-only transformers also suffer from rank collapse at large depths. This may explain some of the training instabilities of Post-LN transformers that have been observed in practice (Xiong et al., 2020; Liu et al., 2020).

## 3   CONSTRUCTING TRAINABLE DEEP TRANSFORMERS WITHOUT SHORTCUTS

To date, the only strategy for rectifying rank collapse in transformers relies on skip/shortcut connections, which "skip" around the trainability issues intrinsic to self-attention layers. We seek instead to tackle this issue directly. To do so, we first develop a better understanding of signal propagation through attention layers, then derive modifications from our insights to achieve faithful signal propagation in deep transformers, allowing them to be trained regardless of the use of skip connections.

To start with, we consider the simplified setting of a deep attention-only vanilla transformer, and suppose we are in a single-head setting ($h = 1$) or a multi-head setting where the attention matrix $\mathbf{A}$ does not vary across heads. If block $l \leq L$ has attention matrix $\mathbf{A}_l$ at initialisation, then the final block's representation $\mathbf{X}_L$ takes the following form:

$$\mathbf{X}_L = [\mathbf{A}_L \mathbf{A}_{L-1} \dots \mathbf{A}_1] \mathbf{X}_0 \mathbf{W}, \tag{5}$$

where $\mathbf{W} = \prod_{l=1}^{L} \mathbf{W}_l^V \mathbf{W}_l^O \in \mathbb{R}^{d \times d}$ can be made to be orthogonal at initialisation (so that $\mathbf{W}^\top \mathbf{W} = \mathbf{I}_d$), if each $\mathbf{W}_l^V$ and $\mathbf{W}_l^O$ are orthogonally initialised (assuming $d^v = \frac{d}{h}$). Going forward we will assume such an orthogonal initialisation, providing an ablation study in Fig. 9.

In that case, if we denote by $\Sigma_0 = \mathbf{X}_0 \mathbf{X}_0^\top \in \mathbb{R}^{T \times T}$ the input kernel matrix across locations, and $\Pi_l = \mathbf{A}_l \mathbf{A}_{l-1} \dots \mathbf{A}_1$ to be the product of attention matrices up to the $l^{\text{th}}$ block, then the location-wise kernel matrix $\Sigma_l = \mathbf{X}_l \mathbf{X}_l^\top \in \mathbb{R}^{T \times T}$ at block $l$ simplifies to[1]

$$\Sigma_l = \Pi_l \cdot \Sigma_0 \cdot \Pi_l^\top. \tag{6}$$

From this simplified formula for kernel matrices in deep attention-only transformers, we identify three requirements on $(\mathbf{A}_l)_l$:

(i)  $\Sigma_l = \Pi_l \cdot \Sigma_0 \cdot \Pi_l^\top$ must be well-behaved at each block, avoiding degenerate situations such as rank collapse and exploding/vanishing diagonal values.

(ii)  $\mathbf{A}_l$ must be elementwise non-negative $\forall l$ (recalling that $\mathbf{A}_l$ is constructed through the softmax operation Eq. (4)).

(iii)  $\mathbf{A}_l$ should be lower triangular $\forall l$, for compatibility with causal masked attention.[2]

In Sections 3.1 and 3.2, we focus on finding attention matrices that satisfy our desiderata above, and demonstrate how to modify softmax attention to achieve these attention matrices in Section 3.3.

---

[1]We note that this formula for the kernel matrix is exact even at finite widths, if $\mathbf{W}$ is orthogonal at initialisation. This is in contrast to the standard kernel correspondence of NNs at initialisation, where the usual kernel equations are only approximations at finite width that become increasingly accurate as width increases (Neal, 2012; Daniely et al., 2016; Yang, 2019; Martens, 2021; Li et al., 2022).

[2]We describe compatibility of our methods with non-causal attention in Appendix A.

## 3.1 IDENTITY ATTENTION: ISSUES AND VALUE-SKIPINIT

An obvious solution to our above requirements on $(\mathbf{A}_l)_l$ is the trivial one: $\mathbf{A}_l = \mathbf{I}_T \ \forall l$, where each sequence location attends only to itself. In this case, $\Sigma_L = \Sigma_0$ perfectly preserves the input kernel matrix, and will avoid rank collapse assuming $\Sigma_0$ is non-degenerate. Unfortunately, identity attention isn't compatible with a viable solution to obtain trainable vanilla transformers. This is because to achieve $\mathbf{A}_l = \mathbf{I}$ we would need to saturate the softmax operation (Eq. (2)) so that gradients do not pass to the query and key parameters, and the attention matrix stays close to identity during training.

To provide a partial solution to this that achieves identity attention matrix at initialisation yet is still trainable, we introduce our first approach, *Value-SkipInit*, based on SkipInit (De & Smith, 2020), and the related ReZero method (Bachlechner et al., 2021). In Value-SkipInit, we modify the attention operation $\text{Attn}(\mathbf{X}) = \mathbf{A}(\mathbf{X})\mathbf{V}(\mathbf{X})$, to

$$\text{Attn}(\mathbf{X}) = \big(\alpha\mathbf{I} + \beta\mathbf{A}(\mathbf{X})\big) \cdot \mathbf{V}(\mathbf{X}) \tag{7}$$

with trainable parameters $\alpha$ and $\beta$ that are initialised to 1 and 0 respectively. Thus, at initialisation, the attention matrix is the identity. For transformers with MLPs blocks, this yields identical behaviour to a standard MLP acting on each sequence location independently at initialisation, so we can apply the DKS or TAT frameworks (Martens et al., 2021; Zhang et al., 2022) to achieve well-behaved signal propagation in the entire model.[3] We note that Value-SkipInit is an approach to remove the standard skip connections in transformer blocks, Eq. (1), but can be construed as adding a skip connection around the value computation, and in that respect isn't strictly "skipless". Moreover, there is useful information contained in the positions of sequence locations that we do not employ when all attention matrices are identity at initialisation. As a result, we treat Value-Skipinit as a baseline for our main two methods, which we describe next.

## 3.2 SIGNAL PRESERVING ATTENTION METHODS

Returning to our requirements on $(\mathbf{A}_l)_l$ in Section 3, we see that controlling their product is key to achieving faithful signal propagation. Given that we are now interested in non-identity $(\mathbf{A}_l)_l$, this becomes more difficult. To overcome this, we consider $\mathbf{A}_l$ of the form

$$\mathbf{A}_l = \mathbf{L}_l\mathbf{L}_{l-1}^{-1} \tag{8}$$

such that individual $\mathbf{L}_i$'s cancel in the product, giving $\Pi_l = \mathbf{L}_l\mathbf{L}_0^{-1}$. Then if $\mathbf{L}_0$ satisfies $\mathbf{L}_0^{-1}\Sigma_0\mathbf{L}_0^{-1\top} = \mathbf{I}_T$, we thus have

$$\Sigma_l = \mathbf{L}_l\mathbf{L}_l^\top. \tag{9}$$

Assuming input embeddings are initialised independently, and no repeated tokens in the input sequence, we have $\Sigma_0 = \mathbf{I}_T$ in the large width limit,[4] and can thus take $\mathbf{L}_0 = \mathbf{I}_T$. In practice, we will apply slight modifications to our methods to account for repeated tokens, as detailed in Appendix B.

Now if $\mathbf{L}_l$ is lower triangular, then from Eq. (9) we see it is simply a Cholesky factor for the kernel matrix $\Sigma_l$. By the uniqueness of Cholesky factors for PSD matrices up to sign, this means we just need to choose $\{\Sigma_l\}_{l=0}^L$ to be a family of well-behaved kernel matrices that satisfy[5] our non-negativity constraint (ii) on $\mathbf{A}_l = \mathbf{L}_l\mathbf{L}_{l-1}^{-1}$. We identify two such families, which give rise to our main method, *Signal Preserving Attention* (SPA):

1. **Uniform** (U-SPA): $\Sigma_l(\rho_l) = (1 - \rho_l)\mathbf{I}_T + \rho_l\mathbf{1}\mathbf{1}^\top$. Here, the kernel matrix $\Sigma_l(\rho_l)$ has diagonal entries equal to 1 and off-diagonal entries equal to $\rho_l$. The condition $\rho_l \leq \rho_{l+1}$ is required for elementwise non-negativity of the $\mathbf{A}$'s, as shown in Theorem 1. Setting $\rho_0 = 0$ yields identity input kernel matrix, and as long as $\rho_L < 1$, we avoid rank collapse in the skipless attention-only setting.

2. **Exponential** (E-SPA): $\big(\Sigma_l(\gamma_l)\big)_{i,j} = \exp(-\gamma_l|i - j|)$. Here, diagonal entries are again 1, but now off-diagonals decay exponentially for more distant locations, with decay rate $\gamma_l$. Thus, unlike U-SPA, E-SPA captures the notion of *positional encoding*, as the vector representations for nearby locations have larger inner products (i.e. are more similar to each other). The condition $\gamma_l \geq \gamma_{l+1}$ is required for elementwise non-negativity of the $\mathbf{A}$'s, as established in Theorem 2. Setting $\gamma_0 = \infty$ yields the identity input kernel matrix, and rank collapse will be prevented as long as $\gamma_L > 0$.

---

[3]This is similar in principle to the Delta-initialisation for CNNs (Balduzzi et al., 2017; Xiao et al., 2018), and modifications to GNNs to enable compatibility with TAT (Zaidi et al., 2022).

[4]Because the embedding matrix $\mathbf{E}$ is initialised with variance 1/fan-out we rescale the embeddings it produces by $\sqrt{d_0}$ to get a $\Sigma_0$ with ones on the diagonal.

[5]Constraint (iii) is satisfied as lower triangular matrices are closed under multiplication and inversion.

We state and prove Theorems 1 and 2 in Appendix I, and in particular provide a closed-form solution for $\mathbf{L}_l\mathbf{L}_{l-1}^{-1}$ in the case of E-SPA in Theorem 3, enabling cheap computation. From these theorems, we see that our proposed SPA approaches are viable as long as the kernel matrix values become progressively larger (in an elementwise fashion) as depth increases, as dictated by $(\rho_l)_{l=1}^L$ and $(\gamma_l)_{l=1}^L$. We find $\rho_L = 0.8$ and $\gamma_L = 0.005$ to be good default choices for the final block, and describe how to vary $\rho_l$ and $\gamma_l$ across depth in Appendix C. In Alg. 1 in Appendix D, we summarise how to construct a trainable self-attention layer with our main method E-SPA using ideas from Section 3.3, given an input-output decay rate pair $\gamma_{\text{in}}, \gamma_{\text{out}}$ (in the notation of Alg. 1). Given a decreasing set of decay rates $(\gamma_l)_{l=0}^L$, at block $l$ we set $\gamma_{\text{in}} = \gamma_{l-1}$ and $\gamma_{\text{out}} = \gamma_l$.

In Fig. 1, we verify that our two proposed SPA schemes, U-SPA and E-SPA, successfully avoid rank collapse in attention-only vanilla transformers even at large depths. Moreover, because $\Sigma_l$ has diagonals equal to 1 for all $l$, there is an implicit mechanism in these two schemes to control the representation vector norms across *all* sequence locations at deep layers. This means that they can be used with or without normalisation layers, as we will verify empirically in Section 4. Moreover, in Fig. 1 we see that E-SPA observes a recency bias as expected, where representation vectors for nearby locations have larger cosine similarity, as seen with positional encoding schemes like ALiBi (Press et al., 2022) (Fig. 7). As a result, even though all three of our approaches successfully avoid rank collapse, we expect E-SPA to outperform both U-SPA and Value-SkipInit.

## 3.3 REVERSE ENGINEERING SELF-ATTENTION LAYERS AT INITIALISATION

In Section 3.2 we identified two families of lower-triangular non-negative attention matrices $(\mathbf{A}_l)_l$ which enable us to obtain well-behaved kernel matrices $\Sigma_L$ at large depth $L$, and hence faithful signal propagation. It remains to show how we can actually realise these attention matrices via parameter initialisation and minor modifications of the attention mechanism.

More precisely, we will show how for any given lower-triangular $\mathbf{A} \in \mathbb{R}^{T \times T}$ with non-negative entries, the masked softmax self-attention layer Eq. (4) can be initialised and augmented such that its output is exactly $\text{Attn}(\mathbf{X}) = \mathbf{AV}(\mathbf{X})$ at initialisation. To do so, we first define matrices $\mathbf{D}, \mathbf{P} \in \mathbb{R}^{T \times T}$ such that $\mathbf{A} = \mathbf{DP}$, where $\mathbf{D}$ is diagonal with positive entries and $\mathbf{P}$ is lower triangular with row sums 1. Then if $\mathbf{B} = \log(\mathbf{P})$ and $\mathbf{M}$ is the causal mask $\mathbf{M}_{i,j} = \mathbb{1}\{i \geq j\}$, we set[6]

$$\text{Attn}(\mathbf{X})=\mathbf{DP}(\mathbf{X})\mathbf{V}(\mathbf{X}), \ \& \ \mathbf{P}(\mathbf{X})=\text{softmax}\left(\mathbf{M}\circ\left[\frac{1}{\sqrt{d^k}}\mathbf{Q}(\mathbf{X})\mathbf{K}(\mathbf{X})^\top+\mathbf{B}\right]-\Gamma(1-\mathbf{M})\right) \quad (10)$$

which reduces to $\text{Attn}(\mathbf{X}) = \mathbf{AV}(\mathbf{X})$ (with a data-independent attention matrix) at initialisation if the query-key dot product $\frac{1}{\sqrt{d^k}}\mathbf{Q}(\mathbf{X})\mathbf{K}(\mathbf{X})^\top = 0$. We note that zero query-key dot products occur if one uses a $\frac{1}{d^k}$ scaling rather $\frac{1}{\sqrt{d^k}}$ in the infinite width limit for independently initialised $\mathbf{W}^Q, \mathbf{W}^K$ (Yang, 2019). In practice, to achieve zero initial query-key dot product, we can initialise either: 1) $\mathbf{W}^Q = 0$, 2) $\mathbf{W}^K = 0$, or 3) both $\mathbf{W}^Q$ and $\mathbf{W}^K$ to have small initial scale (which achieves *approximately* zero dot product). In our experiments we found these three options to all perform similarly, and decided to use option 1): $\mathbf{W}^Q = 0$. An empirical evaluation of the sensitivity to the choice of initial scale in option 3) is provided in Fig. 8.

In Eq. (10), $\mathbf{D}$ acts as a non-trainable *location-dependent rescaling*, and is needed to realise arbitrary attention matrices since the softmax output $\mathbf{P}(\mathbf{X})$ is constrained to have row sums equal to 1.[7] Given target kernel matrices with 1's on the diagonal (which we have in SPA) its inclusion is akin to using RMSNorm at initialisation. However, this will gradually stop being true during training, leading to a (slightly) different model class. The additive pre-softmax biases $\mathbf{B}$ will also have an effect on the model class, but this will be similar to that of the ALiBi positional encoder (Press et al., 2022), which too involves a non-trainable bias matrix being added to the logits. In our early experiments we tried including a trainable gain parameter on the bias matrix, initialised to 1, in order to better preserve the model class, but found that this didn't have a significant impact on training performance.

---

[6]We take $\log(0) = -\infty$ or a large negative constant, e.g. $-10^{30}$, in practice.

[7]In SPA, $\mathbf{D}$ also corrects for the fact that masked softmax attention tends to reduce the norms of representation vectors for locations at the *end* of the sequence. To see this, if we have kernel matrix $\Sigma = \mathbf{XX}^\top/d$ with $\Sigma_{ii} = 1, \ \forall i$, and softmax attention matrix $\mathbf{A}$ with row $i \ \boldsymbol{a}_i$, then $(\mathbf{A}\Sigma\mathbf{A}^\top)_{ii} = \frac{1}{d}\|\boldsymbol{a}_i\mathbf{X}\|_2^2 \leq \frac{1}{d}\|\mathbf{X}\|_2^2\|\boldsymbol{a}_i\|_2^2 = \|\boldsymbol{a}_i\|_2^2$. But $\boldsymbol{a}_i$ sums to 1 (as a softmax output), so $\|\boldsymbol{a}_i\|_2 \leq 1$ with equality if and only if $\boldsymbol{a}_i$ has exactly one non-zero entry (equal to 1). This holds for the first token (which can only attend to itself in masked attention) so $(\mathbf{A}\Sigma\mathbf{A}^\top)_{11} = 1$, but not in general for later tokens, so $(\mathbf{A}\Sigma\mathbf{A}^\top)_{ii}$ will usually be less than 1, for $i > 1$.

We also note that this approach to controlling the attention matrix by zeroing the query-key dot product at initialisation is compatible with popular positional encodings like Relative (Shaw et al., 2018; Huang et al., 2019; Dai et al., 2019) and RoPE (Su et al., 2021), as discussed in Appendix E. Unless stated otherwise, we use RoPE in our experiments, and provide an ablation in Fig. 10.

### 3.4 ADDRESSING MLP BLOCKS AND SKIP CONNECTIONS

Up until this point we have restricted our focus to attention-only skipless transformers, for the sake of simplicity. However, MLP blocks are an important part of the transformer architecture that we must also address. And we would like for our approaches to be compatible with skip connections too, both for the sake of generality, and because they might combine profitably.

**MLP blocks**   Because MLP blocks operate on the representation vectors independently across locations, their effect on the kernel matrix can be easily computed using the standard limiting kernel formulas for MLPs (Neal, 2012) which are exact in the infinite width limit. In particular, there is a known $f$ that maps the kernel matrix $\Sigma_l$ of an MLP block's input sequence to the kernel matrix $\hat{\Sigma}_l$ of its output sequence. In principle, one could modify SPA to account for this change to the kernel matrix by taking $\mathbf{A}_l = \mathbf{L}_l\hat{\mathbf{L}}_{l-1}^{-1}$, where $\mathbf{L}_l$ and $\hat{\mathbf{L}}_l$ are the Cholesky factors of $\Sigma_l$ and $\hat{\Sigma}_l$ respectively. Unfortunately, there is no guarantee that the resulting $\mathbf{A}_l$ would be elementwise non-negative in general. In our experiments we thus elected to ignore the effect on the kernel matrices of the MLP blocks, approximating $f$ as the identity function, which we found to work well enough in practice. As discussed in Martens et al. (2021), this approximation becomes better as the MLP blocks tend to linear functions (for which $f$ is exactly identity), which will happen as we increase the shortcut weight $\alpha$ relative to the residual weight $\beta$ (see Eq. (1)), or when the MLP's activation functions are close to linear. Notably, the latter condition will tend to be true when using DKS or TAT to transform the activation functions in deep networks. Note, when using DKS, $f$ decreases the size of off-diagonal elements of the kernel matrix, which intuitively will help to combat rank collapse.

**Skip connections**   When using skip connections as in Eq. (1), the output kernel matrix of a block is given by $\Sigma_{\text{block}} = \alpha^2\Sigma_{\text{shortcut}} + \beta^2 \cdot \Sigma_{\text{residual}}$. One of the main ways that kernel matrices degenerate in DNNs is when their diagonal entries either explode or shrink with depth. As shown by Martens et al. (2021), this can be prevented in fully-connected and convolutional DNN architectures by rescaling the output of the activation functions (which happens automatically as part of DKS or TAT), and by replacing weighted sums with "normalised sums", which yields *normalised skip connections* defined by the condition $\alpha^2 + \beta^2 = 1$. When using normalised skip connections with U-SPA, both $\Sigma_{\text{shortcut}}$ and $\Sigma_{\text{residual}}$ will have the form $\Sigma(\rho) = (1-\rho)\mathbf{I}_T + \rho\mathbf{11}^\top$ for some $\rho$, and thus so will $\Sigma_{\text{block}}$. Moreover, we will have $\rho_{\text{block}} = \alpha^2\rho_{\text{shortcut}} + \beta^2 \cdot \rho_{\text{residual}}$, which is less than $\rho_{\text{residual}}$. This means we can easily adjust U-SPA to be compatible with normalised skip connections by replacing $\mathbf{L}_l$ in the formula $\mathbf{A}_l = \mathbf{L}_l\mathbf{L}_{l-1}^{-1}$ with the Cholesky factor of $\Sigma((\rho_l - \alpha^2\rho_{l-1})/\beta^2)$. In the case of E-SPA, we note that $\Sigma_{\text{block}}$ won't be of the form $\big(\Sigma(\gamma)\big)_{i,j} = \exp(-\gamma|i-j|)$ for some $\gamma$, even when both $\Sigma_{\text{shortcut}}$ and $\Sigma_{\text{residual}}$ are. To work around this, we use an approximation described in Appendix F, which seeks to make the combined effect on the kernel matrix of the normalised skip and attention block approximately equal to the effect of a skipless attention block.

## 4 EXPERIMENTS

We now assess the capabilities of our proposed methods in training deep skipless and/or normaliser-free transformers. Our main experiment setting uses a transformer with 36 transformer blocks, which is deep enough that the effects of poor signal propagation and rank collapse render skipless training without modifications impossible. We begin our investigation on WikiText-103 (Merity et al., 2017) focusing primarily on training performance of our various methods, before moving to the larger C4 dataset (Raffel et al., 2019), where overfitting is not an issue. We use Adam optimiser (Kingma & Ba, 2014), as is standard for transformers, and train without dropout (Srivastava et al., 2014). Additional results and experimental details are provided in Appendices G and H respectively.

**WikiText-103 baselines**   To start with, we verify that a standard deep transformer without skip connections is untrainable even with normalisation layers (LN) and transformed activations, and that our approaches remedy this. Fig. 2 compares vanilla transformers using our proposed SPA

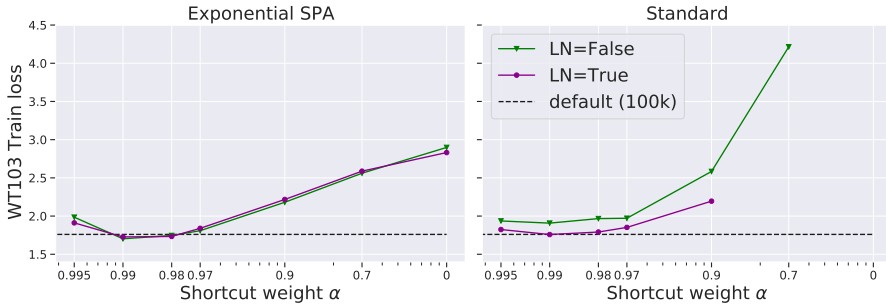

**Figure 3:** Transformers with normalised skip connections, trained for 100K steps. E-SPA (left) without normalisation matches the training speed of a standard transformer. Results denote the mean of 2 seeds.

methods and Value-Skipinit to standard transformers both with and without skips, on a 36 block transformer. We clearly see that removing skip connections from a standard transformer makes it untrainable, with training loss plateauing around 7.5. This holds true too even when we use DKS to transform the GeLU MLPs, highlighting that the issue lies with the attention layers, which suffer from rank collapse as shown in Fig. 1.

On the other hand, all three of our approaches train even for vanilla deep transformers, with our E-SPA method outperforming U-SPA and Value-Skipinit. However, the default transformer with skips and LN still retains a training speed advantage compared to our skipless methods, mirroring the situation for CNNs without powerful second-order optimisers (Martens et al., 2021; Zhang et al., 2022).

In Table 1, we assess the effect of different activation functions in the MLP blocks, as well as the use of LN, in skipless transformers using our proposed methods. We see that at depth 36 we achieve good training performance for a range of activations: DKS-transformed GeLU, TAT-transformed Leaky ReLU, as well untransformed GeLU (Hendrycks & Gimpel, 2016), but not untransformed Sigmoid. We also see that layer normalisation is relatively unimportant for training speed, and can even be harmful with transformed activations when using SPA, which already has an inbuilt mechanism to control activation norms (as discussed at the end of Section 3.2).

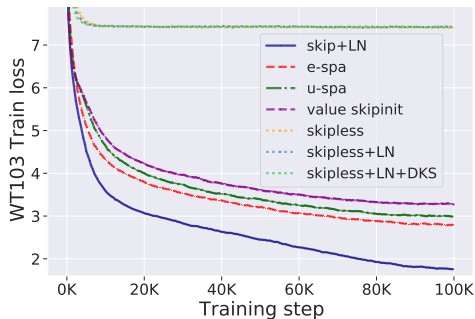

**Figure 2:** Comparison of skipless transformers. Vanilla transformers are not trainable without our modifications. All curves average over 3 seeds.

**Normalised skip connections** In Fig. 3, we see that one way to match the training loss of the default transformer, *without* more iterations, is by using normalised skip connections. While this is perhaps unsurprising, we observe that our E-SPA method (left) matches standard training both with and without normalisation, whereas a standard Transformer with normalised skip connections (right) requires normalisation in order to match the training speed of the default Pre-LN.

**C4 baseline** So far, we tested our proposed methods on WikiText-103 (Merity et al., 2017), on which we observed overfitting without the use of extra regularisation. Therefore, we further compare our methods to standard trans-

**Table 1:** WT103 train loss of skipless transformers for different activations, with or without LN. Mean and standard deviation are computed across 3 random seeds.

| Attention | Activation | LN | w/o LN |
|---|---|---|---|
| E-SPA | DKS-GeLU | $2.99_{\pm.01}$ | $\mathbf{2.79_{\pm.02}}$ |
| | TAT-LReLU | $3.00_{\pm.01}$ | $2.89_{\pm.02}$ |
| | GeLU | $\mathbf{2.86_{\pm.04}}$ | $2.93_{\pm.02}$ |
| | Sigmoid | $7.42_{\pm.01}$ | $7.43_{\pm.01}$ |
| U-SPA | DKS-GeLU | $3.17_{\pm.03}$ | $2.99_{\pm.01}$ |
| | TAT-LReLU | $3.24_{\pm.02}$ | $3.04_{\pm.01}$ |
| | GeLU | $3.10_{\pm.01}$ | $3.25_{\pm.8}$ |
| V-SkipInit | DKS-GeLU | $3.20_{\pm.02}$ | $3.26_{\pm.02}$ |
| | TAT-LReLU | $3.23_{\pm.02}$ | $3.32_{\pm.01}$ |
| | GeLU | $3.18_{\pm.01}$ | $3.17_{\pm.02}$ |

formers on a larger dataset, C4 (Raffel et al., 2019), where overfitting isn't an issue. Importantly, we see similar trends across validation[8] (Fig. 4a), training (Fig. 13) and downstream task (Table 6) performance, and so the benefits of our methods do extend beyond training. Due to the memory

---

[8] Argued by Nakkiran et al. (2021), the validation curves measure the "training speed" of the online setting.

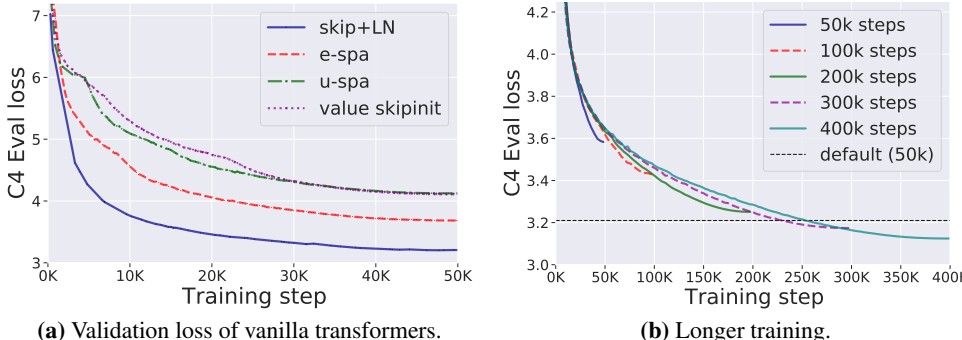

(a) Validation loss of vanilla transformers.    (b) Longer training.

**Figure 4:** Results on C4. (**a**): E-SPA again outperforms our other approaches. (**b**): Vanilla E-SPA matches default Pre-LN after 5x more iterations.

overhead of longer sequences, we use a 32-block transformer. One can notice that E-SPA performs the best among skipless transformers on all settings: training, validation and downstream tasks.

Moreover, in Table 2 we find that E-SPA with normalised skips and LN outperforms the default Pre-LN transformer, achieving 24.0 (vs 24.7) validation perplexity on C4 after 50K steps. Even without LN, E-SPA *exactly matches* the Pre-LN transformer in training speed, and outperforms a range of baselines designed to remove LN, including Stable ResNet (Hayou et al., 2021; Noci et al., 2022) and SkipInit (De & Smith, 2020), both with & without LN.

**Table 2:** Validation perplexity for different skips on C4 with and without LN after 50K iterations.

| Attention | Skip | LN | w/o LN |
|---|---|---|---|
| Default | Pre-LN | 24.7 | – |
| | Normalised | 24.5 | 25.1 |
| | Stable ($\alpha$=1) | 24.4 | 25.3 |
| | SkipInit | 25.9 | 35.3 |
| U-SPA | Normalised | 27.6 | 28.0 |
| E-SPA | Normalised | **24.0** | **24.7** |

**Longer training matches default transformers** In Figs. 2 and 4a, we observed that while our methods are able to train deep skipless transformers (a result which is unprecedented in the literature), there is a significant gap in training speed compared to a standard Pre-LN transformer (with skips). Martens et al. (2021) and Zhang et al. (2022) observed a similar training speed gap for skipless CNNs, and showed that such a gap can be closed by using more sophisticated second order optimisers like K-FAC (Martens & Grosse, 2015) or Shampoo (Gupta et al., 2018; Anil et al., 2020). As second order optimisers for transformers are not well established, we instead demonstrate that the training loss gap can be closed by simply training for longer in Fig. 4b. We observe that our E-SPA method matches the training loss of a standard pre-LN transformer on C4 if one trains for around 5 times longer with Adam, in line with the findings from the convolutional case (Zhang et al., 2022). An equivalent plot for WikiText-103 is provided in Fig. 14.

**Depth Scaling** Finally, as unmodified networks suffer from worse signal propagation properties at larger depths, it is natural to ask how our modified vanilla transformers perform as depth increases. In Table 3, we compare the training performance of different depths (36, 72, and 108) at a range of training step budgets on WikiText-103. We find that whilst the shallower depth 36 network trains fastest initially over 100K steps, it is matched at 400K steps and then surpassed at 1000K steps by the larger capacity depth 72 network. Moreover, our depth 108 vanilla transformer is able to close the gap in performance to its depth 36 counterpart with longer training, going from 0.3 at 100K steps to just 0.01 at 1000K steps.

**Table 3:** WT-103 training loss across depths. Deeper vanilla E-SPA transformers improve with more training.

| Depth | Training steps | | |
|---|---|---|---|
| | 100K | 400K | 1000K |
| 36 | 3.12 | 2.66 | 2.20 |
| 72 | 3.25 | 2.67 | 2.11 |
| 108 | 3.42 | 2.78 | 2.21 |

## 5   CONCLUSION

We have shown for the first time that it is possible to successfully train deep transformers without skip connections or normalisation layers. To do so, we have proposed 3 approaches: E-SPA, U-SPA and Value-Skipinit, each of which control the attention matrices of a transformer to enable faithful signal propagation even at large depths. Our best approach, E-SPA enables deep vanilla transformers to match their standard counterparts with around 5 times more iterations, and also deep transformers without normalisation to match the training speed of standard ones. We hope that our work may potentially pave the way to new and improved architectures, and more research into improving the capabilities of deep learning in practice using insights from theory.

## REPRODUCIBILITY STATEMENT

Pseudocode for our main approach, E-SPA, can be found in Alg. 1, using the notation and setup provided in Section 2. All experimental details can be found in Appendix H, including general and experiment-specific implementation details.

## ACKNOWLEDGEMENTS

We thank Christos Kaplanis for helpful discussions during initial stages of this project, as well as the anonymous reviewers for their feedback. BH is supported by the EPSRC and MRC through the OxWaSP CDT programme (EP/L016710/1).

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

## A  COMPATIBILITY WITH NON-CAUSAL ATTENTION

In Section 3, we focus on causal masked self-attention for two reasons. First, next-token prediction using causal masked self-attention is arguably the most popular setting for self-attention. And second, it is a more challenging setting to work with in terms of controlling signal propagation, due to the additional constraint that attention matrices must be lower triangular. In this section we describe how our methods can be made compatible with non-causal masked attention, where the attention matrices are no longer required to be lower triangular.

To start with, Value-SkipInit does not modify the softmax-attention computation and hence is already compatible with any form of attention. For our SPA methods, it is straightforward to extend to non-causal attention by changing $\mathbf{L}_l$ in Eqs. (8) and (9) from being the Cholesky decomposition of $\Sigma_l$ to being the (symmetric) matrix square root of $\Sigma_l$. In this case, for U-SPA it is possible to analytically calculate that $\mathbf{A}_l$ in Eq. (8) will be element-wise non-negative if $\rho_l \geq \rho_{l-1}$ (exactly like the Cholesky case in Theorem 1). This is easy to see because the matrix square-root, inverses and products of uniform kernel matrices are all still uniform of the form $\Sigma(\rho) = (1 - \rho)\mathbf{I}_T + \rho\mathbf{1}\mathbf{1}^\top$ (up to positive rescaling), so that $\mathbf{A}_l$ will be too, and one simply needs to track $\rho$ and verify that it is positive. For E-SPA, we have verified empirically that the resulting $\mathbf{A}_l = \mathbf{L}_l\mathbf{L}_{l-1}^{-1}$ will be non-negative if $\gamma_l \leq \gamma_{l-1}$, just like the Cholesky case in Theorem 2.

## B  MODIFICATIONS TO SPA METHODS FOR REPEATED TOKENS

For simplicity, we assumed that our input sequences had no repeated tokens when presenting SPA in Section 3. This meant that we could take the input kernel matrix $\Sigma_0$ to be the identity, with zero off-diagonals, which was convenient for our construction of SPA. The effect of repeated tokens, e.g. if the word 'cat' occurs multiple times in the same sentence, is that our input kernel matrices $\Sigma_0$ will have non-zero off-diagonal entries, corresponding to entries where a token is repeated. This will impact the kernel matrices $\Sigma_l$ at deeper layers with SPA, particularly the diagonal values of $\Sigma_l$, which we would like to able to control. In this section we discuss how we can modify our SPA approaches to account for the fact that we will often be working with sequences where a fraction of the tokens are repeated.

We stress that the general principle that all our methods (Value-SkipInit and SPA methods) follow is independent of the input kernel matrix (i.e. independent of duplicate input tokens): we seek to prevent the product of attention matrices from deviating away from the identity matrix and degenerating to a rank-1 matrix. From Eq. (6), we see that if the product of attention-matrices is rank-1 then regardless of the input kernel we will have a rank-1 output kernel i.e. rank collapse (Dong et al., 2021). On the other hand, if we control the deviation of the attention matrix product from the identity then no matter the input kernel, the output kernel will bear some similarity to the input kernel. So as long as the input kernel is non-degenerate and has full rank (regardless of duplicate tokens) so too will be the output kernel.

Recall that with attention matrices $(\mathbf{A}_l)_l$, an input kernel matrix across $T$ locations $\Sigma_0 \in \mathbb{R}^{T \times T}$ gets mapped, at depth $l$, to

$$\Sigma_l = \Pi_l \cdot \Sigma_0 \cdot \Pi_l^\top \tag{11}$$

where $\Pi_l = \mathbf{A}_l\mathbf{A}_{l-1}\ldots\mathbf{A}_1$ is the product of attention matrices.

In SPA, we parameterise $\mathbf{A}_l = \mathbf{L}_l\mathbf{L}_{l-1}^{-1}$ for $(\mathbf{L}_l)_l$ corresponding to the Cholesky factors of some family of kernel matrices $(\Sigma_l)_l$, with either uniform or exponentially decaying off diagonals:

1. *U-SPA*: $\Sigma_l(\rho_l) = (1 - \rho_l)\mathbf{I}_T + \rho_l\mathbf{1}\mathbf{1}^\top$ for $0 \leq \rho \leq 1$
2. *E-SPA*: $\left(\Sigma_l(\gamma_l)\right)_{i,j} = \exp(-\gamma_l|i - j|)$ for $\gamma \geq 0$.

It is important that $\mathbf{A}_l = \mathbf{L}_l\mathbf{L}_{l-1}^{-1}$ is constructed from two Cholesky matrices belonging to the *same* family, because Theorems 1 and 2 show in that case we will satisfy our non-negativity constraint on $\mathbf{A}_l$ (which is computed through a softmax operation), and otherwise there is no prior reason to suppose that non-negativity will be satisfied. Therefore, in an ideal world, $\Sigma_0$ would be an element of our family of kernel matrices, so that we can set $\mathbf{L}_0$ to be the Cholesky factor of $\Sigma_0$, satisfying $\mathbf{L}_0^{-1}\Sigma_0\mathbf{L}_0^{-1^\top} = \mathbf{I}_T$.

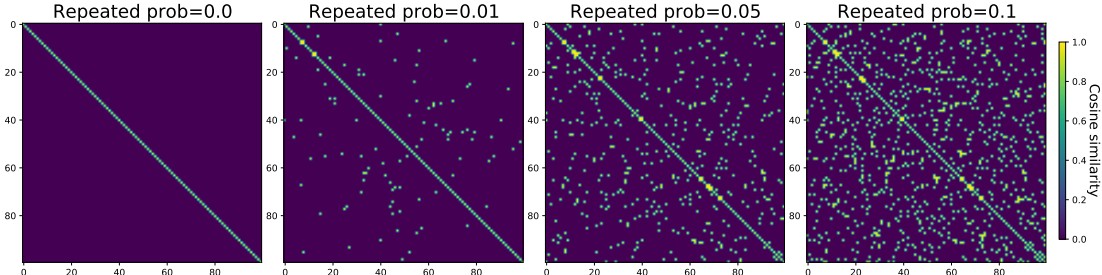

**Figure 5:** Input token kernel matrices for different proportion of repeated tokens.

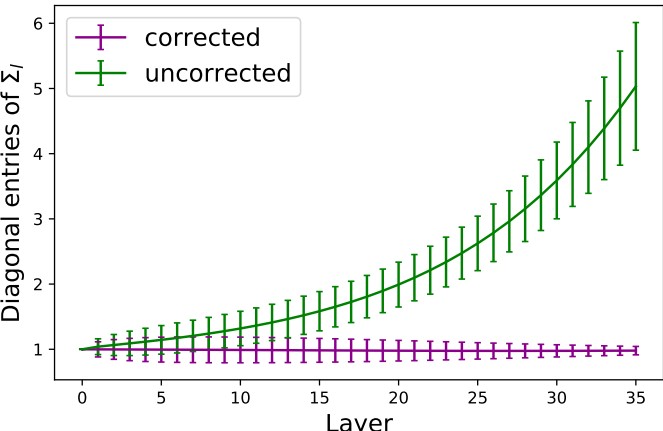

**Figure 6:** Diagonal entries of $\Sigma_l$ for a *single sequence* of length $T = 100$ across blocks for E-SPA in the presence of $r = 0.05$ shared tokens, with and without modifications. We see that without our modifications and simply assuming $\Sigma_0 = \mathbf{I}$ by default (green) the average diagonal diverges at deeper blocks, when $\gamma_l$ is smaller and the off-diagonals of $\Sigma_l$ are larger. On the other hand, our modifications (purple) keep the average diagonal around 1 with controlled standard deviations. Here, the last kernel matrix (at depth 36) has decay rate $\gamma_L = 0.02$ and the standard deviation is computed over the 100 locations.

Typically, $\Sigma_0 = \mathbf{X}\mathbf{X}^\top/d_0$ for input token embeddings $\mathbf{X} \in \mathbb{R}^{T \times d_0}$, which are independently initialised for different tokens identites. Thus, in the wide limit $d \to \infty$, (up to rescaling) $\Sigma_0$ will have diagonals equal to 1, and off-diagonals equal to 1 if there is a repeated token and 0 else. We plot examples of such kernel matrices in Fig. 5 for different *fractions of repeated tokens* $r$, under the assumption that repeated tokens occur independently of location.

Clearly, when $r = 0$, we have $\Sigma_0 = \mathbf{I}$ is a member of both the uniform and exponential families of kernel matrices, corresponding to uniform off-diagonals of $\rho_0 = 0$ or exponentially decaying off-diagonals with rate $\gamma_0 = \infty$. In this case we can set $\mathbf{L}_0 = \mathbf{I}$ too.

On the other hand, for $r > 0$, it is in general difficult to say much more about an individual sequence's $\Sigma_0$, given that different sequences will have repeated tokens in different locations. Moreover, the naive approach of ignoring the repeated tokens and treating $\Sigma_0 = \mathbf{I}$ leads to increasing diagonal values of $\Sigma_l$ (i.e. activation norms) at large depth without corrections, as shown in Fig. 6.[9] This imbalance between blocks could be problematic for training dynamics, and also is incompatible with frameworks like DKS and TAT which suppose that diagonal values of $\Sigma_l$ are constant across blocks and locations, usually set to 1.

To circumvent this, we will derive our modifications by considering the *average input kernel matrix* $\bar{\Sigma}_0$ (averaged over different sequences), under the assumption that repeated tokens occur independent of location:

$$\bar{\Sigma}_0 = (1 - r)\mathbf{I}_T + r\mathbf{1}\mathbf{1}^T \tag{12}$$

---

[9]We find that $r \approx 0.008$ for sentencepiece tokenisations like we use in WikiText-103 and C4, but $r \approx 0.05$ for character-level prediction like in EnWiki-8.

By linearity of Eq. (11) in $\Sigma_0$ (and because we are controlling $\mathbf{A}_l$ to be input-independent at initialisation in Section 3), it also follows that the *average depth l kernel matrix* $\bar{\Sigma}_l$ satisfies:

$$\bar{\Sigma}_l(\mathbf{X}, \mathbf{X}) = \Pi_l \cdot \bar{\Sigma}_0 \cdot \Pi_l^\top \tag{13}$$

so we can modify our SPA approaches to control the average kernel matrix $\bar{\Sigma}_l$ instead.

For U-SPA, the situation is more straightforward as $\bar{\Sigma}_0$ is a uniform kernel matrix with $\rho = r$, so it suffices to simply let $\rho_0 = r$ instead of $\rho_0 = 0$.

For E-SPA, we are unable to view $\bar{\Sigma}_0$ as having exponentially decaying off-diagonals, and so we keep $\gamma_0 = \infty$ which translates to $\mathbf{L}_0 = \mathbf{I}$ and $\Pi_l = \mathbf{L}_l$. Instead, to help us understand the effect of repeated tokens (to motivate our modifications), we first expand on Eq. (13) to simplify a little:

$$\begin{aligned}
\Pi_l \cdot \bar{\Sigma}_0 \cdot \Pi_l^\top &= \mathbf{L}_l \cdot \bar{\Sigma}_0 \cdot \mathbf{L}_l^\top \\
&= (1-r)\mathbf{L}_l\mathbf{L}_l^\top + r\mathbf{L}_l\mathbf{1}\mathbf{1}^\top\mathbf{L}_l^\top \\
&= \Sigma_l(\gamma_l) + r\mathbf{L}_l(\mathbf{1}\mathbf{1}^\top - \mathbf{I})\mathbf{L}_l^\top \\
&= \Sigma_l(\gamma_l) + r\mathbf{L}_l\mathbf{O}_T\mathbf{L}_l^\top
\end{aligned} \tag{14}$$

where $\mathbf{O}_T = \mathbf{1}\mathbf{1}^\top - \mathbf{I}_T \in \mathbb{R}^{T \times T}$ is 0 on the diagonal and 1 off the diagonal.

Note that all terms in Eq. (14) are easily computable (given that $\left(\Sigma_l(\gamma_l)\right)_{i,j} = \exp(-\gamma_l|i-j|)$) and $\mathbf{L}_l$ has an analytic form provided in Lemma 1). So we can use Eq. (14) to compute the expected diagonal $(\bar{\Sigma}_l)_{i,i}$ for each location $i$ (where the expectation is taken over different sequences), which we denote by a diagonal matrix $\bar{D}_l$:

$$\bar{D}_l = \mathrm{Diag}(\mathbf{L}_l \cdot \bar{\Sigma}_0 \cdot \mathbf{L}_l^\top) \in \mathbb{R}^{T \times T}$$

Thus, we propose to replace $\mathbf{A}_l = \mathbf{L}_l\mathbf{L}_{l-1}^{-1}$ with $\mathbf{A}_l = \bar{D}_l^{-\frac{1}{2}}\mathbf{L}_l\mathbf{L}_{l-1}^{-1}\bar{D}_{l-1}^{\frac{1}{2}}$, setting $\bar{D}_0 = \mathbf{I}$ by default. This means that our product $\Pi_l = \prod_{i=1}^l \mathbf{A}_i = \bar{D}_l^{-\frac{1}{2}}\mathbf{L}_l$, and Eq. (13) is updated to:

$$\bar{\Sigma}_l(\mathbf{X}, \mathbf{X}) = \bar{D}_l^{-\frac{1}{2}}\mathbf{L}_l \cdot \bar{\Sigma}_0 \cdot \mathbf{L}_l^\top \bar{D}_l^{-\frac{1}{2}} \tag{15}$$

which has diagonals controlled to 1.

Though our modifications for repeated tokens only consider averages across different sequences, we find that for individual sequences they still lead to well behaved diagonal values of $\Sigma_l$, as shown in Fig. 6. Moreover, we find that their effect on off-diagonals is still favourable for individual sequences, as shown in Fig. 1.

## C   Setting $(\rho_l)_l$ and $(\gamma_l)_l$

In this section we describe how we set the uniform off-diagonals $(\rho_l)_l$ and the exponential decay rates $(\gamma_l)_l$ in SPA, at different depths.

Recall that Theorems 1 and 2 show that we are free to choose $(\rho_l)_{l=0}^L$ and $(\gamma_l)_{l=0}^L$ such that $(\rho_l)_{l=0}^L$ increase with depth and $(\gamma_l)_{l=0}^L$ decrease with depth. Moreover, $\rho_0 = 0$ (or the shared token fraction $r$, as per Appendix B) and $\gamma_0 = \infty$ at the input layer, whilst we have found $\rho_L = 0.8$ and $\gamma_L = 0.005$ to be good default values for the last block.

For U-SPA, in terms of setting how $(\rho_l)_l$ vary with depth, we tried different polynomial rates of increase with depth from $\rho_0$ to $\rho_L$, but did not observe a noticeable difference in performance across different rates so chose to increase from $\rho_0$ to $\rho_L$ linearly in depth.

For E-SPA, we choose $(\gamma_l)_l$ so that the diagonal elements of the attention matrices $\mathbf{A}_l$ are constant across blocks, akin to using a constant shortcut weight $\alpha$ over different blocks. From Theorem 3, we have that the diagonal entries of $\mathbf{A}_l$ satisfy:

$$(\mathbf{A}_l)_{i,i} = \frac{a(\gamma_l)}{a(\gamma_{l-1})}, \qquad \forall i > 1$$

where $a(\gamma) = \sqrt{1 - \exp(-2\gamma)}$ with inverse $\gamma(a) = -\frac{1}{2}\log(1 - a^2)$.

This means that for a set of positive decreasing $(\gamma_l)_l$, there exist a corresponding set of decreasing $(a_l)_l$ with values between 0 and 1.

Because $\gamma_0 = \infty$, we have $a_0 = 1$, and likewise for a given $\gamma_L$ we can compute $a_L$.

Thus to have constant diagonal values of $\mathbf{A}_l$ over different blocks $l$, we choose to set

$$a_l = a_L^{l/L}$$

for $l \leq L$, and thus

$$\gamma_l = \gamma(a_l) = -\frac{1}{2}\log(1 - a_L^{2l/L})$$

We found this scheme to work well empirically, and note the similarity to other works which have discussed the choice of how to scale branches in residual architectures (Hayou et al., 2021). We leave further study of choosing how to set $(\rho_l)_l$ and $(\gamma_l)_l$ to future work.

## D  E-SPA ALGORITHM

In Alg. 1 we present pseudocode to construct a trainable E-SPA masked attention layer.

---

**Algorithm 1:** Modified E-SPA masked attention layer.

---

**Input:** Input sequence representation $\mathbf{X} \in \mathbb{R}^{T \times d}$.
**Output:** Updated sequence representation $\text{Attn}(\mathbf{X}) \in \mathbb{R}^{T \times d}$.
**Hyperparameters:** Input $\gamma_{\text{in}}$ & output $\gamma_{\text{out}}$ exponential decay rates, $\gamma_{\text{in}} \geq \gamma_{\text{out}}$. Number of
$\qquad$ heads $h$ with head dimension $d_h = d/h$. Causal mask $\mathbf{M} \in \mathbb{R}^{T \times T}$ s.t.
$\qquad$ $\mathbf{M}_{i,j} = \mathbb{1}\{i \geq j\}$. Large constant $\Gamma$, e.g. $10^{30}$, to enforce causal mask.
**Trainable parameters:** $\mathbf{W}_n^Q, \mathbf{W}_n^K, \mathbf{W}_n^V \in \mathbb{R}^{d \times d_h}$, for $1 \leq n \leq h$. $\mathbf{W}^O \in \mathbb{R}^{d \times d}$.

1 Compute $\mathbf{L}_{\text{in}}$ and $\mathbf{L}_{\text{out}}$ to be the Cholesky factors of $(\Sigma_i^{\text{in}})_{i,j} = \exp(-\gamma_{\text{in}}|i - j|)$ and
$\quad$ $(\Sigma_i^{\text{out}})_{i,j} = \exp(-\gamma_{\text{out}}|i - j|)$ respectively.
2 Set $\mathbf{A} = \mathbf{L}_{\text{out}}\mathbf{L}_{\text{in}}^{-1}$ (using analytic form in Theorem 3) & decompose $\mathbf{A} = \mathbf{D}\mathbf{P}$ where $\mathbf{D}$
$\quad$ diagonal & $\mathbf{P}$ has row sums 1. Denote $\mathbf{B} = \log(\mathbf{P})$.
3 **for** head $n \in \{1, \dots, h\}$ **do**
4 $\quad$ Initialise $\mathbf{W}_n^K, \mathbf{W}_n^V \overset{i.i.d.}{\sim} \mathcal{N}(0, \frac{1}{d})$ (alternatively orthogonally). Initialise $\mathbf{W}_n^Q = \mathbf{0}$.
5 $\quad$ Set $\mathbf{Q}_n(\mathbf{X}) = \mathbf{X}\mathbf{W}_n^Q$, $\mathbf{K}_n(\mathbf{X}) = \mathbf{X}\mathbf{W}_n^K$ and $\mathbf{V}_n(\mathbf{X}) = \mathbf{X}\mathbf{W}_n^V$.
6 $\quad$ Compute $\mathbf{P}_n(\mathbf{X}) = \text{softmax}\left(\mathbf{M} \circ \left(\frac{1}{\sqrt{d_h}}\mathbf{Q}_n(\mathbf{X})\mathbf{K}_n(\mathbf{X})^\top + \mathbf{B}\right) - \Gamma(1 - \mathbf{M})\right)$
7 $\quad$ Set $\text{Attn}_n(\mathbf{X}) = \mathbf{D}\mathbf{P}_n(\mathbf{X}) \cdot \mathbf{V}_n(\mathbf{X})$
8 Initialise $\mathbf{W}^O \overset{i.i.d.}{\sim} \mathcal{N}(0, \frac{1}{d})$ (alternatively as an orthogonal matrix)
9 **return** $\text{Concat}\left(\text{Attn}_1(\mathbf{X}), \dots, \text{Attn}_h(\mathbf{X})\right)\mathbf{W}^O$

---

## E  COMPATIBILITY OF SPA WITH EXISTING POSITIONAL ENCODINGS

In Section 3.3, we showed how to control the attention matrix at initialisation, using bias matrices and location-dependent rescaling, as well as making the query-key dot product, $\frac{1}{\sqrt{d^k}}\mathbf{Q}(\mathbf{X})\mathbf{K}(\mathbf{X})^\top = \frac{1}{\sqrt{d^k}}\mathbf{X}\mathbf{W}^Q(\mathbf{X}\mathbf{W}^K)^\top$, zero at initialisation. This scheme is used in our SPA methods. We detailed several ways to achieve this, and in practice we chose to initialise $\mathbf{W}^Q$ to zero, and initialise $\mathbf{W}^K$ as usual (Gaussian fan-in or orthogonal).

In this section we show how zero initialising the query-key dot product is also possible when using two standard positional encodings: RoPE (Su et al., 2021) and Relative (Shaw et al., 2018; Huang et al., 2019; Dai et al., 2019). This means that we can use our methods in Section 3.3 in combination with these positional encoders.

Let us denote the unscaled query-key dot product as $\mathbf{S} = \mathbf{X}\mathbf{W}^Q(\mathbf{X}\mathbf{W}^K)^\top$.

For RoPE, the $(i, j)$ query-key dot product, $\mathbf{S}_{i,j}$, is modified from

$$(\mathbf{W}^Q \mathbf{X}_i)^\top (\mathbf{W}^K \mathbf{X}_j)$$

to

$$(\mathbf{R}_i \mathbf{W}^Q \mathbf{X}_i)^\top (\mathbf{R}_j \mathbf{W}^K \mathbf{X}_j), \tag{16}$$

where $\mathbf{R}_{i,j}$ are some location-dependent rotation matrices defined in Su et al. (2021), and $\mathbf{X}_i$ denotes row $i$ of the incoming representation $\mathbf{X} \in \mathbb{R}^{T \times d}$ to the attention block. Clearly, Eq. (16) is 0 for all $i, j$ if $\mathbf{W}^Q$ is zero.

For Relative positional encoding, we take the scheme from (Dai et al., 2019). In that case, the $(i, j)$ query-key dot product, $\mathbf{S}_{i,j}$, is modified from

$$(\mathbf{W}^Q \mathbf{X}_i)^\top (\mathbf{W}^K \mathbf{X}_j)$$

to

$$(\mathbf{W}^Q \mathbf{X}_i)^\top (\mathbf{W}^K \mathbf{X}_j) + (\mathbf{W}^Q \mathbf{X}_i)^\top (\mathbf{W}^{K,R} \mathbf{R}_{i-j}) + \boldsymbol{u}^\top (\mathbf{W}^K \mathbf{X}_j) + \boldsymbol{v}^\top (\mathbf{W}^{K,R} \mathbf{R}_{i-j}) \tag{17}$$

where $\mathbf{R}_{i-j} \in \mathbb{R}^d$ is fixed, and $\mathbf{W}^{K,R} \in \mathbb{R}^{d_k \times d}$, $\boldsymbol{u}, \boldsymbol{v} \in \mathbb{R}^{d_k}$ are trainable. Thus, we can achieve zero query-key dot products at initialisation if we initialise $\boldsymbol{u}, \boldsymbol{v} = 0$, in addition to $\mathbf{W}^Q = 0$.

## F    USING NORMALISED SKIP CONNECTIONS WITH E-SPA

In this section, we describe how to combine our E-SPA method with normalised skip connections in our attention blocks:

$$\mathbf{Z} = \alpha \mathbf{X} + \sqrt{1 - \alpha^2} \mathbf{A} \mathbf{X} \mathbf{W}^V \tag{18}$$

where $\alpha = 0$ is the skipless setting that our methods are originally designed for.

The general gist is that we will look at the combined effect, on the kernel matrix after the residual attention block, of the residual branch and the diagonal terms in the attention matrices for preserving signal propagation, and approximate the combination to match the setting where we are without skips, i.e. $\alpha = 0$.

To combine E-SPA with normalised skip connections, we consider how the cosine-similarity between two locations $T_1, T_2 \leq T$ is affected through the normalised skip connection. Suppose we have an input kernel matrix:

$$\frac{1}{d} \mathbf{X} \mathbf{X}^\top = \Sigma$$

where $(\Sigma)_{i,i} = 1, \forall i$ and $d$ is the width. Then after the residual attention block, Eq. (18), we now have:

$$\frac{1}{d} \mathbf{Z} \mathbf{Z}^\top = \alpha^2 \frac{1}{d} \mathbf{X} \mathbf{X}^\top + (1 - \alpha^2) \frac{1}{d} \mathbf{A} \mathbf{X} \mathbf{W}^V \mathbf{W}^{V^\top} \mathbf{X}^\top \mathbf{A}^\top \tag{19}$$

$$+ \alpha \sqrt{1 - \alpha^2} \frac{1}{d} \left[ \mathbf{X} \mathbf{W}^{V^\top} \mathbf{X}^T \mathbf{A}^T + \mathbf{A} \mathbf{X} \mathbf{W}^V \mathbf{X}^\top \right] \tag{20}$$

Either we have $\mathbf{W}^V$ to be an orthogonal matrix sampled uniformly at random from the Haar measure, or $\mathbf{W}^V \overset{i.i.d.}{\sim} \mathcal{N}(0, \frac{1}{d})$. In both cases we have $\frac{1}{d} \mathbf{A} \mathbf{X} \mathbf{W}^V \mathbf{W}^{V^\top} \mathbf{X}^\top \mathbf{A}^\top$ going towards $\frac{1}{d} \mathbf{A} \mathbf{X} \mathbf{X}^\top \mathbf{A}^\top$, and the cross terms Eq. (20) converging to 0 for large $d$.

Thus, we can consider the large $d$ approximation:

$$\frac{1}{d} \mathbf{Z} \mathbf{Z}^\top = \alpha^2 \frac{1}{d} \mathbf{X} \mathbf{X}^\top + (1 - \alpha^2) \frac{1}{d} \mathbf{A} \mathbf{X} \mathbf{X}^\top \mathbf{A}^\top \tag{21}$$

$$= \alpha^2 \Sigma + (1 - \alpha^2) \mathbf{A} \Sigma \mathbf{A}^\top \tag{22}$$

Then, if we look at the inner product between the $T_1$ and $T_2$ locations for locations $T_1 \neq T_2$ such that $T_1, T_2 > 1$, we have:

$$
\begin{aligned}
\frac{1}{d}(\mathbf{Z}\mathbf{Z}^\top)_{T_1,T_2} =& \alpha^2 \Sigma_{T_1,T_2} + (1-\alpha^2)(\mathbf{A}\Sigma\mathbf{A}^\top)_{T_1,T_2} \\
=& \alpha^2 \Sigma_{T_1,T_2} + (1-\alpha^2)\sum_{i,j} \mathbf{A}_{T_1,i}\Sigma_{i,j}\mathbf{A}_{T_2,j} \\
=& \alpha^2 \Sigma_{T_1,T_2} + (1-\alpha^2)\mathbf{A}_{T_1,T_1}\Sigma_{T_1,T_2}\mathbf{A}_{T_2,T_2} + (1-\alpha^2)\sum_{i\neq T_1\cup j\neq T_2} \mathbf{A}_{T_1,i}\Sigma_{i,j}\mathbf{A}_{T_2,j} \\
=& \left[\alpha^2 + (1-\alpha^2)\lambda^2\right]\Sigma_{T_1,T_2} + (1-\alpha^2)\sum_{i\neq T_1\cup j\neq T_2} \mathbf{A}_{T_1,i}\Sigma_{i,j}\mathbf{A}_{T_2,j} \\
=& \left[\alpha^2 + (1-\alpha^2)\lambda^2\right]\Sigma_{T_1,T_2} + (1-\alpha^2)\delta \\
=& \nu\Sigma_{T_1,T_2} + (1-\alpha^2)\delta
\end{aligned}
\tag{23}
$$

where $\lambda = \mathbf{A}_{T_1,T_1} = \mathbf{A}_{T_2,T_2}$ is the constant diagonal of the attention matrices in E-SPA, c.f. Theorem 3, and $\nu = \alpha^2 + (1-\alpha^2)\lambda^2$.

Moreover, we have defined $\delta = \sum_{i\neq T_1\cup j\neq T_2} \mathbf{A}_{T_1,i}\Sigma_{i,j}\mathbf{A}_{T_2,j}$ as it is a term that is not possible to control with only knowledge of $\Sigma_{T_1,T_2}$ and will vary from sequence to sequence, and hence we argue can be discarded from a signal propagation perspective. For example, one could consider a sequence where the kernel matrix $\Sigma$ has all off-diagonals equal to 0 apart from $\Sigma_{T_1,T_2}$, and $T_1$ could be sufficiently distant from $T_2$ such that there is no $i$ for which $\mathbf{A}_{T_1,i}$ and $\mathbf{A}_{T_2,i}$ are large at the same time (which can be seen using the analytic form of $\mathbf{A}$ given in Theorem 3).

Thus, from Eq. (23), we see that after an residual attention block, an input cosine similarity of $\Sigma_{T_1,T_2}$ between locations $T_1, T_2$ is diluted by a factor of

$$
\nu = \alpha^2 + (1-\alpha^2)\lambda^2 < 1,
$$

with shortcut weight $\alpha$ and attention diagonal probability $\lambda$. Our approximation then seeks to preserve this factor $\nu$ when $\alpha > 0$ to match the skipless case.

Now, in the skipless case $\alpha = 0$ described in Section 3, at block $l$ we suppose that the incoming kernel matrix $\Sigma$ has exponentially decaying off-diagonals with rate $\gamma_{l-1}$, and that we construct the attention matrix $\mathbf{A}$ so that the output kernel matrix has exponentially decaying off diagonals with rate $\gamma_l$. From Theorem 3, we see that this gives diagonal entries of $\mathbf{A}$ to be:

$$
\lambda_0 = \frac{a(\gamma_l)}{a(\gamma_{l-1})}
$$

where $a(\gamma) = \sqrt{1 - \exp(-2\gamma)}$.

Thus, to preserve $\nu$ to match the case for $\alpha = 0$, if we have shortcut weight $\alpha > 0$ we need the attention matrix diagonal probability $\lambda_\alpha$ to satisfy:

$$
\lambda_\alpha = \sqrt{\frac{\lambda_0^2 - \alpha^2}{1 - \alpha^2}}.
\tag{24}
$$

In turn, when we have shortcut weight $\alpha$, this means that we need to choose our outgoing decay rate at block $l$, $\gamma_{l,\alpha}$, such that:

$$
a(\gamma_{l,\alpha}) = \lambda_\alpha a(\gamma_{l-1}).
$$

Inverting the definition of $a(\gamma)$, we see we need to set $\gamma_{l,\alpha}$:

$$
\gamma_{l,\alpha} = -\frac{1}{2}\log(1 - (\lambda_\alpha^2 a(\gamma_{l-1})^2))
\tag{25}
$$

$$
= -\frac{1}{2}\log\left(1 - \frac{\lambda_0^2 - \alpha^2}{1 - \alpha^2}(1 - \exp(-2\gamma_{l-1}))\right)
\tag{26}
$$

**Table 4:** Validation perplexity equivalent of Table 1

| Attention | Activation | With LN | Without LN |
|---|---|---|---|
| E-SPA | DKS-GeLU | $21.0_{\pm.1}$ | $20.0_{\pm.3}$ |
| | TAT-LReLU | $21.5_{\pm.1}$ | $19.7_{\pm.1}$ |
| | GeLU | $20.4_{\pm.2}$ | $23.1_{\pm.1}$ |
| U-SPA | DKS-GeLU | $24.1_{\pm.3}$ | $23.2_{\pm.3}$ |
| | TAT-LReLU | $26.0_{\pm.5}$ | $21.8_{\pm.2}$ |
| | GeLU | $24.7_{\pm.2}$ | $29.2_{\pm.7}$ |
| V-SkipInit | DKS-GeLU | $24.7_{\pm.1}$ | $27.5_{\pm.1}$ |
| | TAT-LReLU | $25.7_{\pm.2}$ | $27.4_{\pm.1}$ |
| | GeLU | $25.5_{\pm.3}$ | $26.1_{\pm.7}$ |

To summarise, if we have a sequence $(\gamma_l)_l$ of decreasing exponential decay rates, our proposed approximation when using shortcut weights $\alpha$ at block $l$ is, using the notation of Alg. 1, to set $\gamma_{\text{in}} = \gamma_{l-1}$ as normal, and to set $\gamma_{\text{out}} = \gamma_{l,\alpha}$ from Eq. (25) in order to preserve the signal propagation from the combined residual attention block Eq. (23). We see that this approximation reduces to the standard skipless setting when $\alpha = 0$, as a sanity check.

## G  ADDITIONAL RESULTS

In this section we present additional results and ablations that were not included in Section 4.

**Normalised kernel matrix evolution for attention-only transformers with skips and normalisation**  In Fig. 7 we plot the evolution of normalised kernel matrices for transformers with skips and or RMSNorm normalisation, in addition to those for vanilla transformers in Fig. 1. We see that both skipless with RMSNorm (fourth row) and Post-LN (bottom row) converge to rank collapse at larger depths. The degeneration of skipless with RMSNorm is expected from the results of (Dong et al., 2021), and while the convergence to rank collapse is slower for Post-LN, it is still expected (Hayou et al., 2021; Noci et al., 2022). This is because the residual and shortcut branches are effectively given a constant weighting at all blocks in Post-LN, even as the network's depth increases.

On the other hand, Pre-LN observes sensible signal propagation even at depth 100, as the positioning of the LN in the residual branch effectively downweights the residual branch at *later* blocks (De & Smith, 2020). Likewise, Pre-LN with skip weight $\alpha = 0.98$ also observes faithful signal propagation, because each block is effectively downweighted. This effect means that standard Pre-LN's (fifth row) kernel matrix increases elementwise faster with depth than Pre-LN with normalised skips (sixth row), despite both kernel matrices being qualitatively similar at block 100. Note, all methods besides our SPA methods used ALiBi positional encoder (Press et al., 2022) (detailed in Appendix H.2) and we observe both Pre-LN kernel matrices also have a *recency bias*, like our main method, E-SPA (third row).

**Validation perplexity across different skipless methods**  In Table 1 we compared the training speeds for our skipless methods, and found that E-SPA outperforms both U-SPA and Value-Skipinit. In particular, we found E-SPA with a DKS-transformed GeLU activation without LN to perform best. In Table 4, we present the corresponding results but for validation perplexity. Again, we see that E-SPA is the best performing of our attention modifications, but in this case TAT with Leaky ReLU and no LN matches or outperforms DKS with GeLU.

**Sensitivity to query-key initialisation scale**  Recall in Section 3.3 that for attention layers using SPA we seek to initialise weights such that the query-key dot product, $\frac{1}{\sqrt{d^k}}\mathbf{X}\mathbf{W}^Q(\mathbf{X}\mathbf{W}^K)^\top$, is zero or small at initialisation. In our main experiments, we achieve this by initialising $\mathbf{W}^Q = 0$, and letting $\mathbf{W}^K$ to be initialised as normal. In Fig. 8, we assess the sensitivity of our E-SPA scheme to the scale of non-zero attention dot product, when $\mathbf{W}^K, \mathbf{W}^Q$ are both orthogonally initialised but with scale $\sigma$ (i.e. at each layer, both $\mathbf{W}^K, \mathbf{W}^Q$ are initialised as two independent uniform

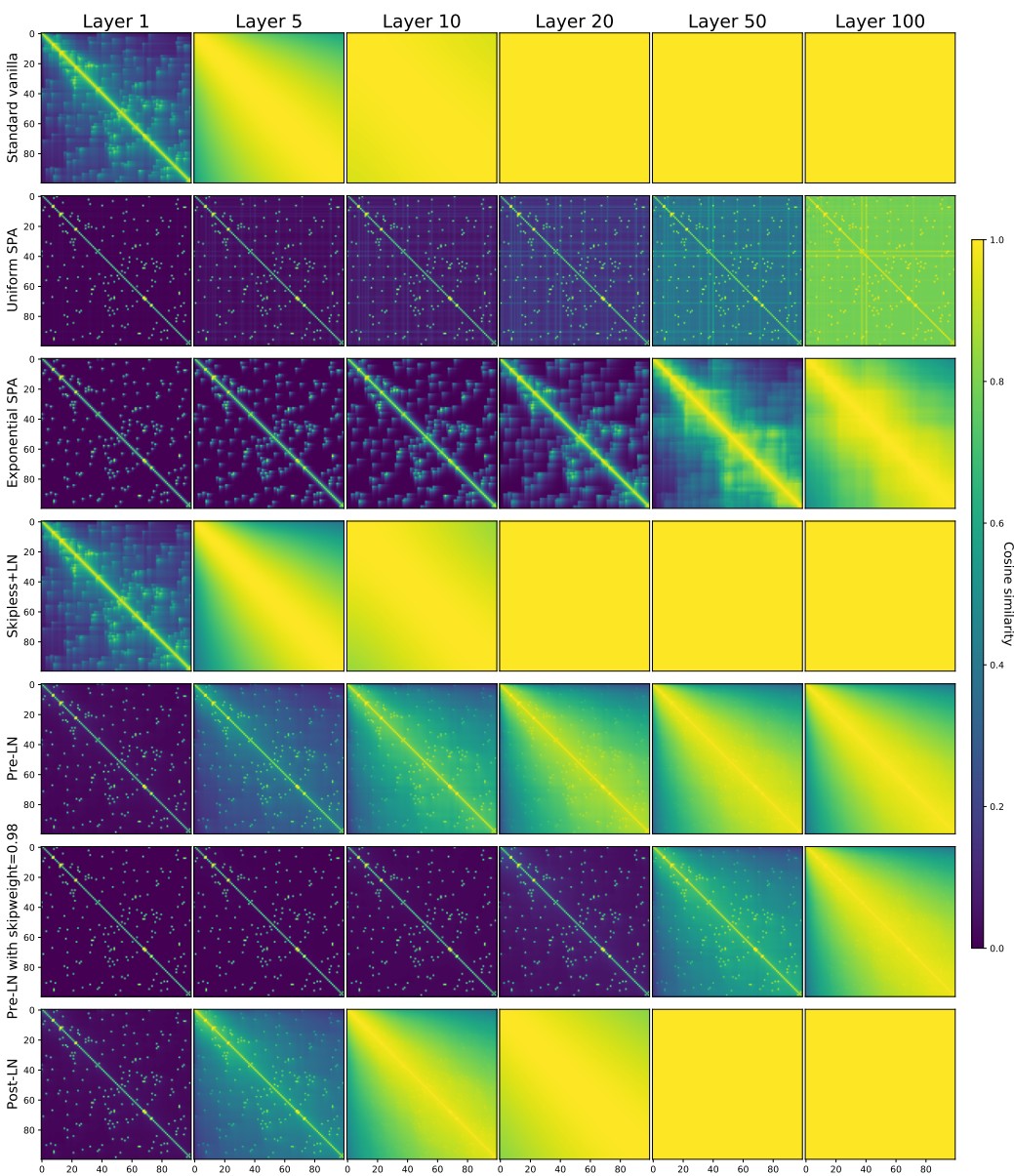

**Figure 7:** Equivalent of Fig. 1 but with four additional configurations for Transformers: 1) Skipless+LN; 2) Pre-LN; 3) Pre-LN with normalised skips $\alpha = 0.98$ and $\beta = \sqrt{1 - \alpha^2}$; and 4) Post-LN. All rows except our SPA methods (i.e all rows except second and third) use pre-softmax bias matrices from the ALiBi (Press et al., 2022) positional encoding to compute attention matrices, assuming zero query-key dot product.

orthgonal matrices multiplied by $\sigma$). We see that for small initialisation scales, there is little effect of varying initial scale, but that training performance degrades at larger scales, when our attention matrix reverse engineering in Section 3.3 (which expects small or zero query-key dot product at initialisation) is less precise.

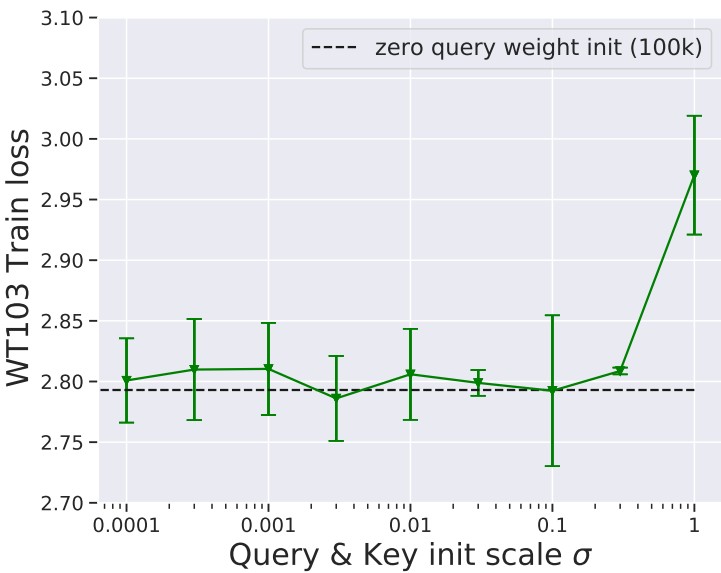

**Figure 8:** The sensitivity of training performance to the initialisation scale $\sigma$ of $\mathbf{W}^Q$ and $\mathbf{W}^K$ using vanilla E-SPA. Mean and error bars are over 3 seeds.

**Ablation over orthogonal initialisation**   Recall in Section 3 that our kernel matrix evolution for attention-only transformers is exact at finite widths using orthogonally initialised weight matrices, and will be approximate using standard (Gaussian) fan-in initialised In Fig. 9, we ablate over using orthogonally initialised weight matrices, compared to Gaussian fan-in initialisation. Across activations, we see that E-SPA with orthogonal initialisation slightly outperforms Gaussian fan-in initialisation (by around 0.15 train loss).

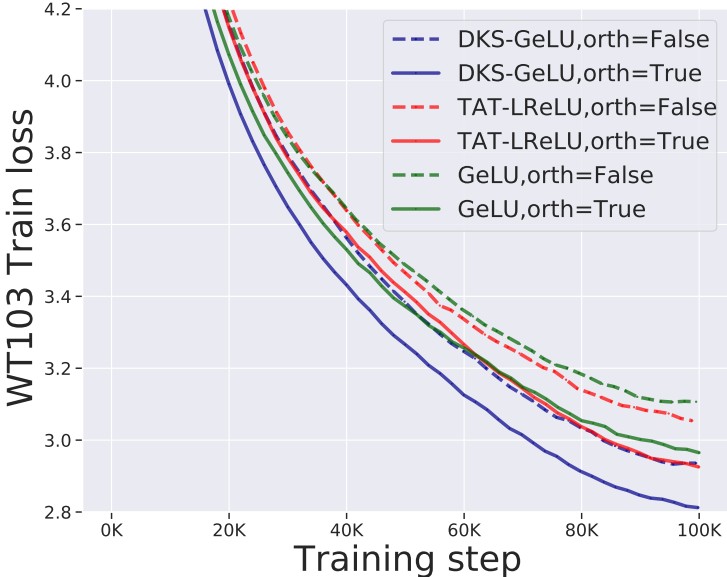

**Figure 9:** Ablation over using orthogonal parameter initialisations, compared to standard Gaussian fan-in, for our main method E-SPA on vanilla transformers over a range of activation functions. We see that orthogonal intialisation leads to a small improvement in training speed. Curves denote mean over 3 seeds

**Ablation over positional encodings**   As noted in Appendix E, our methods are compatible with several standard positional encodings, and by default all our experiments use the popular RoPE (Su et al., 2021) positional encoding. In Fig. 10 and Table 5, we assess the effect of removing positional encodings, on training and validation performance respectively, in our skipless methods. We see that all methods are improved when combined with RoPE, however the improvement is most mild in E-SPA, which as discussed has an in-built recency bias akin to a positional encoder. Moreover, E-SPA without additional positional encoding still outperforms all other approaches, including U-SPA (which on its own has no notion of position) with RoPE.

**Table 5:** Validation perplexity equivalent of Fig. 10

| Attention | PosEnc | With LN | Without LN |
|---|---|---|---|
| E-SPA | None | $22.4_{\pm.1}$ | $22.1_{\pm.2}$ |
|  | RoPE | $21.0_{\pm.1}$ | $20.0_{\pm.3}$ |
| U-SPA | None | $34.7_{\pm1.2}$ | $28.1_{\pm.4}$ |
|  | RoPE | $24.1_{\pm.3}$ | $23.2_{\pm.3}$ |
| V-SkipInit | None | $29.3_{\pm.1}$ | $34.4_{\pm.1}$ |
|  | RoPE | $24.7_{\pm.1}$ | $27.5_{\pm.1}$ |

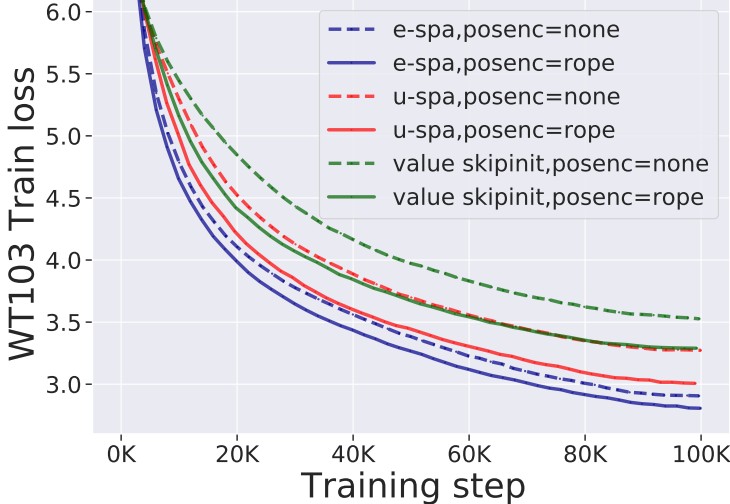

**Figure 10:** Ablation over positional encoding using a vanilla transformer with E-SPA, averaged over 3 random seeds.

**Stable residual connection (Hayou et al., 2021; Noci et al., 2022) on WikiText-103**   In Fig. 11, we provide an equivalent plot to Fig. 3 using a *Stable ResNet* (Hayou et al., 2021) rescaling of the shortcut weights ($\alpha = 1, \beta = O(\frac{1}{\sqrt{L}})$). In this case, the shortcut weight is always 1 and the residual weight $\beta$ is uniform across blocks and scales as $O(\frac{1}{\sqrt{L}})$ in depth $L$. Hayou et al. (2021) showed that such a scaling leads to non-degenerate signal propagation in large depth MLPs/CNNs without normalisation, and Noci et al. (2022) showed that the stable scaling prevents rank collapse in transformers without normalisation. We see in Fig. 11 that for large enough $\beta$, the stable residual weighting with normalisation matches the training speed of the default transformer (which is unsurprising given that once $\beta = 1$ it is exactly default Pre-LN). However, there is a small but consistent gap without normalisation (with optimal $\beta = 0.1$). Here, $L = 36$, so $\frac{1}{\sqrt{L}} = \frac{1}{6} \approx 0.17$, or alternatively if we count the 72 nonlinear layers (one self-attention and element-wise nonlinearity for each transformer block), we have $\frac{1}{\sqrt{72}} \approx 0.12$.

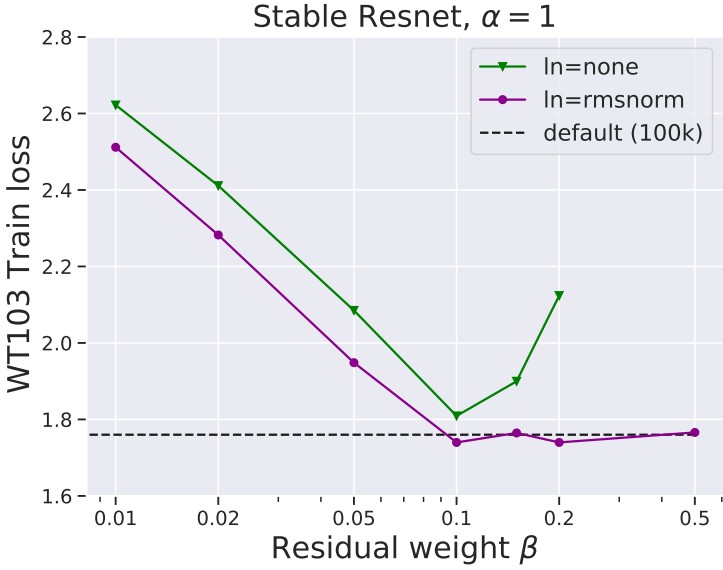

**Figure 11:** Performance of a *Stable* (Hayou et al., 2021; Noci et al., 2022) residual connection weighting on WikiText-103 for different residual weights $\beta$. Mean over 2 seeds.

**Different shortcut weights for MLP and self-attention blocks** Recall that in a transformer block Eq. (1), there are two distinct skip connections: one for the self-attention block and one for the MLP block. Moreover, we observed a training speed gap when we remove both skip connections in Fig. 2. This leads us to ask if it is possible that only one of the skips, MLP or attention, is causing this gap in training speed. In Fig. 12, we investigate this by varying the MLP shortcut weight for skipless attention blocks (left) and varying the attention shortcut weight for skipless MLP blocks (right). For all skip connections (for both MLP and self-attention blocks), we use a *normalised skip connection* ($\beta = \sqrt{1 - \alpha^2}$). We observe that removing either skip connection results in a comparable loss of training speed (the default Pre-LN on WikiText-103 obtained train loss of 1.76 after 100K steps), although having one is still better than having neither. Moreover, we observe that the attention and MLP blocks prefer slightly different shortcut weights, with dense shortcuts performing better on slightly lower weightings ($\alpha = 0.9$ or $0.97$) compared to attention shortcuts ($\alpha = 0.98$ or $0.99$), whereas our experiments in Figs. 3 and 11 use a joint weighting for both.

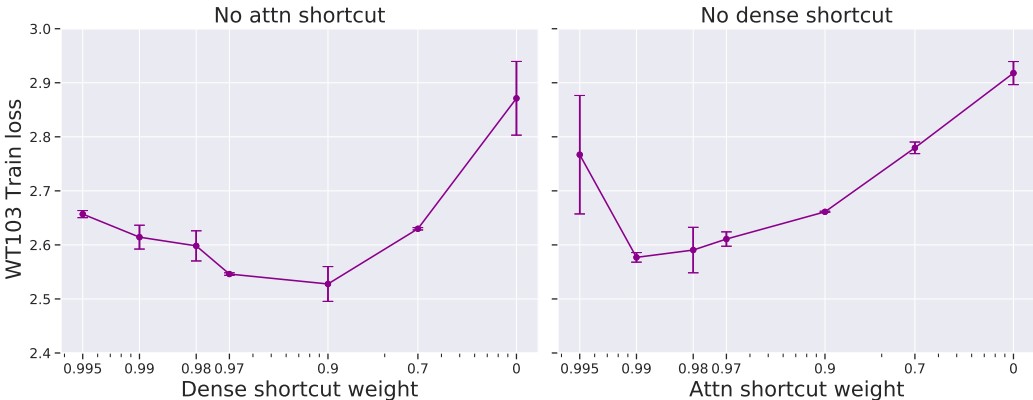

**Figure 12:** Effect of using different shortcut weights for the attention and MLP blocks, trained on WikiText-103 for 100K steps. Mean and standard deviation over 2 seeds.

**Training performance on C4** In Fig. 13 we compare the training performance of our various vanilla transfomers to the default Pre-LN transformer on C4. This is akin to Fig. 4a, which compared validation performance.

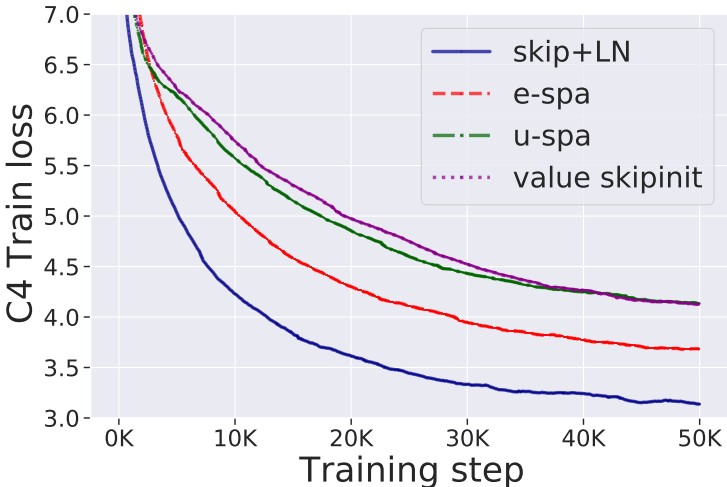

**Figure 13:** Training performance on C4, equivalent to Fig. 4a

**Downstream tasks** Typically, transformers are pre-trained on a large corpus of data before evaluation on a set of downstream tasks. To assess whether our conclusions about training performance in pre-training transfer over to downstream tasks, we assess models trained on C4 on 5 common sense downstream tasks: BoolQ (Clark et al., 2019), HellaSwag (Zellers et al., 2019), Winogrande (Sakaguchi et al., 2020), PIQA (Bisk et al., 2020), and SIQA (Sap et al., 2019). These datasets are commonly used to evaluate large pre-trained transformers (Brown et al., 2020; Rae et al., 2021; Smith et al., 2022; Hoffmann et al., 2022).

In Table 6, we see that the conclusions of pre-training on C4 largely carry over to the downstream tasks:

- Among transformers trained for the same number of steps (50k), E-SPA beats U-SPA and Value-SkipInit each on 4 out of 5 downstream tasks. However, the default transformer outperforms skipless transformers on all tasks with the same amount of training.
- With around 5 times longer training (200K and 300K steps), E-SPA achieves similar performance on downstream tasks to the standard transformer, outperforming on 2 out of 5 tasks (Winogrande and PIQA).

**Table 6:** Downstream evaluation of various models (default Pre-LN and also our skipless models) from Figs. 4a and 4b, (pre-)trained on C4. We evaluate zero-shot on 5 common sense tasks: BoolQ, HellaSwag, Winogrande, PIQA, SIQA. In brackets shows the number of steps each model was trained for. Reported values are percentage accuracies: higher is better.

| Model | BoolQ | HellaSwag | Winogrande | PIQA | SIQA |
|---|---|---|---|---|---|
| Default (50K) | **60.9** | **29.1** | 52.5 | 62.2 | **41.9** |
| V-SkipInit (50K) | 38.7 | 25.8 | 51.6 | 56.0 | 39.0 |
| U-SPA (50K) | 39.2 | 25.8 | 50.9 | 56.7 | 40.6 |
| E-SPA (50K) | 59.5 | 26.2 | 51.3 | 59.7 | 39.8 |
| E-SPA (200K) | 60.2 | 27.7 | 52.8 | **63.1** | 40.9 |
| E-SPA (300K) | 56.8 | 28.4 | **53.0** | 62.5 | 41.1 |

**Longer training on WikiText-103** In Fig. 14 we see that 4.5x more training allows a vanilla E-SPA transformer to match the validation performance of a standard transformer on WikiText-103. This mirrors our findings on C4 in Fig. 4b.

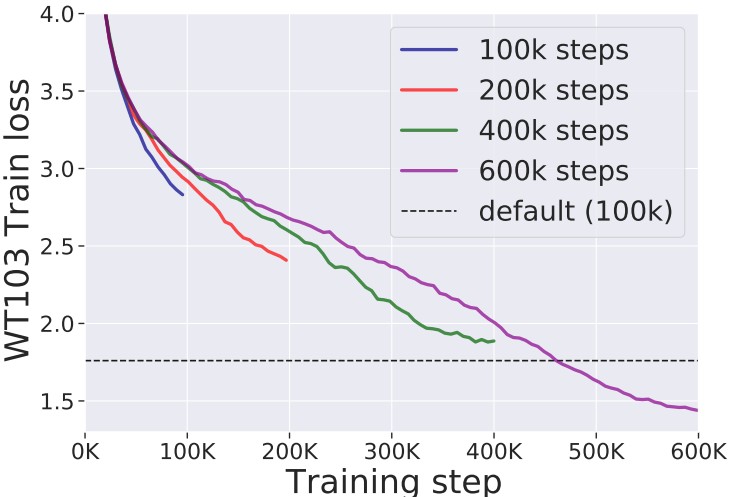

**Figure 14:** Longer training on WikiText-103 with E-SPA on a vanilla transformer. It matches the performance of standard transformer after 4.5x more iterations of training.

**Kernel matrix evolution during training** Fig. 15 shows the evolution of the empirically-computed normalized kernel matrix during training of a (finite width) vanilla E-SPA transformer on WikiText-103. The network has depicted has both attention and MLP blocks, which use DKS-transformer GeLU activations. We note that the untrained network shows good agreement with Fig. 1, despite the fact that Fig. 1 is computed for an attention-only network in the infinite width limit. We also note that while significant changes to the kernel matrix occur during training, it retains the property of being larger close the diagonal.

# H IMPLEMENTATION DETAILS

We first describe all additional general implementation details for our experiments, before going into details relevant for individual results.

## H.1 GENERAL IMPLEMENTATION DETAILS

**Datasets** We present experiments on WikiText-103 (Merity et al., 2017) and C4 (Raffel et al., 2019). For both datasets, we use the SentencePiece tokeniser (Kudo & Richardson, 2018) with vocabulary size $|V| = 32,000$. For WikiText-103 we use sequence length of 512 for both training and validation. For C4, the sequence length is 2048 for both.

**Embedding** By default the embedding layer $\mathbf{E} \in \mathbb{R}^{|V| \times d_0}$ is fan-out initialised at initialisation, so that $\mathbf{E}_{i,j} \overset{\text{i.i.d.}}{\sim} \mathcal{N}(0, \frac{1}{d_0})$. This means that each embedding entry has scale $\frac{1}{d_0}$ at initialisation, whereas our Gram matrices (which are dimension normalised) expect each entry to have scale 1 at initialisation. To get by this, in our proposed vanilla methods we rescale the embedding matrix by a fixed factor of $\sqrt{d_0}$, so that $\mathbf{E} \leftarrow \sqrt{d_0}\mathbf{E}$ which is then treated as a lookup table. At initialisation, this has the same effect using RMSNorm on the embedding layer. In all our experiments, the unembedding layer weights are shared with the embedding weights. We do not use the embedding layer positional encoding introduced by Vaswani et al. (2017).

**Model** In all our experiments apart from Table 3, the model width $d = 1024$ across all blocks. We use 8 head multi-head attention, so that $d^k = 128$. The MLP block consists of a single hidden layer with width $4d$, with input and output dimensions both equal to $d$. All our experiments use the Pre-LN transformer block Eq. (1), rather than Post-LN. On WikiText-103, we use a 36 block transformer by default for all experiments apart from Table 3. On C4, we use a 32 block transformer due to memory constraints. Any normalisation layer we consider is RMSNorm (Zhang & Sennrich, 2019), which is simpler than Layer Normalisation (Ba et al., 2016) and is commonly used in transformers (Rae

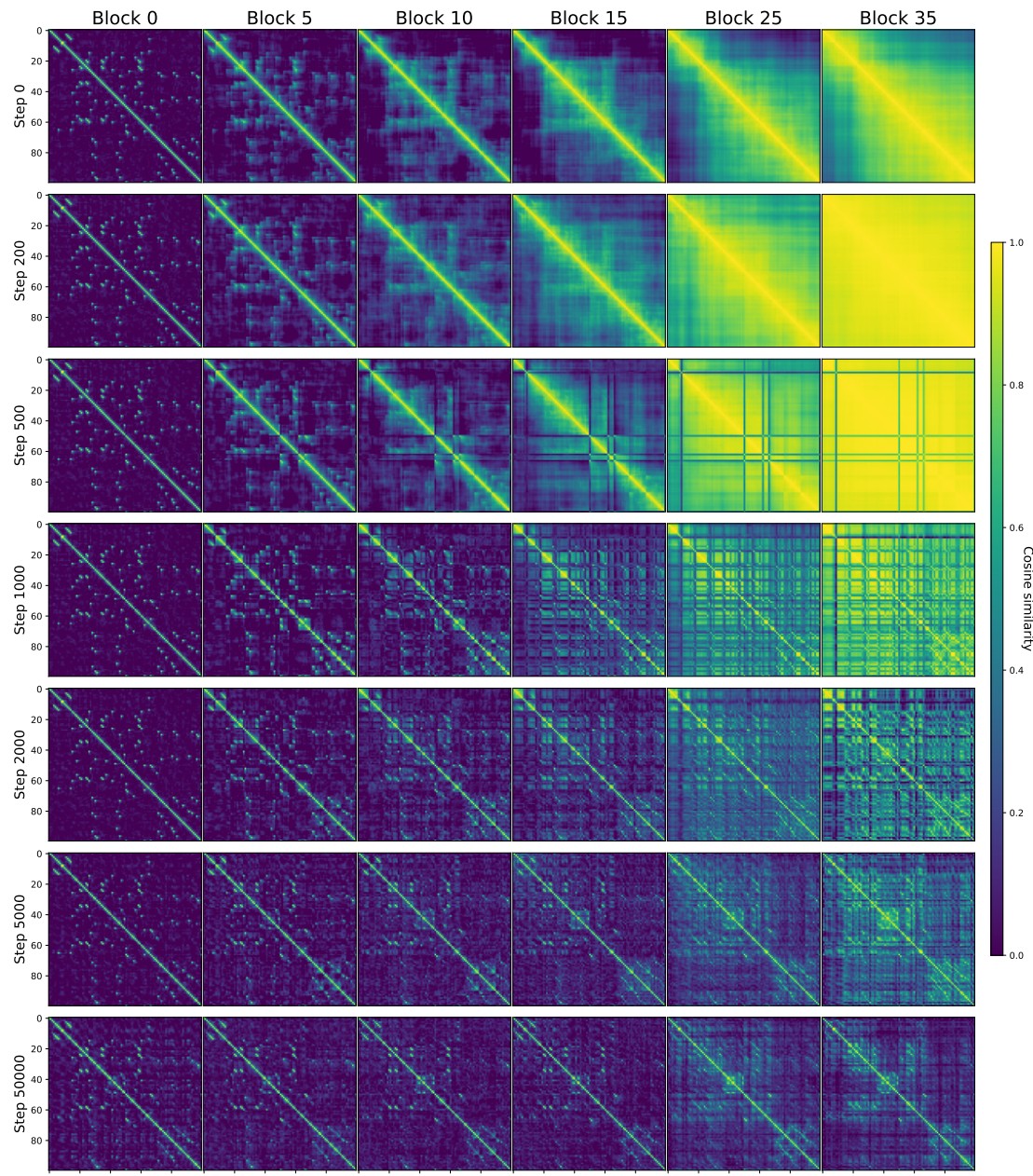

**Figure 15:** The normalized kernel matrix for various transformer blocks at three different stages of training of a vanilla E-SPA transformer on WikiText-103. Step 0 corresponds to the untrained network. Note that this is computed with an actual finite-width network with randomly sampled (and then trained) parameters, unlike Fig. 1 which corresponds to the infinite-width limit (at initialisation).

et al., 2021). By default, our models use RoPE positional encoder (Su et al., 2021) (apart from the ablation in Fig. 10).

**Parameter initialisations** By default, all weight matrices are fan-in initialised $\mathcal{N}(0, \frac{\sigma^2}{\text{fan-in}})$ with $\sigma = 1$. The two exceptions for this are: 1) when using orthogonal intiailisation, we use the scaled-corrected uniform orthogonal initialisation (Martens et al., 2021) with scale $\sigma = 1$ (which for square matrices is just an orthogonal matrix sampled from the Haar measure Meckes (2019) multiplied by $\sigma$), and 2) for the parameter matrix immediately after the activation we set $\sigma$ to take input activation norm ("q-values" or diagonals of Gram matrices) of 1 to output activation norm of 1. In the latter

case, $\sigma = 1$ by construction for activations transformed by DKS/TAT Martens et al. (2021); Zhang et al. (2022). All bias parameters are initialised to 0.

**(Transformed) Activations** For DKS (Martens et al., 2021) we set slope parameter $\xi = 5$, and for TAT (Zhang et al., 2022) with leaky ReLU, we set $\eta = 0.9$. Both values were chosen by a small hyperparameter sweep on WikiText-103. The DKS and TAT transformations are chosen without consideration of the attention blocks, where the transformer can be viewed as an MLP (potentially with residual connections). Unless stated otherwise, all skipless transformers used a DKS-transformed GeLU as the nonlinearity in the MLP block by default.

**Loss** We perform next-token prediction, with softmax cross entropy loss. More concretely, if $\mathbf{X}_L \in \mathbb{R}^{T \times d}$ denotes our final layer representation and $\mathbf{E} \in \mathbb{R}^{|V| \times d}$ denotes our embedding layer, then we obtain logits $\mathbf{X}_L \mathbf{E}^\top \in \mathbb{R}^{T \times |V|}$ for each location for a $|V|$-way classification. The loss is softmax cross entropy, obtained by using each location $i$ to predict the identity, in $|V|$, of the token at location $i + 1$. In our training loss results, we use an exponential moving average with smoothing factor 0.01, to reduce randomness from mini-batching.

**Optimiser** We use Adam optimiser (Kingma & Ba, 2014) with global gradient clipping of 0.1 by default (Pascanu et al., 2013). We do not use weight decay in our experiments.

**Training** We use mini-batching of 16 sequences for WikiText-103 and 8 for C4, due to memory constraints. Unless stated otherwise, we train for 100K steps on WikiText-103 and 50K steps for C4.

**Learning rate** In all experiments we use a linear learning rate warm-up period of 4000 steps which increases from 0 to a maximum value (which is tunable hyperparameter). In the remaining iterations, we use a "one-cycle" cosine learning rate schedule (Loshchilov & Hutter, 2016) which reaches 0 at the final iteration. The maximum learning rate was tuned in $\{1, 2, 3, 5\} \times 10^{-n}$ over $n \in \mathbb{Z}$ for all experiments. As the embedding layer does not change across the different depths and skip/normalisation settings we considered, we use a different, fixed maximum learning rate for the embedding layer, which was chosen to match the optimal learning rate for the default transformer ($2 \times 10^{-4}$ for WikiText-103 and $5 \times 10^{-4}$ for C4) and not tuned further.

## H.2 Additional implementation details for individual experiments

**Attention-only kernel matrix evolution: Figs. 1 and 7** We calculate the kernel matrix $\Sigma$ evolution directly in $T \times T$ kernel matrix-space, where $T = 100$. Our input kernel matrix $\Sigma$ is sampled assuming a fraction $r = 0.02$ of repeated tokens, with value of 1 if the token is repeated, and 0 else. For all configurations of skip/normalisation/attention modifications corresponding to a row of Fig. 7, we use 8 heads.

We now detail how each operation in any configuration of Fig. 7 affects the kernel matrix. From this it should be possible to reconstruct the kernel evolution for any row in Fig. 7. The 3 possible operations are: 1) attention, 2) skip connection, or 3) LN/RMSNorm operation.

1. **Attention** Because our SPA methods (second and third rows) are agnostic to the number of heads at initialisation (all attention matrices in a self-attention block are the same across heads at initialisation), we apply Eq. (6) directly, so that a single attention block amounts to $\Sigma \leftarrow \mathbf{A} \Sigma \mathbf{A}^\top$ for attention matrix $\mathbf{A}$ and incoming kernel matrix $\Sigma$. Our E-SPA method uses $\gamma_L = 0.005$ and our U-SPA method uses $\rho_L = 0.8$.

   For all other rows, the self-attention operation uses ALiBi (Press et al., 2022), a popular positional encoder which uses *head-dependent* pre-softmax bias matrices. More specifically, from the default pre-softmax bias matrices given by ALiBi (for 8 heads), we obtain 8 attention matrices $\{\mathbf{A}_h\}_{h=1}^8$ using the softmax operation (which is exact assuming zero query-key dot product at initialisation). Because the different heads in transformers are typically concatenated along 8 equal size *fractions* of the total width $d$, the kernel evolution of an attention block on kernel matrix $\Sigma$ with 8-head ALiBi corresponds to (Martens et al.,

**Table 7:** Shortcut and residual weights for Table 2. Weights are presented as $(\alpha_{\text{attn}}, \beta_{\text{attn}})/(\alpha_{\text{MLP}}, \beta_{\text{MLP}})$, where $\alpha_{\text{attn}}$ denotes the shortcut weight for attention block and $\alpha_{\text{MLP}}$ denotes the shortcut weight for the MLP block. Residual weights $\beta$ are defined similarly. * denotes that the initial value of a trainable parameter.

| Attention | Skip | LN | w/o LN |
|---|---|---|---|
| Default | Default | (1,1)/(1,1) | – |
| | Normalised | $(0.99, \sqrt{1 - 0.99^2})/(0.99, \sqrt{1 - 0.99^2})$ | $(0.999, \sqrt{1 - 0.999^2})/(0.99, \sqrt{1 - 0.99^2})$ |
| | Stable ($\alpha$=1) | (1,0.2)/(1,0.15) | (1,0.05)/(1,0.1) |
| | SkipInit | (1,0*)/(1,0*) | (1,0*)/(1,0*) |
| U-SPA | Normalised | $(0.98, \sqrt{1 - 0.98^2})/(0.99, \sqrt{1 - 0.99^2})$ | $(0.995, \sqrt{1 - 0.995^2})/(0.995, \sqrt{1 - 0.995^2})$ |
| E-SPA | Normalised | $(0.9975, \sqrt{1 - 0.9975^2})/(0.995, \sqrt{1 - 0.995^2})$ | $(0.9975, \sqrt{1 - 0.9975^2})/(0.99, \sqrt{1 - 0.99^2})$ |

2021):

$$\Sigma \leftarrow \frac{1}{8} \sum_{h=1}^{8} \mathbf{A}_h \Sigma \mathbf{A}_h^\top$$

2. For a skip connection with shortcut weight $\alpha$ and residual weight $\beta$, if $\Sigma_{\text{attn}}(\Sigma)$ denotes the output of a kernel matrix $\Sigma$ after an self-attention operation (i.e. from point 1. above), then an incoming kernel matrix $\Sigma$ gets mapped to:

$$\Sigma \leftarrow \alpha^2 \Sigma + \beta^2 \Sigma_{\text{attn}}(\Sigma).$$

3. For a normalisation operation, the incoming kernel matrix $\Sigma$ gets mapped to:

$$\Sigma \leftarrow \text{diag}(\Sigma)^{-\frac{1}{2}} \cdot \Sigma \cdot \text{diag}(\Sigma)^{-\frac{1}{2}}$$

**Hyperparameter tuning of skipless transformers** For experiments on WikiText-103 with 100K steps, for our U-SPA transformers we tuned $\rho_L \in \{0.6, 0.8\}$, and for our E-SPA transformers we tuned $\gamma_L \in \{0.005, 0.2\}$. For all other settings (longer/deeper training on WikiText-103 or any C4 experiment), we used the default $\gamma_L = 0.005$ and $\rho_L = 0.8$. All hyperparameters throughout our work are tuned based on training loss.

**Hyperparameter tuning of transformers with normalised skips** All experiments with skip connections (i.e. shortcut weight $\alpha \neq 0$ for either the MLP or attention block) use untransformed GeLU activation in the MLP blocks. We combine E-SPA with normalised skips as described in Appendix F. We note that for high shortcut weights $\alpha$ and small final decay rate $\gamma_L$, then the value of attention matrix diagonal $\lambda_\alpha$, Eq. (24), may not be real. This is because the input to the square root in Eq. (24) may be negative. To get by this, we tune $\gamma_L \in \{0.005, 0.2, 0.4, 0.6\}$ when using a 36 block transformer on WikiText-103 and $\gamma_L \in \{0.005, 0.2, 0.4, 0.7, 1\}$ for Table 2, which used a 32 block transformer on C4.

For Table 2, the normalised skip connections had *separately* tuned attention and MLP shortcut weights, as we observed a difference in the optimal shortcut weight for self-attention vs MLP blocks in Fig. 12. We tuned the attention shortcut weight in the range $\alpha \in \{0.98, 0.99, 0.995, 0.9975, 0.999\}$ and the MLP shortcut weight in the range $\alpha \in \{0.98, 0.99, 0.995\}$. Likewise, both the stable residual weights were tuned in $\beta \in \{0.05, 0.1, 0.15, 0.2\}$ separately for self-attention and MLP skips. The selected shortcut/residual weights (using validation performance) are presented in Table 7.

**Depth scaling** Due to memory constraints, for our deeper networks in Table 3 we use width $d = 512$ rather than 1024, with 8 heads to give $d^k = 64$. All depth scaling runs used a DKS-transformed GeLU with $\xi = 5$.

## I  THEORETICAL RESULTS

In this section we state and prove our theoretical results, including Theorems 1 and 2:

**Theorem 1.** *(Non-negativity for U-SPA) Let* $\Sigma = (1 - \rho)\mathbf{I}_T + \rho\mathbf{1}\mathbf{1}^\top$ *and* $\Sigma' = (1 - \rho')\mathbf{I}_T + \rho'\mathbf{1}\mathbf{1}^\top$, *with respective (positive) Cholesky factors* $\mathbf{L}$ *and* $\mathbf{L}'$. *Then if* $\rho \leq \rho'$, *we have* $\mathbf{L}'\mathbf{L}^{-1}$ *is elementwise non-negative.*

**Theorem 2.** *(Non-negativity for E-SPA) Let matrices* $(\Sigma)_{i,j} = \exp(-\gamma|i - j|))$ *and* $(\Sigma')_{i,j} = \exp(-\gamma'|i - j|))$ *with respective (positive) Cholesky factors* $\mathbf{L}$ *and* $\mathbf{L}'$. *Then if* $\gamma \geq \gamma'$, *we have* $\mathbf{L}'\mathbf{L}^{-1}$ *is elementwise non-negative.*

We prove Theorem 1 second as it is more involved.

### I.1   PROOF OF THEOREM 2

We actually prove Theorem 2 as a corollary of Theorem 3, which provides the analytic form for $\mathbf{L}'\mathbf{L}^{-1}$.

**Theorem 3.** *Let matrices* $(\Sigma)_{i,j} = \exp(-\gamma|i - j|))$ *and* $(\Sigma')_{i,j} = \exp(-\gamma'|i - j|))$ *with respective (positive) Cholesky factors* $\mathbf{L}$ *and* $\mathbf{L}'$. *Then if* $\gamma \geq \gamma' > 0$, *we have* $\mathbf{A} = \mathbf{L}'\mathbf{L}^{-1}$ *takes the following form:*

$$\mathbf{A}_{i,j} = \begin{cases} 1, & \text{if } i = j = 1 \\ \frac{a(\gamma')}{a(\gamma)}, & \text{if } i = j \neq 1 \\ \left[\exp(-\gamma') - \frac{a(\gamma')}{a(\gamma)}\exp(-\gamma)\right]\exp\left(-\gamma'(i - j - 1)\right), & \text{if } j = 1, i > j \\ \frac{a(\gamma')}{a(\gamma)}\left[\exp(-\gamma') - \exp(-\gamma)\right]\exp\left(-\gamma'(i - j - 1)\right), & \text{if } j \neq 1, i > j \\ 0 & \text{else} \end{cases} \quad (27)$$

*where* $a(\gamma) = \sqrt{1 - \exp(-2\gamma)}$, *and likewise* $a(\gamma') = \sqrt{1 - \exp(-2\gamma')}$

*Proof of Theorem 2.* From Theorem 3, we have the analytic form of $\mathbf{A}$. Clearly $a(\gamma), a(\gamma')$ are positive, and moreover if $\gamma' \leq \gamma$, then $\exp(-\gamma') - \exp(-\gamma)$ is non-negative.

Finally, because $\frac{a(\gamma')}{a(\gamma)} \leq 1$ when $\gamma' \leq \gamma$, then $\exp(-\gamma') - \frac{a(\gamma')}{a(\gamma)}\exp(-\gamma)$ is non-negative too. □

To prove Theorem 3, we first compute what the analytic form of Cholesky factor $\mathbf{L}$ for $(\Sigma)_{i,j} = \exp(-\gamma|i - j|))$ takes in Lemma 1.

**Lemma 1.** *Let* $(\Sigma)_{i,j} = \exp(-\gamma|i - j|))$ *with (positive) Cholesky factors* $\mathbf{L}$ *such that* $\mathbf{L}\mathbf{L}^\top = \Sigma$. *Then, we have:*

$$\mathbf{L}_{i,j} = \begin{cases} \exp(-\gamma|j - i|), & \text{if } j = 1, i \geq j \\ \sqrt{1 - \exp(-2\gamma)}\exp(-\gamma|j - i|), & \text{if } j \neq 1, i \geq j \\ 0 & \text{else} \end{cases} \quad (28)$$

*Proof of Lemma 1.* It is clear that $\Sigma$ is positive semi definite, as it is the covariance matrix of a stationary Ornstein-Uhlenbeck process, hence a Cholesky factor must exist. We now show that it is $\mathbf{L}$, Eq. (28).

If we define $l = \min(m, n)$, then we have:

$$(\mathbf{L}\mathbf{L}^\top)_{m,n} = \mathbf{L}_{m,1}\mathbf{L}_{n,1} + \sum_{i=2}^{l}\mathbf{L}_{m,i}\mathbf{L}_{n,i}$$

$$= \exp(-\gamma(m + n - 2)) + (1 - \exp(-2\gamma))\sum_{i=2}^{l}\exp(-\gamma(m + n - 2i))$$

$$= \exp(-\gamma(m + n - 2))\left[1 + \left(1 - \exp(-2\gamma)\right)\sum_{i=1}^{l-1}\exp(2\gamma i)\right]$$

$$= \exp(-\gamma(m + n - 2))\left[1 + \left(1 - \exp(-2\gamma)\right)\exp(2\gamma)\frac{1 - \exp(2\gamma(l - 1))}{1 - \exp(2\gamma)}\right]$$

$$
\begin{aligned}
&= \exp(-\gamma(m+n-2))\big[1 - \big(1 - \exp(2\gamma(l-1))\big)\big]\\
&= \exp(-\gamma(m+n-2l))\\
&= \exp(-\gamma|m-n|)\\
&= \Sigma_{m,n}
\end{aligned}
$$

By the uniqueness of (positive) Cholesky factors, the proof is complete. $\qquad\square$

*Proof of Theorem 3.* We want to show $(\mathbf{AL})_{k,l} = (\mathbf{L}')_{k,l}, \forall k,l$. This is clearly true for the top diagonal $k = l = 1$.

We now show this for the rest of the first column, when $l = 1, k > 1$:

$$
\begin{aligned}
(\mathbf{AL})_{k,1} =&\, \mathbf{A}_{k,1} + \mathbf{A}_{k,k}\mathbf{L}_{k,1} + \sum_{i=2}^{k-1} \mathbf{A}_{k,i}\mathbf{L}_{i,1}\\
=&\, \Big[\exp(-\gamma') - \frac{a(\gamma')}{a(\gamma)}\exp(-\gamma)\Big]\exp\big(-\gamma'(k-2)\big)\\
&+ \frac{a(\gamma')}{a(\gamma)}\exp(-\gamma(k-1))\\
&+ \sum_{i=2}^{k-1} \frac{a(\gamma')}{a(\gamma)}\big[\exp(-\gamma') - \exp(-\gamma)\big]\exp\big(-\gamma'(k-i-1)\big)\exp(-\gamma(i-1)) \quad (29)
\end{aligned}
$$

Applying the geometric sum to Eq. (29) yields:

$$
\begin{aligned}
(\mathbf{AL})_{k,1} =&\, \Big[\exp(-\gamma') - \frac{a(\gamma')}{a(\gamma)}\exp(-\gamma)\Big]\exp\big(-\gamma'(k-2)\big)\\
&+ \frac{a(\gamma')}{a(\gamma)}\exp(-\gamma(k-1))\\
&+ \frac{a(\gamma')}{a(\gamma)}\Big[\exp(-\gamma') - \exp(-\gamma)\big]\exp\big(-\gamma'(k-3)-\gamma\big)\frac{1-\exp((k-2)(\gamma'-\gamma))}{1-\exp(\gamma'-\gamma)}\Big]\\
=&\, \exp(-\gamma'(k-1))\\
&+ \frac{a(\gamma')}{a(\gamma)}\Big[-\exp\big(-\gamma'(k-2)-\gamma\big) + \exp(-\gamma(k-1))\\
&\qquad + \exp\big(-\gamma'(k-3)-\gamma\big)\exp(-\gamma')\big(1-\exp((k-2)(\gamma'-\gamma))\big)\Big]\\
=&\, \exp(-\gamma'(k-1))\\
&+ \frac{a(\gamma')}{a(\gamma)}\Big[-\exp\big(-\gamma'(k-2)-\gamma\big) + \exp(-\gamma(k-1))\\
&\qquad + \exp\big(-\gamma'(k-2)-\gamma\big)\big(1-\exp((k-2)(\gamma'-\gamma))\big)\Big]\\
=&\, \exp(-\gamma'(k-1))\\
&+ \frac{a(\gamma')}{a(\gamma)}\Big[-\exp\big(-\gamma'(k-2)-\gamma\big) + \exp\big(-\gamma'(k-2)-\gamma\big)\\
&\qquad + \exp(-\gamma(k-1)) - \exp(-\gamma(k-1))\Big]\\
=&\, \exp(-\gamma'(k-1))\\
=&\, \mathbf{L}'_{k,1}
\end{aligned}
$$

as desired. Likewise, if $l > 1, l \le k$:

$$
(\mathbf{AL})_{k,l} = \mathbf{A}_{k,k}\mathbf{L}_{k,l} + \sum_{i=l}^{k-1} \mathbf{A}_{k,i}\mathbf{L}_{i,l}
$$

$$
=a(\gamma')\left[\exp(-\gamma(k-l)+\sum_{i=l}^{k-1}\left[\exp(-\gamma')-\exp(-\gamma)\right]\exp\big(-\gamma'(k-i-1)\big)\exp(-\gamma(i-l))\right]
$$

$$
=a(\gamma')\left[\exp(-\gamma(k-l)+\left[\exp(-\gamma')-\exp(-\gamma)\right]\exp\big(-\gamma'(k-l-1)\big)\frac{1-\exp((\gamma'-\gamma)(k-l))}{1-\exp(\gamma'-\gamma)}\right]
$$

$$
=a(\gamma')\left[\exp(-\gamma(k-l)+\exp(-\gamma')\exp\big(-\gamma'(k-l-1)\big)\big(1-\exp((\gamma'-\gamma)(k-l))\big)\right]
$$

$$
=a(\gamma')\left[\exp(-\gamma(k-l)+\exp\big(-\gamma'(k-l)\big)\big(1-\exp((\gamma'-\gamma)(k-l))\big)\right]
$$

$$
=a(\gamma')\left[\exp\big(-\gamma'(k-l)\big)+\exp\big(-\gamma(k-l)\big)-\exp\big(-\gamma(k-l)\big)\right]
$$

$$
=a(\gamma')\exp\big(-\gamma'(k-l)\big)
$$

$$
=\mathbf{L}'_{k,l}
$$

$\square$

## I.2 Proof of Theorem 1

**Theorem 1.** *(Non-negativity for U-SPA)  Let $\Sigma = (1-\rho)\mathbf{I}_T + \rho\mathbf{1}\mathbf{1}^\top$ and $\Sigma' = (1-\rho')\mathbf{I}_T + \rho'\mathbf{1}\mathbf{1}^\top$, with respective (positive) Cholesky factors $\mathbf{L}$ and $\mathbf{L}'$. Then if $\rho \leq \rho'$, we have $\mathbf{L}'\mathbf{L}^{-1}$ is elementwise non-negative.*

### I.2.1 Preliminaries

Before diving into the actual proof of the theorem we will derive several useful properties and notations. First we make a slight notational change from the main text of the theorem by replacing $\Sigma$, which depends on $\rho$, with $C_n(x)$, where $n$ represents the size of the matrix, while $x$ replaces $\rho$. In addition, since the case of $\rho = \rho'$ (or $x = y$ in the rest of the proof) is trivial, since then the resulting matrix is the identity, we will restrict ourselves to dealing with the case where $0 < x < y < 1$. In any of the mathematical derivations we will denote with capital English letters (e.g. $A, B, C, T$) any temporary expressions, that will be expanded on the following lines. Note that these are never general definitions, so they might be used multiple times for different expressions.

We will denote vectors and vector functions in bold and scalars and scalar function in standard font.

Let $\mathbf{1}_n$ be the $n$-dimensional vector with only ones:

$$\mathbf{1}_n = \begin{bmatrix} 1 \\ \dots \\ 1 \end{bmatrix} \in \mathbb{R}^n$$

We will denote with $\boldsymbol{x}_n$ the $n$-dimensional vector with only $x$'s:

$$\boldsymbol{x}_n = x\mathbf{1}_n$$

First, we define the linear map $p_n(x)$ as:

$$p_n(x) = nx + 1$$

**Proposition 1.** *The vector $\mathbf{1}_n$ is an eigenvector of $C_n(x)$ with an eigenvalue $p_{n-1}(x)$.*

*Proof.* Directly calculating the $k$-th entry of the product $C_n(x)\mathbf{1}_n$ gives:

$$[C_n(x)\mathbf{1}_n]_k = \sum_{i=1}^n [C_n(x)]_{k,i} = \sum_{i=1}^n \delta_i^k + (1-\delta_i^k)x = 1 + (n-1)x$$

which directly implies that:

$$C_n(x)\mathbf{1}_n = p_{n-1}(x)\mathbf{1}_n. \qquad \square$$

**Corollary 1.** *The vector $\mathbf{1}_n$ is an eigenvector of $C_n(x)^{-1}$ with an eigenvalue $\frac{1}{p_{n-1}(x)}$.*

**Corollary 2.** $C_n(x)^{-1}\boldsymbol{x}_n = \frac{1}{p_{n-1}(x)}\boldsymbol{x}_n.$

**Corollary 3.** $\boldsymbol{x}_n^T C_n(x)^{-1}\boldsymbol{x}_n = \frac{nx^2}{p_{n-1}(x)}.$

Further we define the following useful functions:

$$
\begin{aligned}
d_n(x) &= 1 - \boldsymbol{x}_n^T C_n(x)^{-1}\boldsymbol{x}_n = 1 - \frac{nx^2}{1+(n-1)x} = \frac{1+nx-x-nx^2}{1+(n-1)x} = \frac{(1-x)(nx+1)}{1+(n-1)x} \\
&= (1-x)\frac{p_n(x)}{p_{n-1}(x)}. \\
r_n(x) &= \sqrt{d_n(x)}p_{n-1}(x) = \sqrt{(1-x)p_n(x)p_{n-1}(x)} \\
&\Rightarrow \frac{r_n(x)}{p_{n-1}(x)} = \sqrt{d_n(x)} \\
&\Rightarrow \frac{r_n(x)}{\sqrt{d_n(x)}} = p_{n-1}(x)
\end{aligned}
$$

**Definition 1.** *The function $\boldsymbol{v}_n : \mathbb{I}[0,1] \times \mathbb{I}[0,1] \to \mathbb{R}^n$ maps the two bounded scalars $x$ and $y$ to the vector:*

$$\boldsymbol{v}_n(x,y) = L_n(x)^{-T} L_n(y)^{-1} \boldsymbol{y}_n - \sqrt{\frac{d_n(y)}{d_n(x)}} C_n(x)^{-1}\boldsymbol{x}_n.$$

### I.2.2 LEMMAS

**Lemma 2.** *If $0 < x < y < 1$ then all entries of the vector $\boldsymbol{v}_n(x,y)$ are non-negative - $[\boldsymbol{v}_n(x,y)]_i \geqslant 0$.*

**Lemma 3.** *If $0 < x < y < 1$ then the function $g_n(x,y) = \frac{y(1-y)p_{n-1}(x)}{r_n(x)r_n(y)} - \sqrt{\frac{d_{n+1}(y)}{d_{n+1}(x)}}\frac{x}{p_n(x)}$ is non-negative.*

**Lemma 4.** *If $0 < x < y < 1$ then the function $f_n(x,y) = \frac{xy(1-y)}{r_n(x)r_n(y)} + \sqrt{\frac{d_{n+1}(y)}{d_{n+1}(x)}}\frac{x}{p_n(x)} - \sqrt{\frac{d_n(y)}{d_n(x)}}\frac{x}{p_{n-1}(x)}$ is negative.*

**Definition 2.** *The partial sum the functions $f_n(x,y)$ from $k+1$ to $n-1$ will be denote by $h_k^n(x,y) = \sum_{i=k+1}^{n-1} f_i(x,y)$, which from Lemma 4 follow are always negative.*

### I.2.3 MAIN PROOF

The theorem is proven by induction.

First for $n=1$ we have that $C_1(x) = C_1(y) = [1]$, which implies that $L_1(x) = L_1(y) = [1]$ and the condition is trivially satisfied.

Now assuming that the statement is true for all integers up to $n$, we will prove that it also holds for $n+1$:

$$C_{n+1}(x) = \begin{bmatrix} C_n(x) & \boldsymbol{x}_n \\ \boldsymbol{x}_n^T & 1 \end{bmatrix}$$

$$L_{n+1}(x) = \begin{bmatrix} L_n(x) & \boldsymbol{0}_n \\ \boldsymbol{x}_n^T L_n(x)^{-T} & \sqrt{d_n(x)} \end{bmatrix}$$

$$L_{n+1}(x)L_{n+1}(x)^T = \begin{bmatrix} L_n(x) & \boldsymbol{0}_n \\ \boldsymbol{x}_n^T L_n(x)^{-T} & \sqrt{d_n(x)} \end{bmatrix}\begin{bmatrix} L_n(x)^T & L_n(x)^{-1}\boldsymbol{x}_n \\ \boldsymbol{0}_n^T & \sqrt{d_n(x)} \end{bmatrix}$$

$$= \begin{bmatrix} L_n(x)L_n(x)^T & \boldsymbol{x}_n \\ \boldsymbol{x}_n^T & A \end{bmatrix}$$

$$A = \boldsymbol{x}_n^T L_n(x)^{-T} L_n(x)^{-1}\boldsymbol{x}_n + d_n(x) = \boldsymbol{x}_n^T C_n(x)^{-1}\boldsymbol{x}_n + d_n(x) = 1$$

$$L_{n+1}(x)^{-1} = \begin{bmatrix} L_n(x)^{-1} & \boldsymbol{0}_n \\ -\frac{1}{r_n(x)}\boldsymbol{x}_n^T & \frac{1}{\sqrt{d_n(x)}} \end{bmatrix}$$

$$L_{n+1}(x)L_{n+1}(x)^{-1} = \begin{bmatrix} L_n(x) & \boldsymbol{0}_n \\ \boldsymbol{x}_n^T L_n(x)^{-T} & \sqrt{d_n(x)} \end{bmatrix}\begin{bmatrix} L_n(x)^{-1} & \boldsymbol{0}_n \\ -\frac{1}{r_n(x)}\boldsymbol{x}_n^T & \frac{1}{\sqrt{d_n(x)}} \end{bmatrix}$$

$$= \begin{bmatrix} I_n & \boldsymbol{0}_n \\ \boldsymbol{x}_n^T L_n(x)^{-T} L_n(x)^{-1} - \frac{\sqrt{d_n(x)}}{r_n(x)}\boldsymbol{x}_n^T & 1 \end{bmatrix}$$

$$= \begin{bmatrix} I_n & \boldsymbol{0}_n \\ \boldsymbol{x}_n^T C_n(x)^{-1} - \frac{1}{p_{n-1}(x)}\boldsymbol{x}_n^T & 1 \end{bmatrix} = \begin{bmatrix} I_n & \boldsymbol{0}_n \\ \boldsymbol{0}_n^T & 1 \end{bmatrix}$$

$$L_{n+1}(y)L_{n+1}(x)^{-1} = \begin{bmatrix} L_n(y) & \boldsymbol{0}_n \\ \boldsymbol{y}_n^T L_n(y)^{-T} & \sqrt{d_n(y)} \end{bmatrix}\begin{bmatrix} L_n(x)^{-1} & \boldsymbol{0}_n \\ -\frac{1}{r_n(x)}\boldsymbol{x}_n^T & \frac{1}{\sqrt{d_n(x)}} \end{bmatrix}$$

$$= \begin{bmatrix} L_n(y)L_n(x)^{-1} & \boldsymbol{0}_n \\ \boldsymbol{y}_n^T L_n(y)^{-T} L_n(x)^{-1} - \sqrt{\frac{d_n(y)}{d_n(x)}}\frac{1}{p_{n-1}(x)}\boldsymbol{x}_n^T & \sqrt{\frac{d_n(y)}{d_n(x)}} \end{bmatrix}$$

$$= \begin{bmatrix} L_n(y)L_n(x)^{-1} & \boldsymbol{0}_n \\ \boldsymbol{v}_n(x,y)^T & \sqrt{\frac{d_n(y)}{d_n(x)}} \end{bmatrix}$$

Using the fact that $L_n(y)L_n(x)^{-1}$ is a lower triangular and non-negative matrix by the inductive assumption combined with Lemma 2 and the fact that $d_n(x) \geqslant 0$ it follows that $L_{n+1}(y)L_{n+1}(x)^{-1}$ is also a lower triangular non-negative matrix.

### I.2.4  PROOF OF LEMMA 2

First we will inspect the evolution of $\boldsymbol{v}_n(x, y)$ as we increase $n$:

$$
\boldsymbol{v}_{n+1}(x, y) = L_{n+1}(x)^{-T} L_{n+1}(y)^{-1} \boldsymbol{y}_{n+1} - \sqrt{\frac{d_{n+1}(y)}{d_{n+1}(x)}} C_{n+1}(x)^{-1} \boldsymbol{x}_{n+1}
$$

$$
= A - \sqrt{\frac{d_{n+1}(y)}{d_{n+1}(x)}} \frac{1}{p_n(x)} \boldsymbol{x}_{n+1}
$$

$$
A = L_{n+1}(x)^{-T} L_{n+1}(y)^{-1} \boldsymbol{y}_{n+1}
$$

$$
= \begin{bmatrix} L_n(x)^{-1} & -\frac{1}{r_n(x)} \boldsymbol{x}_n \\ \boldsymbol{0}_n^T & \frac{1}{\sqrt{d_n(x)}} \end{bmatrix} \begin{bmatrix} L_n(y)^{-1} & \boldsymbol{0}_n \\ -\frac{1}{r_n(y)} \boldsymbol{y}_n^T & \frac{1}{\sqrt{d_{n+1}(y)}} \end{bmatrix} \boldsymbol{y}_{n+1}
$$

$$
= \begin{bmatrix} L_n(x)^{-1} & -\frac{1}{r_n(x)} \boldsymbol{x}_n \\ \boldsymbol{0}_n^T & \frac{1}{\sqrt{d_n(x)}} \end{bmatrix} \begin{bmatrix} L_n(y)^{-1} \boldsymbol{y}_n \\ -\frac{ny^2}{r_n(y)} + \frac{y}{\sqrt{d_n(y)}} \end{bmatrix}
$$

$$
= \begin{bmatrix} L_n(x)^{-1} & -\frac{1}{\sqrt{d_n(x)p_{n-1}(x)}} \boldsymbol{x}_n \\ \boldsymbol{0}_n^T & \frac{1}{\sqrt{d_n(x)}} \end{bmatrix} \begin{bmatrix} L_n(y)^{-1} \boldsymbol{y}_n \\ \frac{B}{r_n(y)} \end{bmatrix}
$$

$$
B = y p_{n-1}(y) - ny^2 = y((n-1)y+1) - ny^2 = y - y^2 = y(1-y)
$$

$$
A = \begin{bmatrix} L_n(x)^{-1} & -\frac{1}{r_n(x)} \boldsymbol{x}_n \\ \boldsymbol{0}_n^T & \frac{1}{\sqrt{d_n(x)}} \end{bmatrix} \begin{bmatrix} L_n(y)^{-1} \boldsymbol{y}_n \\ \frac{y(1-y)}{r_n(y)} \end{bmatrix}
$$

$$
= \begin{bmatrix} L_n(x)^{-1} L_n(y)^{-1} \boldsymbol{y}_n - \frac{y(1-y)}{r_n(x)r_n(y)} \boldsymbol{x}_n \\ \frac{y(1-y)}{\sqrt{d_n(x)}r_n(y)} \end{bmatrix}
$$

$$
= \begin{bmatrix} B \\ \frac{y(1-y)}{\sqrt{d_n(x)}r_n(y)} \end{bmatrix}
$$

$$
B = L_n(x)^{-1} L_n(y)^{-1} \boldsymbol{y}_n - \frac{y(1-y)}{r_n(x)r_n(y)} \boldsymbol{x}_n
$$

$$
= \boldsymbol{v}_n(x, y) + \sqrt{\frac{d_n(y)}{d_n(x)}} C_n(x)^{-1} \boldsymbol{x}_n - \frac{y(1-y)}{r_n(x)r_n(y)} \boldsymbol{x}_n
$$

$$
= \boldsymbol{v}_n(x, y) - \frac{y(1-y)}{r_n(x)r_n(y)} \boldsymbol{x}_n + \sqrt{\frac{d_n(y)}{d_n(x)}} \frac{1}{p_{n-1}(x)} \boldsymbol{x}_n
$$

$$
A = \begin{bmatrix} \boldsymbol{v}_n(x, y) - \frac{y(1-y)}{r_n(x)r_n(y)} \boldsymbol{x}_n + \sqrt{\frac{d_n(y)}{d_n(x)}} \frac{1}{p_{n-1}(x)} \boldsymbol{x}_n \\ \frac{y(1-y)p_{n-1}(x)}{r_n(x)r_n(y)} \end{bmatrix}
$$

$$
\boldsymbol{v}_{n+1}(x, y) = A - \begin{bmatrix} \sqrt{\frac{d_{n+1}(y)}{d_{n+1}(x)}} \frac{1}{p_n(x)} \boldsymbol{x}_n \\ \sqrt{\frac{d_{n+1}(y)}{d_{n+1}(x)}} \frac{x}{p_n(x)} \end{bmatrix}
$$

$$
= \begin{bmatrix} \boldsymbol{v}_n(x, y) - \left( \frac{y(1-y)}{r_n(x)r_n(y)} + \sqrt{\frac{d_{n+1}(y)}{d_{n+1}(x)}} \frac{1}{p_n(x)} - \sqrt{\frac{d_n(y)}{d_n(x)}} \frac{1}{p_{n-1}(x)} \right) \boldsymbol{x}_n \\ \frac{y(1-y)p_{n-1}(x)}{r_n(x)r_n(y)} - \sqrt{\frac{d_{n+1}(y)}{d_{n+1}(x)}} \frac{x}{p_n(x)} \end{bmatrix}
$$

Using the definition of from Lemma 3, Lemma 4 and Definition 2 we have that:

$$
\begin{aligned}
\boldsymbol{v}_1 &= [g_0]^T \\
\boldsymbol{v}_2 &= [g_0 - f_1, g_1]^T \\
\boldsymbol{v}_3 &= [g_0 - f_1 - f_2, g_1 - f_2, g_2]^T \\
&\cdots
\end{aligned}
$$

$$\boldsymbol{v}_n = \left[ g_0 - \sum_{i=1}^{n-1} f_i, g_1 - \sum_{i=2}^{n-1} f_i, \ldots, g_k - \sum_{i=k+1}^{n-1} f_i, \ldots \right]^T$$

$$= [g_0 - h_0^n, g_1 - h_1^n, \ldots, g_k - h_k^n, \ldots]^T$$

Or in other words we have that:

$$[\boldsymbol{v}_n(x, y)]_k = g_k(x, y) - h_k^n(x, y)$$

Thus proving the lemma reduces to proving that:

$$g_k(x, y) - h_k^n(x, y) \geqslant 0 \quad \forall k, n$$

Expanding on the equation that we need to prove is positive:

$$\mathcal{L}_k^n(x, y) = g_k(x, y) - h_k^{n+1}(x, y) = g_k(x, y) - \sum_{i=k+1}^{n} f_i(x, y)$$

$$= \frac{y(1-y)p_{k-1}(x)}{r_k(x)r_k(y)} - \sqrt{\frac{d_{k+1}(y)}{d_{k+1}(x)}} \frac{x}{p_k(x)}$$

$$- \sum_{i=k+1}^{n} \left( \frac{xy(1-y)}{r_i(x)r_i(y)} + \sqrt{\frac{d_{i+1}(y)}{d_{i+1}(x)}} \frac{x}{p_i(x)} - \sqrt{\frac{d_i(y)}{d_i(x)}} \frac{x}{p_{i-1}(x)} \right)$$

$$= \frac{y(1-y)p_{k-1}(x)}{r_k(x)r_k(y)} - \sum_{i=k+1}^{n} \frac{xy(1-y)}{r_i(x)r_i(y)} - B$$

$$B = \sqrt{\frac{d_{k+1}(y)}{d_{k+1}(x)}} \frac{x}{p_k(x)} + \sum_{i=k+1}^{n} \sqrt{\frac{d_{i+1}(y)}{d_{i+1}(x)}} \frac{x}{p_i(x)} - \sum_{i=k+1}^{n} \sqrt{\frac{d_i(y)}{d_i(x)}} \frac{x}{p_{i-1}(x)}$$

$$= \sum_{i=k}^{n} \sqrt{\frac{d_{i+1}(y)}{d_{i+1}(x)}} \frac{x}{p_i(x)} - \sum_{i=k}^{n-1} \sqrt{\frac{d_{i+1}(y)}{d_{i+1}(x)}} \frac{x}{p_i(x)}$$

$$= \sqrt{\frac{d_{n+1}(y)}{d_{n+1}(x)}} \frac{x}{p_n(x)}$$

$$\mathcal{L}_k^n(x, y) = \frac{y(1-y)p_{k-1}(x)}{r_k(x)r_k(y)} - \sum_{i=k+1}^{n} \frac{xy(1-y)}{r_i(x)r_i(y)} - \sqrt{\frac{d_{n+1}(y)}{d_{n+1}(x)}} \frac{x}{p_n(x)}$$

Now we turn our attention to the sum in the middle:

$$B = \sum_{i=k+1}^{n} \frac{xy(1-y)}{r_i(x)r_i(y)} = xy(1-y) \sum_{i=k+1}^{n} \frac{1}{r_i(x)r_i(y)}$$

$$= xy\sqrt{\frac{1-y}{1-x}} \sum_{i=k+1}^{n} \frac{\sqrt{(1-x)(1-y)}}{r_i(x)r_i(y)}$$

Using Cacuhy $-$ Schwartz inequality :

$$\left( \sum_{i=k+1}^{n} \frac{\sqrt{(1-x)(1-y)}}{r_i(x)r_i(y)} \right)^2 \leqslant \left( \sum_{i=k+1}^{n} \frac{1-x}{r_i(x)^2} \right) \left( \sum_{i=k+1}^{n} \frac{1-y}{r_i(y)^2} \right)$$

Let's define the partial sum in the brackets as:

$$I_k^n(x) = \sum_{i=k}^{n} \frac{1-x}{r_i(x)^2} = \sum_{i=k}^{n} \frac{1-x}{(1-x)p_i(x)p_{i-1}(x)} = \sum_{i=k}^{n} \frac{1}{p_i(x)p_{i-1}(x)}$$

We will prove by induction that:

$$I_k^n(x) = \frac{n-k+1}{p_{k-1}(x)p_n(x)}$$

First for $n = k$, we have that $I_n^n(x) = \frac{1}{p_n(x)p_{n-1}(x)}$ which clearly satisfies the above equation. Assuming this is true for $n$, we will now show it holds for $n + 1$:

$$
\begin{aligned}
I_k^{n+1}(x) &= \sum_{i=k}^{n+1} \frac{1}{p_n(x)p_{n-1}(x)} = I_k^n(x) + \frac{1}{p_n(x)p_{n+1}(x)} \\
&= \frac{n-k+1}{p_{k-1}(x)p_n(x)} + \frac{1}{p_n(x)p_{n+1}(x)} \\
&= \frac{p_{n+1}(x)(n-k+1) - p_{k-1}(x)}{p_{k-1}(x)p_n(x)p_{n+1}(x)} = \frac{((n+1)x+1)(n-k+1) + ((k-1)x+1)}{p_{k-1}(x)p_n(x)p_{n+1}(x)} \\
&= \frac{x(n^2 - nk + n + n - k + 1 + k - 1) + n - k + 1 + 1}{p_{k-1}(x)p_n(x)p_{n+1}(x)} \\
&= \frac{x(n^2 - nk + 2n) + n - k + 2}{p_{k-1}(x)p_n(x)p_{n+1}(x)} = \frac{xn(n - k + 2) + n - k + 2}{p_{k-1}(x)p_n(x)p_{n+1}(x)} \\
&= \frac{(n-k+2)(nx+1)}{p_{k-1}(x)p_n(x)p_{n+1}(x)} = \frac{n-k+2}{p_{k-1}(x)p_{n+1}(x)}
\end{aligned}
$$

Which concludes the proof. This now means that:

$$
B \leqslant xy\sqrt{\frac{1-y}{1-x}}\sqrt{I_{k+1}^n(x)I_{k+1}^n(y)} = xy\sqrt{\frac{1-y}{1-x}}\sqrt{\frac{(n-k)^2}{p_k(x)p_n(x)p_k(y)p_n(y)}}
$$

Thus we can conclude that

$$
\mathcal{L}_k^n(x,y) \geqslant \frac{y(1-y)p_{k-1}(x)}{r_k(x)r_k(y)} - xy\sqrt{\frac{1-y}{1-x}}\sqrt{\frac{(n-k)^2}{p_k(x)p_n(x)p_k(y)p_n(y)}} - \sqrt{\frac{d_{n+1}(y)}{d_{n+1}(x)}}\frac{x}{p_n(x)}
$$

From the fact that $h_k^n(x,y) \geqslant 0$ it impliest that $\mathcal{L}_k^n$ is a decreasing function of $n$, hence to we only need to show that for any fixed $x$ and $y$ $\mathcal{L}_k^\infty(x,y)$ is positive.

For this, we need to take the limit of the second and third term in the above equation.

$$
\begin{aligned}
A &= \lim_{n\to\infty} \sqrt{\frac{d_{n+1}(y)}{d_{n+1}(x)}}\frac{x}{p_n(x)} = \lim_{n\to\infty} \sqrt{\frac{(1-y)\frac{p_{n+1}(y)}{p_n(y)}}{(1-x)\frac{p_{n+1}(x)}{p_n(x)}}}\frac{x}{p_n(x)} \\
&= \lim_{n\to\infty} \sqrt{\frac{x^2(1-y)p_{n+1}(y)}{(1-x)p_{n+1}(x)p_n(x)p_n(y)}} \\
&= \sqrt{\frac{x^2(1-y)}{(1-x)}}\sqrt{\lim_{n\to\infty}\frac{p_{n+1}(y)}{p_{n+1}(x)p_n(x)p_n(y)}} = 0
\end{aligned}
$$

where the last line is true since the denominator is of higher degree in $n$.

Taking the limit of the second term corresponds to computing the limit:

$$
A = \lim_{n\to\infty}\frac{n-k}{p_n(x)} = \text{im}_{n\to\infty}\frac{n-k}{nx+1} = \frac{1}{x}
$$

Hence, this means that:

$$
\begin{aligned}
\mathcal{L}_k^\infty(x,y) &\geqslant \frac{y(1-y)p_{k-1}(x)}{r_k(x)r_k(y)} - xy\sqrt{\frac{1-y}{1-x}}\sqrt{\frac{1}{p_k(x)p_k(y)xy}} \\
&= \frac{y(1-y)p_{k-1}(x)}{\sqrt{(1-x)p_k(x)p_{k-1}(x)(1-y)p_k(y)p_{k-1}(y)}} - \sqrt{xy}\sqrt{\frac{1-y}{1-x}}\sqrt{\frac{1}{p_k(x)p_k(y)}} \\
&= y\sqrt{\frac{1-y}{1-x}}\frac{\sqrt{p_{k-1}(x)}}{\sqrt{p_k(x)p_k(y)p_{k-1}(y)}} - \sqrt{xy}\sqrt{\frac{1-y}{1-x}}\sqrt{\frac{1}{p_k(x)p_k(y)}}
\end{aligned}
$$

$$= \sqrt{y}\sqrt{\frac{1-y}{1-x}}\sqrt{\frac{1}{p_k(x)p_k(y)p_{k-1}(y)}}\left(\sqrt{yp_{k-1}(x)} - \sqrt{xp_{k-1}(y)}\right)$$

$$A = yp_{k-1}(x) - xp_{k-1}(y) = yx(k-1) + y - xy(k-1) - x = y - x \geqslant 0$$

$$\Rightarrow \mathcal{L}_k^\infty(x,y) \geqslant 0$$

**Note:**

All of the limits must be taken for fixed $x$ and $y$ (e.g. we can't have them approach 0 or 1 simultanously with $n$).

$$\mathcal{L}_k^n(x,y) \geqslant \frac{y(1-y)p_{k-1}(x)}{r_k(x)r_k(y)} - xy\sqrt{\frac{1-y}{1-x}}\sqrt{\frac{(n-k)^2}{p_k(x)p_n(x)p_k(y)p_n(y)}} - \sqrt{\frac{d_{n+1}(y)}{d_{n+1}(x)}}\frac{x}{p_n(x)}$$

$$= A - (B + C)$$

$$A = \frac{y(1-y)p_{k-1}(x)}{r_k(x)r_k(y)} = \frac{y(1-y)p_{k-1}(x)}{\sqrt{(1-x)p_k(x)p_{k-1}(x)}\sqrt{(1-y)p_k(y)p_{k-1}(y)}}$$

$$= y\sqrt{\frac{1-y}{1-x}}\frac{1}{\sqrt{p_k(x)p_k(y)}}\sqrt{\frac{p_{k-1}(x)}{p_{k-1}(y)}}$$

$$C = \sqrt{\frac{d_{n+1}(y)}{d_{n+1}(x)}}\frac{x}{p_n(x)} = \sqrt{\frac{(1-y)\frac{p_{n+1}(y)}{p_n(y)}}{(1-x)\frac{p_{n+1}(x)}{p_n(x)}}}\frac{x}{p_n(x)}$$

$$= x\sqrt{\frac{(1-y)p_{n+1}(y)}{(1-x)p_n(y)p_n(x)p_{n+1}(x)}} = x\sqrt{\frac{1-y}{1-x}}\sqrt{\frac{p_{n+1}(y)}{p_n(y)p_n(x)p_{n+1}(x)}}$$

$$= x\sqrt{\frac{1-y}{1-x}}\frac{1}{\sqrt{p_k(x)p_k(y)}}\sqrt{\frac{p_k(x)p_k(y)p_{n+1}(y)}{p_n(x)p_n(y)p_{n+1}(x)}}$$

$$A - B - C = \sqrt{\frac{1-y}{1-x}}\frac{1}{\sqrt{p_k(x)p_k(y)}}\left(y\sqrt{\frac{p_{k-1}(x)}{p_{k-1}(y)}} - xy\sqrt{\frac{(n-k)^2}{p_n(x)p_n(y)}} - x\sqrt{\frac{p_k(x)p_k(y)p_{n+1}(y)}{p_n(x)p_n(y)p_{n+1}(x)}}\right)$$

$$= \sqrt{\frac{1-y}{1-x}}\frac{A - B - C}{\sqrt{p_k(x)p_k(y)p_{k-1}(y)p_n(x)p_n(y)p_{n+1}(x)}}$$

$$A = y\sqrt{p_{k-1}(x)p_n(x)p_{n+1}(x)p_n(y)}$$

$$B = xy(n-k)\sqrt{p_{k-1}(x)p_{n+1}(x)} = x$$

$$C = x\sqrt{p_k(x)p_k(y)p_{k-1}(y)p_{n+1}(y)}$$

$$A^2 = y^2(kx - x + 1)(nx + 1)(ny + 1)(nx + 1)$$

$$B^2 = x^2y^2(n-k)(kx - x + 1)(nx + x + 1)$$

$$C^2 = x^2(kx + 1)(ky + 1)(ky - y + 1)(ny + y + 1)$$

$$A = xy\sqrt{\frac{1-y}{1-x}}\sqrt{\frac{(n-k)^2}{p_k(x)p_n(x)p_k(y)p_n(y)}} +$$

$$B = \sqrt{\frac{d_{n+1}(y)}{d_{n+1}(x)}}\frac{x}{p_n(x)} = \sqrt{\frac{(1-y)\frac{p_{n+1}(y)}{p_n(y)}}{(1-x)\frac{p_{n+1}(x)}{p_n(x)}}}\frac{x}{p_n(x)}$$

$$= x\sqrt{\frac{(1-y)p_{n+1}(y)}{(1-x)p_n(y)p_n(x)p_{n+1}(x)}} = x\sqrt{\frac{1-y}{1-x}}\sqrt{\frac{p_{n+1}(y)}{p_n(y)p_n(x)p_{n+1}(x)}}$$

$$C = \sqrt{xy}\sqrt{\frac{1-y}{1-x}}\sqrt{\frac{1}{p_k(x)p_k(y)}}$$

$$C - A = \sqrt{xy}\sqrt{\frac{1-y}{1-x}}\sqrt{\frac{1}{p_k(x)p_k(y)}}\left(1 - \frac{\sqrt{xy}(n-k)}{\sqrt{p_n(x)p_n(y)}}\right)$$

$$1 \pm R^2 = 1 - \frac{xy(n-k)^2}{p_n(x)p_n(y)} = \frac{p_n(x)p_n(y) \pm xy(n-k)^2}{p_n(x)p_n(y)}$$

$$= \frac{n^2xy + nx + ny + 1 \pm xy(n-k)^2}{p_n(x)p_n(y)} = \frac{xyk(n^2 \pm (n-k)^2) + 1}{p_n(x)p_n(y)} \geqslant 0$$

$$(1-R)^2 = 1 + R^2 - 2R$$

$$(C-A)^2 - B^2 = xy\frac{1-y}{1-x}\frac{1}{p_k(x)p_k(y)}(1 + R^2 - 2R) - x^2\frac{1-y}{1-x}\frac{p_{n+1}(y)}{p_n(y)p_n(x)p_{n+1}(x)}$$

$$= x\frac{1-y}{1-x}\frac{xyk(n^2+(n-k)^2)+1}{p_k(x)p_k(y)p_n(x)p_n(y)} - x^2\frac{1-y}{1-x}\frac{p_{n+1}(y)}{p_n(y)p_n(x)p_{n+1}(x)} -$$

$$xy\frac{1-y}{1-x}\frac{2R}{p_k(x)p_k(y)}$$

$$= x\frac{1-y}{1-x}\frac{x}{p_n(x)p_n(y)p_k(x)p_k(y)p_{n+1}(x)}K - xy\frac{1-y}{1-x}\frac{2R}{p_k(x)p_k(y)}$$

$$K = p_{n+1}(x)(yk(n^2+(n-k)^2)+1) - xp_{n+1}(y)p_k(x)p_k(y)$$

$$= (nx+x+1)y(2n^2-2nk+k^2) - x(ny+1)(kx+1)(ky+1)$$

$$C - (A+B) = \sqrt{xy}\sqrt{\frac{1-y}{1-x}}\sqrt{\frac{1}{p_k(x)p_k(y)}}\left(1 - \frac{\sqrt{xy}(n-k)}{\sqrt{p_n(x)p_n(y)}} - \sqrt{\frac{xp_{n+1}(y)p_k(x)p_k(y)}{yp_n(y)p_n(x)p_{n+1}(x)}}\right)$$

$$= \sqrt{xy}\sqrt{\frac{1-y}{1-x}}\sqrt{\frac{1}{p_k(x)p_k(y)}}\left(1 - \sqrt{\frac{x}{p_n(x)p_n(y)}}(A+B)\right)$$

$$(A+B)^2 = \left(\sqrt{x}(n-k) + \sqrt{\frac{p_{n+1}(y)p_k(x)p_k(y)}{yp_{n+1}(x)}}\right)^2$$

$$= x(n-k)^2 + \frac{p_{n+1}(y)p_k(x)p_k(y)}{yp_{n+1}(x)} + 2\sqrt{x}(n-k)\sqrt{\frac{p_{n+1}(y)p_k(x)p_k(y)}{yp_{n+1}(x)}}$$

$$= \frac{xy(n-k)^2p_{n+1}(x) + p_{n+1}(y)p_k(x)p_k(y)}{yp_{n+1}(x)} +$$

$$\frac{2\sqrt{xy}(n-k)\sqrt{p_{n+1}(y)p_{n+1}(x)p_k(x)p_k(y)}}{yp_{n+1}(x)}$$

$$1 - (A+B)^2 = \frac{yp_{n+1}(x) - xy(n-k)^2p_{n+1}(x) - p_{n+1}(y)p_k(x)p_k(y) - 2E}{yp_{n+1}(x)}$$

$$E = \sqrt{xy}(n-k)\sqrt{p_{n+1}(y)p_{n+1}(x)p_k(x)p_k(y)}$$

### I.2.5 PROOF OF LEMMA 3

From the definition of $g_n(x,y)$ and the fact that $p_n(x)$ and $r_n(x)$ are non-negative functions we can conclude that:

$$g_n(x,y) \geqslant 0 \Leftrightarrow \left(\frac{y(1-y)p_{n-1}(x)}{r_n(x)r_n(y)}\right)^2 - \left(\sqrt{\frac{d_{n+1}(y)}{d_{n+1}(x)}}\frac{x}{p_n(x)}\right)^2 \geqslant 0$$

Denoting with $A$ the left hand side of the second equation we have:

$$A = \frac{y^2(1-y)^2p_{n-1}(x)^2}{r_n(x)^2r_n(y)^2} - \frac{d_{n+1}(y)}{d_{n+1}(x)}\frac{x^2}{p_n(x)^2}$$

$$
= \frac{y^2(1-y)^2 p_{n-1}(x)^2}{(1-x)p_n(x)p_{n-1}(x)(1-y)p_n(y)p_{n-1}(y)} - \frac{(1-y)\frac{p_{n+1}(y)}{p_n(y)}}{(1-x)\frac{p_{n+1}(x)}{p_n(x)}} \frac{x^2}{p_n(x)^2}
$$

$$
= \frac{(1-y)y^2 p_{n-1}(x)}{(1-x)p_n(x)p_n(y)p_{n-1}(y)} - \frac{(1-y)x^2 p_{n+1}(y)}{(1-x)p_{n+1}(x)p_n(x)p_n(y)}
$$

$$
= \frac{(1-y)}{(1-x)p_n(x)p_n(y)} \left( \frac{y^2 p_{n-1}(x)}{p_{n-1}(y)} - \frac{x^2 p_{n+1}(y)}{p_{n+1}(x)} \right)
$$

$$
= \frac{(1-y)B}{(1-x)p_n(x)p_n(y)p_{n-1}(y)p_{n+1}(x)}
$$

$$
B = y^2 p_{n-1}(x)p_{n+1}(x) - x^2 p_{n+1}(y)p_{n-1}(y)
$$

$$
= y^2((nx+1)-x)((nx+1)+x) - x^2((ny+1)-y)((ny+1)+y)
$$

$$
= y^2(nx+1)^2 - y^2 x^2 - x^2(ny+1)^2 + x^2 y^2
$$

$$
= (y(nx+1) + x(ny+1))(y(nx+1) - x(ny+1))
$$

$$
= (2nxy + x + y)(y - x)
$$

$$
A = (y-x)\frac{(1-y)(2nxy+x+y)}{(1-x)p_n(x)p_n(y)p_{n-1}(y)p_{n+1}(x)}
$$

Hence for $x \leqslant y$ we have that $A \geqslant 0 \Leftrightarrow g_n(x,y) \geqslant 0$.

### I.2.6 PROOF OF LEMMA 4

$$
f_n(x,y) = \frac{xy(1-y)}{r_n(x)r_n(y)} + \sqrt{\frac{d_{n+1}(y)}{d_{n+1}(x)}} \frac{x}{p_n(x)} - \sqrt{\frac{d_n(y)}{d_n(x)}} \frac{x}{p_{n-1}(x)} = A + B - C
$$

First we will prove that $C - B \geqslant 0$

$$
C^2 - B^2 = \frac{(1-y)\frac{p_n(y)}{p_{n-1}(y)}}{(1-x)\frac{p_n(x)}{p_{n-1}(x)}} \frac{x^2}{p_{n-1}(x)^2} - \frac{(1-y)\frac{p_{n+1}(y)}{p_n(y)}}{(1-x)\frac{p_{n+1}(x)}{p_n(x)}} \frac{x^2}{p_n(x)^2}
$$

$$
= x^2 \frac{1-y}{1-x} \left( \frac{p_n(y)}{p_{n-1}(y)p_n(x)p_{n-1}(x)} - \frac{p_{n+1}(y)}{p_n(y)p_{n+1}(x)p_n(x)} \right)
$$

$$
= x^2 \frac{1-y}{1-x} \frac{1}{p_n(x)} \frac{p_n(y)^2 p_{n+1}(x) - p_{n+1}(y)p_{n-1}(y)p_{n-1}(x)}{p_{n-1}(x)p_n(y)p_{n-1}(y)}
$$

$$
A = p_n(y)^2 p_{n+1}(x) - p_{n+1}(y)p_{n-1}(y)p_{n-1}(x)
$$

$$
= p_n(y)^2(p_n(x) + x) - (p_n(x) - x)(p_n(y) + y)(p_n(y) - y)
$$

$$
= p_n(y)^2 p_n(x) + x p_n(y)^2 - (p_n(x) - x)(p_n(y)^2 - y^2)
$$

$$
= p_n(y)^2 p_n(x) + x p_n(y)^2 - p_n(y)^2 p_n(x) + y^2 p_n(x) - x p_n(y)^2 + xy^2
$$

$$
= y^2 p_n(x) + xy^2 \geqslant 0
$$

$$
\Rightarrow C \geqslant B
$$

With this we can now conclude that $f_n(x,y) \geqslant 0 \Leftrightarrow A^2 - (C-B)^2 \geqslant 0$

$$
L = A^2 - C^2 - B^2 + 2BC
$$

$$
2L_1 = A^2 - C^2 - B^2
$$

$$
= \frac{x^2 y^2 (1-y)^2}{(1-x)p_n(x)p_{n-1}(x)(1-y)p_n(y)p_{n-1}(y)} -
$$

$$
x^2 \frac{1-y}{1-x} \frac{1}{p_n(x)} \left( \frac{p_n(y)}{p_{n-1}(y)p_{n-1}(x)} + \frac{p_{n+1}(y)}{p_n(y)p_{n+1}(x)} \right)
$$

$$
= x^2 \frac{1-y}{1-x} \frac{1}{p_n(x)} \left( \frac{y^2}{p_{n-1}(x)p_n(y)p_{n-1}(y)} - \frac{p_n(y)}{p_{n-1}(y)p_{n-1}(x)} - \frac{p_{n+1}(y)}{p_n(y)p_{n+1}(x)} \right)
$$

$$
= x^2 \frac{1-y}{1-x} \frac{1}{p_n(x)} \left( \frac{y^2 - p_n(y)^2}{p_{n-1}(x)p_n(y)p_{n-1}(y)} - \frac{p_{n+1}(y)}{p_n(y)p_{n+1}(x)} \right)
$$

$$
\begin{aligned}
&= & x^2 \frac{1-y}{1-x} \frac{1}{p_n(x)} \left( \frac{-p_{n+1}(y)p_{n-1}(y)}{p_{n-1}(x)p_n(y)p_{n-1}(y)} - \frac{p_{n+1}(y)}{p_n(y)p_{n+1}(x)} \right) \\
&= & -x^2 \frac{1-y}{1-x} \frac{1}{p_n(x)} \left( \frac{p_{n+1}(y)}{p_{n-1}(x)p_n(y)} + \frac{p_{n+1}(y)}{p_n(y)p_{n+1}(x)} \right) \\
&= & -x^2 \frac{1-y}{1-x} \frac{p_{n+1}(y)}{p_n(x)p_n(y)} \frac{(p_{n+1}(x) + p_{n-1}(x))}{p_{n-1}(x)p_{n+1}(x)} \\
&= & -x^2 \frac{1-y}{1-x} \frac{p_{n+1}(y)}{p_n(x)p_n(y)} \frac{(2nx+2)}{p_{n-1}(x)p_{n+1}(x)} = -2x^2 \frac{1-y}{1-x} \frac{p_{n+1}(y)}{p_n(x)p_n(y)} \frac{p_n(x)}{p_{n-1}(x)p_{n+1}(x)} \\
&= & -2x^2 \frac{1-y}{1-x} \frac{p_{n+1}(y)}{p_n(y)p_{n-1}(x)p_{n+1}(x)} \\
\Rightarrow & & L \geqslant 0 \Leftrightarrow B^2 C^2 - L_1^2 \geqslant 0
\end{aligned}
$$

$$
\begin{aligned}
B^2 C^2 - L_1^2 &= & \left( x^2 \frac{1-y}{1-x} \right)^2 \frac{1}{p_n(x)^2} \frac{p_n(y)p_{n+1}(y)}{p_{n-1}(y)p_{n-1}(x)p_n(y)p_{n+1}(x)} - \\
& & \left( x^2 \frac{1-y}{1-x} \right)^2 \frac{p_{n+1}(y)^2}{p_n(y)^2 p_{n-1}(x)^2 p_{n+1}(x)^2} \\
&= & \left( x^2 \frac{1-y}{1-x} \right)^2 \left( \frac{p_{n+1}(y)}{p_{n-1}(y)p_{n-1}(x)p_{n+1}(x)p_n(x)^2} - \frac{p_{n+1}(y)^2}{p_n(y)^2 p_{n-1}(x)^2 p_{n+1}(x)^2} \right) \\
&= & \left( x^2 \frac{1-y}{1-x} \right)^2 \frac{p_{n+1}(y)D}{p_{n-1}(y)p_{n-1}(x)^2 p_{n+1}(x)^2 p_n(x)^2 p_n(y)^2} \\
D &= & p_{n-1}(x)p_{n+1}(x)p_n(y)^2 - p_{n+1}(y)p_{n-1}(y)p_n(x)^2 \\
&= & p_n(y)^2 (p_n(x)^2 - x^2) - (p_n(y)^2 - y^2)p_n(x)^2 \\
&= & p_n(y)^2 p_n(x)^2 - x^2 p_n(y)^2 - p_n(y)^2 p_n(x)^2 + y^2 p_n(x)^2 \\
&= & y^2 p_n(x)^2 - x^2 p_n(y)^2 = y^2(nx+1)^2 - x^2(ny+1)^2 \\
&= & nx^2 y^2 + 2nxy^2 + y^2 - nx^2 y^2 - 2nx^2 y - x^2 \\
&= & 2nxy(y-x) + (y-x)(y+x) \\
&= & (y-x)(2nxy + x + y) \geqslant 0
\end{aligned}
$$

Hence with this we can conclude that $f_n(x,y) \geqslant 0 \quad \forall n$.

