# OpenReview forum: "Deep Transformers without Shortcuts: Modifying Self-attention for Faithful Signal Propagation"
_ICLR.cc/2023/Conference — ICLR 2023 poster_

### Official Review · Reviewer_QFbf · 2022-10-24

**Confidence:** 3
**Correctness:** 3
**Technical Novelty And Significance:** 2
**Empirical Novelty And Significance:** 3
**Recommendation:** 8

**Clarity, Quality, Novelty And Reproducibility:**

For the most part, the paper and the experiments communicate that it is at least possible to train Transformers without skip connections + normalization layers. The justification that this works in Transformers is novel.

**Strength And Weaknesses:**

**Strengths**

- (+ +) The proposed solutions truly do enable transformers without skip connections to train at larger depths
- (+) The paper is clearly laid out and explained, though I did not leave understanding all the math.

**Weaknesses**

- (- -) From a practical point of view, there is little incentive to use the proposed methods. They cannot handle duplicate tokens in input without hacks. It is not evident that the proposed methods offer any real advantages over training with skip connections either from the computational efficiency or interpretability standpoint.
- (-) The authors only consider causal masked attention which ensures a lower-triangular Attention matrix, a requisite for their analysis, which does not cover the broad spectrum of data on which Transformers are applied.
- (-) Doubling and tripling the number of learnable parameters (in depth by adding more blocks) of a skipless Transformer does not bring noticeable increase in performance at even 1000k training steps.
- (-) No code is mentioned to be released with this submission, though the implementation details in Appendix G are thorough.

Having stated these weaknesses, the authors are not trying to create a practical alternative but instead to provide theoretical proof and empirical justification that it is at least possible to train Transformers without skip connections. To this end the authors succeed, though the paper does not improve our understanding of the roles that normalization and skip connections play.

Questions

- Pg 4: The notation $(\mathbf{A}_l)_l$ is not described when it is introduced. Why the double index?
- Why does Figure 2 come after Figure 3 in the text?
- Are Fig 1 and 7 created from a Transformer *******trained******* on a task, or a randomly initialized Transformer?

**Summary Of The Paper:**

Paper tries to answer: what do skip connections and normalisation layers do? Can we train without them?

The authors attempt to analyze how our dependency on normalization and skip connections can be removed in deep NNs (specifically, in the self-attention operation of transformers). They do this by porting the “Deep Kernel Shaping” (DKS) technique to Transformer attention, with a few modifications that result in 3 proposed methods: *Value-SkipInit*, Uniform SPA,  and *Exponential SPA*. All of these help train more vanilla Transformers to varying degrees of effectiveness.

**Summary Of The Review:**

The authors show that you can theoretically train the Transformer (attention + MLP) without skip connections. However, their methods and assumptions are difficult to apply in the multitude of domains that Transformers are used, and there is no evidence that skip-less or normalization free Transformers are more meaningful than the original Transformer. I recommend this paper as a borderline accept.

I did not read the appendices in depth, nor am I particularly familiar with the literature surrounding DKS methods in Transformers.

---

> ### Author Response · Authors · 2022-11-18
> **Response to Reviewer QFbf 1/3**
>
> We thank reviewer QFbf for their comments and are pleased that they found our work to be clearly laid out and novel. We address the following concerns below:
>
> i) **Duplicate tokens in inputs** To account for the fact that it is not possible to know the positions of duplicate tokens in a sequence, we propose to control the evolution of the average input kernel matrix over sequences for SPA methods in Appendix B (of updated draft). By the linearity of the output kernel of deep skipless transformers in terms of the input kernel (Eqn 6), this means our modifications also control the average output kernel. Though controlling the average output kernel is an approximation in our SPA methods, we respectfully disagree that this constitutes a hack. It should be added that Value-SkipInit maintains identity attention and is unaffected by duplicate tokens.
>
> Moreover, we stress that the general principle that our methods follow is independent of the input kernel matrix (i.e. independent of duplicate input tokens): we seek to prevent the product of attention matrices from deviating away from the identity matrix and degenerating to a rank-1 matrix. If the product of attention-matrices is rank-1 then regardless of the input kernel we will have a rank-1 output kernel i.e. rank collapse (Dong et al. 2021) which prevents trainability (Noci et al. 2022). OTOH, if we control the deviation of the attention matrix product from the identity then no matter the input kernel, the output kernel will bear some similarity to the input kernel. So as long as the input kernel is non-degenerate and has full rank (regardless of duplicate tokens) so too will be the output kernel. We have updated our appendix B with this discussion, which we hope clarifies things.
>
> ii) **Practical advantages** While we are not advocating skipless transformers to be an immediately practical method, we do see potential practical value for the community in the future. Many works (Oyedotun et al. 2020; Ding et al. 2021; Zhang et al. 2022) have highlighted drawbacks to multi-branch residual topologies including memory inefficiencies, as inputs to residual blocks need to be kept in memory until the final addition during inference. (Shortcut branches account for about 40% of the memory usage in a ResNet-50.) Either removing skips completely, or designing new topologies of transformer architectures with reduced memory costs, while still matching the performance of standard Transformers would hold significant practical value to the community. Our work represents a potential first step towards this goal.
>
> We also believe our results on transformers with skips demonstrate practical advantages. Since submission we have improved the performance of our E-SPA transformers on C4 in Table 2, by widening our grid-search over normalised skip weights. Without LayerNorm we now achieve 24.7 validation perplexity, and with LayerNorm we achieve 24.0, whereas the default transformer achieves 24.7 (lower is better). Moreover, in Table 2 we show that E-SPA outperforms previous works that have tried to remove LayerNorm in deep Transformers: 1) SkipInit/ReZero (concurrently proposed by De and Smith, 2020 and Bachlechner et al., 2021) which achieves 35.3 without LayerNorm, and 2) Stable ResNet (proposed by Hayou et al., 2021 for MLP/CNNs and by Noci et al., 2022 for Transformers) which achieves 25.3 perplexity without LayerNorm.
>
> Thus our experiments on C4 show that our methods allow us to: 1) remove normalisation layers in transformers, with no loss of performance, and 2) improve the performance of default transformers (with normalisation layers and skip connections) through better initialisation. Similar conclusions on WikiText-103 can be taken from Figure 3. Both of these findings should be of immediate practical interest.

---

> > ### Author Response · Authors · 2022-11-18
> > **Response to Reviewer QFbf 2/3**
> >
> > iii) **Causal masking** We focus on causal masked attention as it is a popular setting for transformers and is a more difficult setting to work with from a signal propagation point of view due to the lower-triangular attention matrices. However, we add that our methods are either already compatible with, or can be straightforwardly modified to be compatible with, other forms of attention. To start with, Value-SkipInit does not modify the softmax-attention computation itself and is applicable to any form of attention. For our SPA methods, it is straightforward to extend to non-causal attention by changing L in equation 9 to be the (symmetric) matrix square root instead of being the Cholesky matrix. In this case, for U-SPA it is clear that $A_l$ in Equation 8 will still be non-negative as the matrix square root, inverses and products of uniform kernel matrices are all still uniform. For E-SPA, we have empirically verified that the resulting $A_l$ will be non-negative. We have added Appendix A to our draft to detail these extensions to non-causal attention.
> >
> > Finally, we add that it is not uncommon for published papers on transformers to focus purely on causal masked attention, e.g. the ALiBi positional encoder (Press et al. 2022) which was published at ICLR last year.
> >
> > iv) **Depth scaling** The purpose of our depth scaling experiments is to demonstrate that our theoretical justification does enable vanilla transformers to train at very large depths. Similar results have been shown for vanilla CNNs, e.g. Xiao et al. 2018 who trained 10,000-layer CNNs, which we view as an impressive feat despite the fact that the 10,000-layer CNN trained a lot slower than a shallower 32-layer CNN (Figure 3 of Xiao et al. 2018). Having said this, we agree with reviewer QFbf that improving the depth scaling for our skipless transformers is interesting future work.
> >
> > v) **Code** We present pseudocode for how to construct an E-SPA multi-head attention layer in Algorithm 1 in Appendix D.
> >
> > vi) **Understanding the roles of normalisation and skip connections** We respectfully disagree with reviewer QFbf that our work does not improve our understanding of the roles that normalisation and skip connections play in a transformer. By removing skips and normalisation, we expose the beneficial interactions that these components have when combined with the attention module, which are crucial to enable transformers to be trainable. Having isolated these interactions, we then show how we would need to modify the attention module itself in order to account for the lack of skips/normalisation.
> >
> > In particular, given that all existing transformers use skip connections (to the best of our knowledge), it was conceivable that the attention module is simply not compatible with skipless architectures. We believe such a dependence on skip/normalisation is problematic as it muddies the roles of the different components in an architecture: do transformers work because self-attention works, or do they only work because self-attention works in combination with skips/normalisation?
> >
> > Our work finds that it is possible to remove skip connections and normalisation layers from the default transformer architecture, if we simultaneously improve the signal propagation through the self-attention module. At a high level, we achieve this by modifying the attention layer to ensure it is closer to the identity at initialisation, which has been argued to be a key benefit of Pre-LN skip connections and normalisation layers (Balduzzi et al., 2017; Xiao et al., 2018; De and Smith, 2020; Martens et al., 2021; Zhang et al., 2022). This is crucial to avoid rank collapse (Dong et al., 2021) which prevents trainability (Noci et al., 2022). However we also extend beyond this central insight, by developing the E-SPA technique which incorporates a bias towards nearby tokens into the attention weights at initialization.
> >
> > Our experiments demonstrate that our modifications at initialisation allow deep skipless transformers to be trainable for the first time, but at reduced speeds. This highlights that the initialisation-time benefit of skip connections can be replaced by better initialisation of the self-attention module to enable trainable deep skipless transformers, but also that there are further optimisation benefits with skips during training for transformers, mirroring the case for CNNs (Martens et al., 2021; Zhang et al., 2022). To our knowledge, no existing theory comprehensively explains this optimisation benefit of skip connections in deep Neural Networks (NNs), and this is an interesting and important area for future work.

---

> > > ### Author Response · Authors · 2022-11-18
> > > **Response to Reviewer QFbf 3/3**
> > >
> > > vii) **Literature on DKS for transformers** We would like to point out that prior to our work there was no existing literature on DKS for transformers. As discussed in section 1, this is because the evolution of the high dimensional kernel matrices for transformers is significantly more difficult to analyse and control than the lower dimensional recursions that occur for MLPs/CNNs, which DKS was designed for. Moreover, transformers suffer from a distinct, but related, form of signal propagation degeneration known as rank collapse (Dong et al., 2021), compared to MLPs/CNNs. As such, a distinct approach to DKS is needed to enable controlled signal propagation in skipless transformers, and we propose 3 such approaches based off a general set of requirements, in section 3.
> > >
> > > **Additional questions**
> > > - The double index in $(A_l)_l$ denotes the sequence of all attention matrices, across depths.
> > > - Figure 2 comes after Figure 3 in the text as Figure 3 is placed at the top of the page 8 in order to save space, whereas FIgure 2 is inline. We discuss Figure 2 first in our writing.
> > > - Fig 1 and 7 are created from a randomly initialised Transformer, when query-key dot product initialised to 0 (which is the case either by our construction, or is the large $d^k$ limit if one uses a $\frac{1}{d^k}$ rather than $\frac{1}{\sqrt{d^k}}$ scaling in computing the dot product in Eq. 10). All projection/value parameters are initialised randomly as orthogonal matrices, which enables closed-form computation of the kernel matrix evolution. We outline how we do so in Appendix H.2.
> > >
> > > We hope our response has satisfied Reviewer QFbf’s concerns and if so, that they may consider raising their score. Otherwise, we are happy to respond to any further questions.

---

> > > > ### Comment · Reviewer_QFbf · 2022-12-06
> > > > **Feedback Response**
> > > >
> > > > I thank the authors for their clarifications to my objections to the paper. My concerns have been mostly addressed — while I am still not convinced that the field should pursue skipless+norm-free transformers, I can see the value in seeking to understand the Transformer paradigm without them. I have increased my score accordingly.

---

> > > > > ### Author Response · Authors · 2022-12-06
> > > > > **Thank you!**
> > > > >
> > > > > Thanks for replying with your feedback and score update, we really appreciate it!

---

### Official Review · Reviewer_sotB · 2022-10-25

**Confidence:** 4
**Correctness:** 4
**Technical Novelty And Significance:** 3
**Empirical Novelty And Significance:** 4
**Recommendation:** 6

**Clarity, Quality, Novelty And Reproducibility:**

The paper is well written. The attention initialization introduced in this work is novel and the underlying idea behind the initialization is simple and easy to read.

**Strength And Weaknesses:**

The major strength of the paper is the extensive set of experimentations to showcase that transformers can indeed be trained without skip connections and normalization layers. Such a study is important for the community to understand the necessity of each component inside the transformer. Overall, this paper shows that we simply need a good initialization to train a transformer.


However, I have the following questions:

a) How do the trained transformer (on C4) with E-SPA perform on simple downstream task like SQUAD or SuperGLUE tasks? Does achieving the same pre-training loss with E-SPA also lead to similar performance as a default transformer in downstream tasks?


b) During the training of vanilla transformer with E-SPA, how do the attention matrices behave?  Do the position biases stay dominant term for most of the attention matrices in the transformer? If so, then it might explain why the vanilla transformer continues to train well during training, without the usage of skip connections.

c) How fragile is the performance of vanilla transformer with E-SPA ? Is a small learning rate necessary for the transformer to train? For all figures, do you plot the best performing model across the different hyperparameters? How do you select the best performing model across different hyperparameters?

d) What does "In" denote in the legend of Figure 3? Moreover, how should I read the plot? Is it that at $\alpha=0.99$ E-SPA has the same train loss as the default transformer after $100K$ steps? Also, why did the authors decide to compare the train loss at $100K$ step?


Furthermore, what is the motivation to measure the performance of a standard transformer with normalized skip connections? Can the authors comment on the necessity of normalization for normalized skip connection in the standard transformer?

e) Does the standard transformer (transformer with skip connections and layer normalization) work with E-SPA initialization? Is there a change in the rate of decrease of the training loss?



**Summary Of The Paper:**

The authors show that with a careful initialization of parameters, we can train skipless attention-only models without skip connections or normalization layers when trained longer. The major idea is to maintain the signal propagation at initialization in an attention-only model. The authors corroborate their claims with an experimental study on WikiText-103 and C4.

**Summary Of The Review:**

Overall, my score is on the positive side. The experiments show that the model can achieve similar training loss as the default transformer, even without skip connections and layer normalization but with longer training. However, the question remains on whether good pretraining performance transfers to downstream tasks.  Hence, I would ask the authors to think about the questions I have posted above during the rebuttal.

---

> ### Author Response · Authors · 2022-11-18
> **Response to Reviewer sotB 1/2**
>
> We thank reviewer sotB for their comments and are pleased that they found our work to be important for the community, and our experiments showing the trainability of skipless and normalisation-less transformers to be extensive. We address the following questions below:
>
> a) **Downstream tasks** We took the final checkpoints of both the standard transformer and our skipless models pretrained on C4 (Figure 4a) and b)) and then evaluated them on five Common Sense downstream tasks. The results are presented in the table below (percentage accuracies are reported, higher is better), where in brackets denotes the number of training steps for each model. These datasets have been commonly used to evaluate pretrained language models, included GPT-3 (arXiv:2005.14165), Megatron-Turing NLG (arXiv:2201.11990), Gopher (arXiv:2112.11446) and Chinchilla (arXiv:2203.15556). Our pretrained models were evaluated without fine-tuning, in a zero-shot manner.
>
> |       Model      | BoolQ | HellaSwag | Winogrande | PIQA | SIQA |
> |:----------------:|:-----:|:---------:|------------|------|------|
> |   Default (50K)  |  60.9 |    29.1   | 52.5       | 62.2 | 41.9 |
> |    E-SPA (50K)   |  59.5 |    26.2   | 51.3       | 59.7 | 39.8 |
> |    U-SPA (50K)   |  39.2 |    25.8   | 50.9       | 56.7 | 40.6 |
> | V-SkipInit (50K) |  38.7 |    25.8   | 51.6       | 56.0 | 39.0 |
> |   E-SPA (200K)   |  60.2 |    27.7   | 52.8       | 63.1 | 40.9 |
> |   E-SPA (300K)   |  56.8 |    28,4   | 53.0       | 62.5 | 41.1 |
>
> We see that the conclusions of Figures 4a) and b) on C4 largely carry over to the downstream setting:
>
> - Among transformers trained for the same number of steps (50k), E-SPA beats U-SPA and V-SkipInit on 4 out of 5 downstream tasks. However, the standard transformer outperforms skipless transformers on all tasks with the same amount of training.
> - With ~5x longer training (200K and 300K steps), E-SPA achieves similar performance on downstream tasks to the standard transformer, outperforming on 2 out of 5 tasks (Winogrande and PIQA).
>
> b) **Evolution of attention matrix experiment** Thank you for this suggestion. We agree that this is an interesting experiment and have included it as Figure 15 of Appendix G. Overall we see significant changes to the kernel matrix during training, suggesting that the biases do not remain dominant. That being said, the kernel matrices continue to be somewhat diagonally dominant, which agrees with results for transformers with skip connections (https://arxiv.org/abs/2211.03495, section 5.1 and figure 4).
>
> c) **Hyperparameters/fragility** For all experiments we reported the best performing model across all hyperparameters (detailed in Appendix H.2), tuned to optimise training speed (i.e. lowest training loss after fixed number of training steps). The learning rate hyperparameter was tuned on a logarithmic scale, and because we optimised for training speed, we typically observed that the chosen learning rate was (close to) the maximum stable learning rate (that reduced training loss and did not produce NaNs). The reviewer sotB is correct that the chosen learning rates for vanilla transformers were typically smaller, by a factor of around 10 (e.g. 5e-5 vs 5e-4) compared to those chosen by standard transformers, which accounts for much of the training speed gap we observe. For the unmodified skipless transformers in Figure 2 that were untrainable and did not decrease in training loss, we extended our search space for learning rates down to very small values, e.g. 1e-8, to no avail.

---

> > ### Author Response · Authors · 2022-11-18
> > **Response to Reviewer sotB 2/2**
> >
> > d) **Figure 3** In Figure 3, the captions should read ‘LN=False’ and ‘LN=True’ where ‘LN’ stands for layer normalisation. We apologise for the confusion: we used small caps in the submission but have updated this to large caps in our draft. The reviewer sotB is correct that the plot tells us that at $\alpha=0.99$ E-SPA matches the training speed of the default transformer (after 100K steps), both with and without layer normalisation. We chose 100K steps for WikiText-103 by default as it was sufficiently long to have a clear picture of training performance across methods, but also not prohibitively long to bottleneck our results. For C4, we reduced our default training steps to 50K.
> >
> > **Standard transformers with normalised skips in Fig 3** In Figure 3, we measure the performance of a standard transformer with normalised skip connections rather than the standard Pre-LN transformer because the standard Pre-LN transformer suffers from rank collapse and is untrainable if one removes the normalisation layers (Noci et al. 2022). This allows us to highlight E-SPA suffers from little to no degradation in training speed when removing LN, across different shortcut weights $\alpha$, whereas the standard transformer with normalised skips observes a consistent worsening in performance without LN.  Moreover, this comparison in Figure 3 highlights that normalised skips connections with standard attention do allow one to train both with and without LayerNorm, but only if the shortcut weight
> > $\alpha$ is large enough. This can be seen in Figure 3 (right), where for small shortcut weights no points are plotted as those models failed to train at all, with or without LN. The reason for this goes back to the initialisation benefits of downweighting residual branches, as seen in e.g. Hayou et al., 2021 and Bachlechner et al., 2021, which happens naturally in Pre-LN networks (De and Smith, 2020). On the other hand, transformers with E-SPA train for all values of the shortcut weights $\alpha.$
> >
> > We add that without LN, standard Pre-LN transformers suffer both from rank collapse, and exploding activation norms (Noci et al., 2021), and fail to train. This is because the normalisation layers are crucial to downweighting the residual branch in Pre-LN architectures (De and Smith, 2020).
> >
> > e) **Standard transformer with E-SPA initialisation**  We have not experimented with standard Pre-LN transformers with our E-SPA method, although this could be interesting to try. Instead, when combining skip connections with E-SPA we used normalised skips in order to control the diagonal values of our kernel matrices (i.e. the activation norms). This was motivated by the fact that we seek to prevent exploding or shrinking kernel values at large depths, and also the fact that for the MLP sub-blocks of transformers, Deep Kernel Shaping (Martens et al. 2022) and Tailored Activation Transformations (Zhang et al. 2022) require the diagonals of kernel matrices to be uniform equal to 1 across layers.
> >
> > We hope our response has satisfied Reviewer sotB’s concerns and if so, that they may consider raising their score. Otherwise, we are happy to respond to any further questions.

---

> > > ### Author Response · Authors · 2022-12-08
> > > **Follow up before end of discussion period**
> > >
> > > Dear reviewer sotB, please do let us know if you had a chance to see our author response and have any further comments or feedback. Thanks again!

---

### Official Review · Reviewer_2AHB · 2022-10-25

**Confidence:** 3
**Correctness:** 3
**Technical Novelty And Significance:** 4
**Empirical Novelty And Significance:** 2
**Recommendation:** 8

**Clarity, Quality, Novelty And Reproducibility:**

The models (and baselines) need to be evaluated on downstream tasks such as SuperGLUE. Eval perplexity is insufficient to make claims about model utility.

I think the authors could do a better job justifying the case for why skip- and normalization-free transformers are important.

The authors mention
> As second order optimisers for transformers are not well established…

This does not to me seem like a reason to avoid using second order optimizers. On the contrary, it seems like a very good reason to try them here. It would be a meaningful result if they substantially reduced training time!

There’s a large body of literature on interventions to improve the training stability of vanilla (norm + skip) transformers that is not discussed. Is none of this work relevant at all? Examples include: Davis et al., 2021, Catformer; Liu et al., 2020, Understanding the Difficulty of Training Transformers; Xu et al., 2020, Lipschitz Constrained Parameter Initialization for Deep Transformers; Touvron et al., 2021, Going Deeper with Image Transformers; Zhang et al., 2019, Improving Deep Transformer with Depth-Scaled Initialization; Huang et al., 2020, Improving Transformer Optimization through Better Initialization

I assume the quantity displayed in Table 1 is WT103 train loss, but that should be explicit.


**Strength And Weaknesses:**

**Strengths**

Extensive theoretical groundwork to justify design choices

Promising incremental advance for understanding and improving transformer stability

Able to train deep skipless transformers!

**Weaknesses**

No downstream evaluation

The motivation for skip and normalization-free transformers could be better justified

No immediate practical use

Light on related work


**Summary Of The Paper:**

**Note: Score updated from 5 to 8 after author response**

The authors develop methods to train deep transformers that lack skip connections and/or normalization layers. They achieve this through the following theoretically-motivated interventions: modified initialization, bias matrices, and location-dependent scaling. They are able to train skip connection, normalization-free networks to the same quality as standard transformers (though requiring 5x the number of steps), to within 0.3 perplexity of a standard transformer in the same amount of time, and to depths of 108 layers without divergence.

**Summary Of The Review:**

Meaningful progress towards stable training of normalization- and skip-free transformers, but not well-justified why we want normalization- and skip-free transformers. No downstream evaluation, and related work section feels light.

---

> ### Author Response · Authors · 2022-11-18
> **Response to Reviewer 2AHB 1/2**
>
> We thank reviewer 2AHB for their comments and are pleased that they found our theoretical groundwork to be extensive, and our proposals to train deep skipless transformers to be ‘meaningful progress towards stable training of normalization- and skip-free transformers’. We address the following concerns below:
>
> i) **Downstream tasks** Thank you for this suggestion. We took the final checkpoints of both the standard transformer and our skipless models pretrained on C4 (Figure 4a) and b)) and then evaluated them on five Common Sense downstream tasks. The results are presented in the table below (percentage accuracies are reported, higher is better), where in brackets denotes the number of training steps for each model. These datasets have been commonly used to evaluate pretrained language models, included GPT-3 (arXiv:2005.14165), Megatron-Turing NLG (arXiv:2201.11990), Gopher (arXiv:2112.11446) and Chinchilla (arXiv:2203.15556). Our pretrained models were evaluated without fine-tuning, in a zero-shot manner.
>
> |       Model      | BoolQ | HellaSwag | Winogrande | PIQA | SIQA |
> |:----------------:|:-----:|:---------:|------------|------|------|
> |   Default (50K)  |  60.9 |    29.1   | 52.5       | 62.2 | 41.9 |
> |    E-SPA (50K)   |  59.5 |    26.2   | 51.3       | 59.7 | 39.8 |
> |    U-SPA (50K)   |  39.2 |    25.8   | 50.9       | 56.7 | 40.6 |
> | V-SkipInit (50K) |  38.7 |    25.8   | 51.6       | 56.0 | 39.0 |
> |   E-SPA (200K)   |  60.2 |    27.7   | 52.8       | 63.1 | 40.9 |
> |   E-SPA (300K)   |  56.8 |    28,4   | 53.0       | 62.5 | 41.1 |
>
>
> We see that the conclusions of Figures 4a) and b) on C4 largely carry over to the downstream setting:
>
> - Among transformers trained for the same number of steps (50k), E-SPA beats U-SPA and V-SkipInit on 4 out of 5 downstream tasks. However, the standard transformer outperforms skipless transformers on all tasks with the same amount of training.
> - With ~5x longer training (200K and 300K steps), E-SPA achieves similar performance on downstream tasks to the standard transformer, outperforming on 2 out of 5 tasks (Winogrande and PIQA).
>
> ii) **Motivation for skip and normalisation-free transformers** Our primary motivation for skip and normalisation-free transformers is to better understand the roles of different components of the standard transformer architecture, and in particular how the skip connections/normalisation layers interact with the self-attention module. As Reviewer sotB writes: ‘Such a study is important for the community to understand the necessity of each component inside the transformer’.
>
> To improve our understanding of these architectural components, we ask whether they can be replaced, and how we would need to modify the self-attention module itself in order to account for the lack of skips/normalisation. In particular, given that all existing transformers use skip connections (to the best of our knowledge), it was conceivable that the self-attention module is simply not compatible with skipless architectures. We believe such a dependence on skip/normalisation is problematic as it muddies the roles of the different components in an architecture: do transformers work because self-attention works, or do they only work because self-attention works in combination with skips/normalisation?
>
> Our work finds that it is possible to remove skip connections and normalisation layers from the default transformer architecture, if we simultaneously improve the signal propagation through the self-attention module. At a high level, we achieve this by modifying the attention layer to ensure it is closer to the identity at initialisation, which has been argued to be a key benefit of Pre-LN skip connections and normalisation layers (Balduzzi et al., 2017; Xiao et al., 2018; De and Smith, 2020; Martens et al., 2021; Zhang et al., 2022). This is crucial to avoid rank collapse (Dong et al., 2021) which prevents trainability (Noci et al., 2022). However we also extend beyond this central insight, by developing the E-SPA technique which incorporates a bias towards nearby tokens into the attention weights at initialization.
>
> Our experiments demonstrate that our modifications at initialisation allow deep skipless transformers to be trainable for the first time, but at reduced speeds. This highlights that the initialisation-time benefit of skip connections can be replaced by better initialisation of the self-attention module to enable trainable deep skipless transformers, but also that there are further optimisation benefits with skips during training for transformers, mirroring the case for CNNs (Martens et al., 2021; Zhang et al., 2022). To our knowledge, no existing theory comprehensively explains this optimisation benefit of skip connections in deep Neural Networks (NNs), and this is an interesting and important area for future work.

---

> > ### Author Response · Authors · 2022-11-18
> > **Response to Reviewer 2AHB 2/2**
> >
> > ii) **Motivation for skip and normalisation-free transformers (continued)** Finally, we note that skip and normalisation layers add computational and memory overheads to the transformer. For skips, many works (Oyedotun et al. 2020; Ding et al. 2021; Zhang et al. 2022) have highlighted memory inefficiencies of multi-branch residual topologies, as inputs to residual blocks need to be kept in memory until the final addition (shortcut branches account for about 40% of the memory usage in a ResNet-50). Either removing skips completely, or designing new topologies of transformer architectures with reduced memory costs, while still matching the performance of standard Transformers would hold significant practical value to the community. Our work represents a potential first step towards this goal.
> >
> > We have revised the introduction in our draft to better highlight our motivation.
> >
> > iii) **No immediate practical use** We respectfully disagree that our results show *no* immediate practical value. Since submission we have improved the performance of our E-SPA transformers on C4 in Table 2, by widening our grid-search over normalised skip weights. Without LayerNorm we now achieve 24.7 validation perplexity, and with LayerNorm we achieve 24.0, whereas the default transformer achieves 24.7 (lower is better). Moreover, in Table 2 we show that E-SPA outperforms previous works that have tried to remove LayerNorm in deep Transformers: 1) SkipInit/ReZero (concurrently proposed by De and Smith 2020 and Bachlechner et al., 2021) which achieves 35.3 without LayerNorm, and 2) Stable ResNet (proposed by Hayou et al., 2021 for MLP/CNNs and by Noci et al., 2022 for Transformers) which achieves 25.3 perplexity without LayerNorm.
> >
> > Thus our experiments on C4 show that our methods allow us to: 1) remove normalisation layers in transformers, with no loss of performance, and 2) improve the performance of default transformers (with normalisation layers and skip connections) through better initialisation. Similar conclusions on WikiText-103 can be taken from Figure 3. Both of these findings should be of immediate practical interest.
> >
> > iv) **Related work**  We thank reviewer 2AHB for informing us of these related works, which we have added citations for in our updated draft. In terms of how these works relate to ours, these works are similar to related papers we discuss in our original submission (De and Smith, 2020; Bachlechner et al, 2021; Wang et al. 2022; Noci et al., 2022 etc). This is because they propose to stabilise transformer training by downweighting the residual branch of a residual architecture at initialisation, which has known benefits for signal propagation (Hayou et al., 2021). The exception is CatFormer (Davis et al. 2021) which uses concatenation as an alternative combination operation (rather than addition in a skip) to mirror this effect. Without skips (or any form of combination), we show how we can replicate these initialisation benefits using only the self-attention module, which is crucial for avoiding rank collapse and ensuring trainability.
> >
> > v) **Second order optimisers** Since submission, we tried a preliminary set of experiments with K-FAC (Martens et al., 2015) but haven’t seen performance improvements so far. We add that we view second order optimisers for transformers (with or without skips) to be somewhat orthogonal to, and outside the scope of, the main focus of our submission. We agree that applying second order optimisers to transformers is an interesting future research direction.
> >
> > **Additional questions** In Table 1 the quantity displayed is WT103 train loss. Thank you for pointing this out, we have updated our draft to make this clearer.
> >
> > We hope our response has satisfied Reviewer 2AHB’s concerns and if so, that they may consider raising their score. Otherwise, we are happy to respond to any further questions.

---

> > > ### Comment · Reviewer_2AHB · 2022-11-18
> > > **Great response!**
> > >
> > > Thanks for your thorough response! The motivation, value, and evaluation all seem much stronger to me. I've updated my score from a 5 to a 8!

---

> > > > ### Author Response · Authors · 2022-11-21
> > > > **Thank you!**
> > > >
> > > > Thanks for the reply and update, and again for your helpful feedback. We're pleased that our response has addressed your concerns!

---

### Official Review · Reviewer_HAdV · 2022-11-01

**Confidence:** 3
**Correctness:** 3
**Technical Novelty And Significance:** 4
**Empirical Novelty And Significance:** Not applicable
**Recommendation:** 6

**Clarity, Quality, Novelty And Reproducibility:**

As the skip connection and normalisation layers have been demonstrated by dozens of theories and applications, the application value of this paper is still not clear. For example, the performance of the proposed method in 'shallow' transformers? How many hyper-parameters should be tuned to reach a good performance? Is it also the case for images?

**Strength And Weaknesses:**

Strength:
This paper challenged the well-known skip connection and normalisation layers and tried to provide a new way of training deep transformers.

Weakness:
Converge slow, 5 times more iterations in fact huge in practice.
The proposed training framework is more complex compared with skip connection and normalisation layers, means more inductive bias may be introduced to reach the same performance.

**Summary Of The Paper:**

This paper tried to train the deep transformer without skip connection and/or normalisation layers.
This paper tried to train the deep transformer directly by combining parameter initialisation, bias matrices, and location-dependent rescaling in the signal propagation view.
The performance indicates that though the proposed methods converge slowly compared to the previous method of using skip connection and normalisation layers, it could potentially reach the matching performance with enough training cost.

**Summary Of The Review:**

This submission provides an interesting way of training deep transformers without skip connection and normalisation layers.

---

> ### Author Response · Authors · 2022-11-18
> **Response to Reviewer HAdV 1/2**
>
> We thank reviewer HAdV for their comments and are pleased that they found the ideas in our work to be interesting. We address the following concerns below:
>
> i) **Slow convergence without skips / unclear application value** The primary goal of our work is to better understand the different components of the standard transformer architecture, particularly skip connections/normalisation layers and their interaction with the self-attention module, as opposed to proposing a method to train skipless transformers with immediate practical utility. As Reviewer sotB writes: ‘Such a study is important for the community to understand the necessity of each component inside the transformer’.
>
> Having said that, we respectfully disagree that our results show *no* immediate practical value. Since submission we have improved the performance of our E-SPA transformers on C4 in Table 2, by widening our grid-search over normalised skip weights. Without LayerNorm we now achieve 24.7 validation perplexity, and with LayerNorm we achieve 24.0, whereas the default transformer achieves 24.7 (lower is better). Moreover, in Table 2 we show that E-SPA outperforms previous works that have tried to remove LayerNorm in deep Transformers: 1) SkipInit/ReZero (concurrently proposed by De and Smith 2020 and Bachlechner et al., 2021) which achieves 35.3 without LayerNorm, and 2) Stable ResNet (proposed by Hayou et al., 2021 for MLP/CNNs and by Noci et al., 2022 for Transformers) which achieves 25.3 perplexity without LayerNorm.
>
> Thus our experiments on C4 show that our methods allow us to: 1) remove normalisation layers in transformers, with no loss of performance, and 2) improve the performance of default transformers (with normalisation layers and skip connections) through better initialisation. Similar conclusions on WikiText-103 can be taken from Figure 3. Both of these findings should be of immediate practical interest.
>
> From the ‘understanding’ side, we provide a positive answer to the question: ‘Can deep transformers be trained without skip connections (and normalisation layers)’? Given that all existing transformers use skip connections (to the best of our knowledge), it was conceivable that the self-attention module is simply not compatible with skipless architectures. We believe such a dependence on skip/normalisation is problematic as it muddies the roles of the different components in an architecture: do transformers work because self-attention works, or do they only work because self-attention works in combination with skips/normalisation?
>
> Our work finds that it is possible to remove skip connections and normalisation layers from the default transformer architecture, if we simultaneously improve the signal propagation through the self-attention module. At a high level, we achieve this by modifying the attention layer to ensure it is closer to the identity at initialisation, which has been argued to be a key benefit of Pre-LN skip connections and normalisation layers (Balduzzi et al., 2017; Xiao et al., 2018; De and Smith, 2020; Martens et al., 2021; Zhang et al., 2022). This is crucial to avoid rank collapse (Dong et al., 2021) which prevents trainability (Noci et al., 2022). However we also extend beyond this central insight, by developing the E-SPA technique which incorporates a bias towards nearby tokens into the attention weights at initialization.
>
> We demonstrate empirically that our modifications at initialisation allow deep skipless transformers to be trainable for the first time, but at reduced speeds. This highlights that the initialisation-time benefit of skip connections can be replaced by better initialisation of the self-attention module to enable trainable deep skipless transformers, but also that there are further optimisation benefits with skips during training for transformers, mirroring the case for CNNs (Martens et al., 2021; Zhang et al., 2022). To our knowledge, no existing theory comprehensively explains this optimisation benefit of skip connections in deep Neural Networks (NNs), and this is an interesting and important area for future work.
>
> Finally, while our skipless transformers train slower at the moment, we believe that they may hold application value for the community in the future. Many works (Oyedotun et al. 2020; Ding et al. 2021; Zhang et al. 2022) have highlighted drawbacks to multi-branch residual topologies including memory inefficiencies, as inputs to residual blocks need to be kept in memory until the final addition (shortcut branches account for about 40% of the memory usage in a ResNet-50). Either removing skips completely, or designing new topologies of transformer architectures with reduced memory costs, while still matching the performance of standard Transformers would hold significant practical value to the community. Our work represents a potential first step towards this goal.

---

> > ### Author Response · Authors · 2022-11-18
> > **Response to Reviewer HAdV 2/2**
> >
> > ii) **More complex inductive bias** We respectfully disagree that our training frameworks are more complex than the standard transformer. Firstly, the fixed pre-softmax bias matrices in our SPA methods are similar to those found in the popular ‘Attention with Linear Biases’ or ALiBi positional encoder (Press et al. 2022), particularly for E-SPA which also takes an exponential decaying form as seen in Theorem 3. We note that positional encodings provide an inductive bias that is essential to achieve good performance in default transformers. Secondly, our methods and standard transformers all obtain closely matching train *and* test losses (in Figures 4a and 13), whereas one would expect a better inductive bias to achieve lower test loss at the same train loss.
> >
> > **Additional comments**  We ran initial experiments on shallower transformers of depth 12 or 18 (where our methods worked like in the deeper case), but chose to focus on larger depths (30+) in our submission, which is standard when studying signal propagation as it is more challenging to train NNs at large depth (as seen in Figure 2, where skipless transformers fail without our modifications).
> >
> > We focused on the language modelling setting to empirically verify our theoretically motivated proposals, as this is arguably the most popular and important application area for transformers. Having said this, we believe it should be relatively straightforward to extend our techniques to transformers for image classification e.g. vision transformers: the primary adjustment is to extend our SPA methods for non-causal masking attention (Value-SkipInit is already compatible), which we detail in the updated Appendix A.
> >
> > We provide all details regarding hyperparameter tuning in Appendix H. For convenience, in all experiments we tuned the learning rate on a logarithmic scale. For our SPA methods we tuned the final decay rate $\gamma_L$ (E-SPA), and the final uniform off-diagonal values $\rho_L$ (U-SPA), on a small grid of values, but found the default values of $\gamma_L=0.005$ and $\rho_L=0.8$ to work well across different training settings (varying training lengths, use of LN, use of DKS/TAT, etc) and datasets for skipless transformers. No additional hyperparameters are required for Value-SkipInit.
> >
> > We hope our response has satisfied Reviewer HAdV’s concerns and if so, that they may consider raising their score. Otherwise, we are happy to respond to any further questions.

---

> > > ### Author Response · Authors · 2022-12-08
> > > **Follow up before end of discussion period**
> > >
> > > Dear reviewer HAdV, please do let us know if you had a chance to see our author response and have any further comments or feedback. Thanks again!

---

### Author Response · Authors · 2022-11-18
**Overall response and updated draft**

We would like to thank all reviewers for their effort in reviewing our work, the comments have been very helpful to improve our work and we really appreciate it!

We were pleased to see all reviewers recognise our work as the first to train deep transformers without skip connections (and normalisation layers), and in particular that both our: i) theoretical groundwork (Reviewer 2AHB) and ii) experiments (Reviewer sotB) were deemed to be ‘extensive’.

There are 3 main concerns raised by the reviewers. These are: 1) the motivation for skip and normalisation-free transformers, 2) the immediate practical value of our work, and 3) whether our conclusions regarding pretraining performance are mirrored in downstream tasks.

For the reviewers’ and meta-reviewer’s convenience, we repeat our response to these 3 main concerns in the thread below.

Also, we have uploaded a new draft to OpenReview, with changes highlighted in red. The main updates are:

- The first few paragraphs of section 1 to better highlight our motivation.
- Updated related work in section 2.
- 5 common sense downstream tasks added in Table 6.
- Improved results in Table 2 for E-SPA with and without LN. Without LN, we now match the default Pre-LN transformer in terms of training speed.

---

> ### Author Response · Authors · 2022-11-18
> **Main concerns 1/2**
>
> i) **Motivation for skip and normalisation-free transformers** (Reviewers HAdV, 2AHB, QFbf)
>
> Our primary motivation for skip and normalisation-free transformers is to better understand the roles of different components of the standard transformer architecture, and in particular how the skip connections/normalisation layers interact with the self-attention module. As Reviewer sotB writes: ‘Such a study is important for the community to understand the necessity of each component inside the transformer’.
>
> To improve our understanding of these architectural components, we ask whether they can be replaced, and how we would need to modify the self-attention module itself in order to account for the lack of skips/normalisation. In particular, given that all existing transformers use skip connections (to the best of our knowledge), it was conceivable that the self-attention module is simply not compatible with skipless architectures. We believe such a dependence on skips/normalisation is problematic as it muddies the roles of the different components in an architecture: do transformers work because self-attention works, or do they only work because self-attention works in combination with skips/normalisation?
>
> Our work finds that it is possible to remove skip connections and normalisation layers from the default transformer architecture, if we simultaneously improve the signal propagation through the self-attention module. At a high level, we achieve this by modifying the attention layer to ensure it is closer to the identity at initialisation, which has been argued to be a key benefit of Pre-LN skip connections and normalisation layers (Balduzzi et al., 2017; Xiao et al., 2018; De and Smith, 2020; Martens et al., 2021; Zhang et al., 2022). This is crucial to avoid rank collapse (Dong et al., 2021) which prevents trainability (Noci et al., 2022). However we also extend beyond this central insight, by developing the E-SPA technique which incorporates a bias towards nearby tokens into the attention weights at initialization.
>
> From our experiments, we see that our modifications at initialisation allow deep skipless transformers to be trainable for the first time, but at reduced speeds. This highlights that the initialisation-time benefit of skip connections can be replaced by better initialisation of the self-attention module to enable trainable deep skipless transformers, but also that there are further optimisation benefits with skips during training of transformers, mirroring the case for CNNs (Martens et al., 2021; Zhang et al., 2022). To our knowledge, no existing theory comprehensively explains this optimisation benefit of skip connections in deep Neural Networks (NNs), and this is an interesting and important area for future work.
>
> Finally, we note that skip and normalisation layers add computational and memory overheads to the transformer. For skips, many works (Oyedotun et al. 2020; Ding et al. 2021; Zhang et al. 2022) have highlighted memory inefficiencies of multi-branch residual topologies, as inputs to residual blocks need to be kept in memory until the final addition (shortcut branches account for about 40% of the memory usage in a ResNet-50). Either removing skips completely, or designing new topologies of transformer architectures with reduced memory costs, while still matching the performance of standard Transformers, would hold significant practical value to the community. Our work represents a potential first step towards this goal.

---

> > ### Author Response · Authors · 2022-11-18
> > **Main concerns 2/2**
> >
> > ii) **Immediate practical value** (Reviewers HAdV, 2AHB, QFbf)
> >
> > Reviewer QFbf writes: ‘the authors are not trying to create a practical alternative but instead to provide theoretical proof and empirical justification that it is at least possible to train Transformers without skip connections.’ This is a sentiment we mostly agree with.
> >
> > Having said that, we respectfully disagree that our results show *no* immediate practical value. Since submission we have improved the performance of our E-SPA transformers on C4 in Table 2, by widening our grid-search over normalised skip weights. Without LayerNorm we now achieve 24.7 validation perplexity, and with LayerNorm we achieve 24.0, whereas the default transformer achieves 24.7 (lower is better). Moreover, in Table 2 we show that E-SPA outperforms previous works that have tried to remove LayerNorm in deep Transformers: 1) SkipInit/ReZero (proposed by De and Smith 2020 and Bachlechner et al., 2021) which achieves 35.3 without LayerNorm, and 2) Stable ResNet (proposed by Hayou et al., 2021 for MLP/CNNs and by Noci et al., 2022 for Transformers) which achieves 25.3 perplexity without LayerNorm.
> >
> > Thus our experiments on C4 show that our methods allow us to: 1) remove normalisation layers in transformers, with no loss of performance, and 2) improve the performance of default transformers (with normalisation layers and skip connections) through better initialisation. Similar conclusions on WikiText-103 can be taken from Figure 3. Both of these findings should be of immediate practical interest.
> >
> > We also believe our work could act as an initial step towards realising future practical benefits such as more efficient architecture design, as discussed in the 'motivation' point above.
> >
> > iii) **Downstream tasks** (Reviewers 2AHB, sotB)
> >
> > We took the final checkpoints of both the standard transformer and our skipless models pretrained on C4 (Figure 4a) and b)) and then evaluated them on five Common Sense downstream tasks. The results are presented in the table below (percentage accuracies are reported, higher is better), where in brackets denotes the number of training steps for each model. These datasets have been commonly used to evaluate pretrained language models, included GPT-3 (arXiv:2005.14165), Megatron-Turing NLG (arXiv:2201.11990), Gopher (arXiv:2112.11446) and Chinchilla (arXiv:2203.15556). Our pretrained models were evaluated without fine-tuning, in a zero-shot manner.
> >
> > |       Model      | BoolQ | HellaSwag | Winogrande | PIQA | SIQA |
> > |:----------------:|:-----:|:---------:|------------|------|------|
> > |   Default (50K)  |  60.9 |    29.1   | 52.5       | 62.2 | 41.9 |
> > |    E-SPA (50K)   |  59.5 |    26.2   | 51.3       | 59.7 | 39.8 |
> > |    U-SPA (50K)   |  39.2 |    25.8   | 50.9       | 56.7 | 40.6 |
> > | V-SkipInit (50K) |  38.7 |    25.8   | 51.6       | 56.0 | 39.0 |
> > |   E-SPA (200K)   |  60.2 |    27.7   | 52.8       | 63.1 | 40.9 |
> > |   E-SPA (300K)   |  56.8 |    28,4   | 53.0       | 62.5 | 41.1 |
> >
> >
> > We see that the conclusions of Figures 4a) and b) on C4 largely carry over to the downstream setting:
> >
> > - Among transformers trained for the same number of steps (50k), E-SPA beats U-SPA and V-SkipInit on 4 out of 5 downstream tasks. However, the standard transformer outperforms skipless transformers on all tasks with the same amount of training.
> > - With ~5x longer training (200K and 300K steps), E-SPA achieves similar performance on downstream tasks to the standard transformer, outperforming on 2 out of 5 tasks (Winogrande and PIQA).

---

### Decision · Program_Chairs · 2023-01-20

**Decision:**

Accept: poster

**Justification For Why Not Higher Score:**

The negatives are that the exercise is more intellectual at this point as these types of transformers require five times as many iterations.

**Justification For Why Not Lower Score:**

The positives of the paper are that it is neat to see a deep transformer work without parts that are considered standard.

**Metareview: Summary, Strengths And Weaknesses:**

This paper develops methods to build transformers without residuals or normalizations using ideas from faithful signal propagation.  Their techniques include initialization and bias matrices. The positives of the paper are that it is neat to see a deep transformer work without parts that are considered standard. The negatives are that the exercise is more intellectual at this point as these types of transformers require five times as many iterations. This was still cool to see. The reviewers are all positive; a few increased their score after the author rebuttal.

**Note From Pc:**

if the above contains the word "oral" or "spotlight" please see: "oral" presentation means -> notable-top-5% and "spotlight" means -> notable-top-25%. As stated in our emails, we are disassociating presentation type from AC recommendations